# Nanomedicine-based co-delivery of a calcium channel inhibitor and a small molecule targeting CD47 for lung cancer immunotherapy

Yuedong Guo [1,2], Qunqun Bao [3], Ping Hu [1,3] ✉ & Jianlin Shi [1,3] ✉

Pro-tumoral macrophages in lung tumors present a significant challenge in immunotherapy. Here, we introduce a pH-responsive nanomedicine approach for activating anti-tumoral macrophages and dendritic cells. Using a layered double hydroxide nanosheet carrier, we co-deliver a T-type calcium channel inhibitor (TTA-Q6) and a CD47 inhibitor (RRX-001) into lung tumors. In the tumor acidic environment, TTA-Q6 is released, disrupting cancer cell calcium uptake, causing endoplasmic reticulum stress and inducing calreticulin transfer to the cell surface. Surface calreticulin activates macrophages and triggers dendritic cell maturation, promoting effective antigen presentation and therefore activating antitumor T cells. Simultaneously, RRX-001 reduces CD47 protein levels, aiding in preventing immune escape by calreticulin-rich cancer cells. In lung tumor models in male mice, this combined approach shows anti-tumor effects and immunity against tumor re-exposure, highlighting its potential for lung cancer immunotherapy.

According to the latest statistics from the International Agency for Research on Cancer, lung cancer has the second-highest incidence, and is one of the leading causes of cancer-related mortality worldwide, with 2.1 million new lung cancer cases and 1.8 million deaths annually[1]. In the last decade, despite the prominent developments in immunotherapy, such as immune checkpoint block (ICB) therapy for T cell activating and CAR-T therapy, there are still multiple factors that deteriorate the efficacy of lung cancer immunotherapy, such as the scarcity of CD8+ T cell infiltration in "cold" tumors, the so-called immune deserts[2,3], leading to the short median overall survival[4,5], which may be related to the complicated immunosuppressive microenvironment of lung tumor[6,7].

Innate immune cells play a vital role in the lung tumor immunosuppressive microenvironment[8]. For example, large numbers of macrophages and dendritic cells (DCs) infiltrating lung tumors could be domesticated in the tumor microenvironment (TME), which in contrast promotes tumor development[9]. Compared with other kinds of

cancer, diversified types of macrophages make the tumor immune microenvironment increasingly complicated, such as bone marrow-derived macrophages (BMDMs), alveolar macrophages (AMs), and interstitial macrophages (IMs), which promote tumor development by inducing angiogenesis, secreting various factors and inhibiting cytotoxic T cell responses by several means[10]. Besides, immature DCs can also suppress CD8+ T cells by producing transforming growth factor-β (TGF-β)[11] and expressing programmed cell death protein 1 (PD-1) ligand 1 (PD-L1)[12]. It is therefore considered that the intratumoral innate immune reversion should be the key in suppressing the lung cancer development prior to or during the currently prevailing T-cell-based therapy.

As abundant macrophages could infiltrate in lung TME, immunotherapies by activating the antitumor functions of macrophages are now recognized as promising way to treat lung cancer[13–16]. However, due to poor immunologic memory of macrophages, tumor immunotherapy based solely on macrophages may raise concerns about

[1]State Key Laboratory of High Performance Ceramics and Superfine Microstructures, Shanghai Institute of Ceramics, Chinese Academy of Sciences, Research Unit of Nanocatalytic Medicine in Specific Therapy for Serious Disease, Chinese Academy of Medical Sciences (2021RU012), 200050 Shanghai, P. R. China. [2]Center of Materials Science and Optoelectronics Engineering, University of Chinese Academy of Sciences, 100049 Beijing, P. R. China. [3]Shanghai Tenth People's Hospital, Shanghai Frontiers Science Center of Nanocatalytic Medicine, School of Medicine, Tongji University, 200331 Shanghai, P. R. China. ✉e-mail: huping@mail.sic.ac.cn; jlshi@mail.sic.ac.cn

cancer recurrence and metastasis. Therefore, combined macrophages/ T cells immunotherapy could integrate the advantages of the fast response of macrophages and the strong immunologic memory of T cells, which may achieve more efficient therapeutic outcomes. As a stimulatory signal, calreticulin (CRT) has been confirmed by extensive studies to be capable of both polarizing macrophages towards a pro-inflammatory M1 phenotype from anti-inflammatory M2 phenotype and activating the antigen-presenting DC[17], which further activates T cells through antigen presentation[18,19]. Such a CRT may be an efficient target for lung cancer immunotherapy, which, unfortunately, has been rarely reported in the orthotopic lung cancer therapy. Alternatively but also importantly, tumor cells of high CRT expressions in the plasma membrane may escape the macrophage clearance by overexpressing the "don't-eat-me" CD47 checkpoint signaling molecule on tumor cell surfaces[20,21]. As a result, both the CRT expression upregulation and the CD47 protein downregulation play key roles in strengthening macrophages polarization[22].

We herein present a combinational immunotherapeutic strategy against orthotopic lung tumor by establishing a sub-micro-sized layered double hydroxide (LDH)-based co-delivery system for dual small-molecular drugs, which activates the antitumor effect of tumor-associated macrophages (TAM), induces DC maturation, and activates T cell anti-tumor immune responses through the antigen presentation function of DC. As illustrated in Fig. 1, mild acidity-responsive and laterally micro-sized Zn-Al LDH is chosen as the drug carrier to load and deliver two kinds of small-molecule drugs, calcium channel inhibitor, TTA-Q6, and CD47 inhibitor, RRX-001. Compared with other nanocarriers, the submicron size and sheet shape make LDH more likely to locate in the tumor stroma rather than enter the cancer cells, enabling TTA-Q6 to functionalize because the calcium channel inhibitor can only enter the calcium channels from the extracellular space. Subsequently, these layered nanocarriers release these interlayer-loaded drugs in the acidic tumor microenvironment. This combinational immunotherapeutic strategy featuring tumor microenvironment-responsive drug releases could successfully circumstance systemic immune response and thus guarantee good biological safety of the LDH/RRX-001/TTA-Q6 co-delivery system, which is denoted as LRT nanomedicines after surface modification hereafter. Furthermore, as a specific T-type calcium channel inhibitor, TTA-Q6 released from the nanomedicine prevents the tumor cells from uptaking extracellular calcium ions, therefore inducing intracellular calcium deficiency and endoplasmic reticulum (ER) stress, ultimately resulting in the translocation of CRT expression from ER onto plasma membrane[23–28]. Besides, the released RRX-001, as a CD47 inhibitor currently under clinical trials, downregulates CD47 proteins on tumor cell membrane[29,30]. Subsequently, the combined CRT over-expression and CD47 low-expression on the tumor cell plasma membrane cooperatively activate the anti-tumor effect of TAMs and induce DCs to capture tumor-associated antigens and present them to T cells. The present study makes full use of the lung tumor immune microenvironment regulations to concurrently activate both the innate and adaptive immune responses in lung cancer therapy. In lung tumor models, this combined approach shows anti-tumor effects and immunity against tumor re-exposure, highlighting its potential for lung cancer immunotherapy.

## Results

### Synthesis and characterization of LRT nanomedicines

To build the LRT nanomedicine, Zn-Al LDH nanosheets were first synthesized using a homogeneous alkalization method, then modified with carboxyl PEG-trimethyl silane via hydrolysis reaction of silane coupling agents to improve the biocompatibility. The transmission electron microscopy (TEM) image of Zn-Al LDH-PEG-COOH (Fig. 2a), selected area electron diffraction (SAED) pattern (Fig. 2b), and

scanning electron micrograph (SEM) image (Supplementary Fig. 1) display that the as-synthesized hexagonal-crystallized LDHs is about 800–900 nm in the lateral dimension. The energy-dispersive X-ray spectroscopy (EDS) reveals the presence of the silicon element in the LDH after PEGylation compared to the LDH nanosheets before PEGylation (Fig. 2c and Supplementary Fig. 2), and the molar ratio among the main components Zn/Al/Si was further determined to be 1.98:1:0.13 by inductively coupled plasma optical emission spectrometry (ICP-OES). Furthermore, the Fourier transform infrared spectra (FTIR) in Fig. 2d demonstrate the successful modification of PEG. The characteristic absorption band at 1256 $cm^{-1}$ belongs to the stretching vibration of C−O in −COOH[31]. The new absorption at about 1108 $cm^{-1}$ is from the vibration of the Si−O− group formed in the hydrolysis of silanes. The existence of broad bands at around 2887 $cm^{-1}$ and 1360 $cm^{-1}$ in LDH nanoparticles are ascribed respectively to the stretching and flexural vibrations of −$CH_2$− in the carbon skeleton of PEG[32]. Besides, the zeta potential change (Fig. 2e) also demonstrates successful PEGylation.

LRT nanomedicine was then prepared by loading RRX-001 and TTA-Q6 in the interlayer of LDH. As revealed in TEM and SEM images (Fig. 2f and Supplementary Fig. 3), the dimension and morphology of LRT remain unchanged after the encapsulation of drugs. Corresponding element mappings in Fig. 2g display the uniform distributions of Si/Al/Zn/O elements. The average topographic height of LRT nanosheets was determined to be approximately 40 nm by atomic force microscopy (AFM) (Fig. 2h). The basal d-spacing variations in the X-ray diffraction (XRD) patterns in Fig. 2i depict that the basal d-spacing of the 003 plane becomes markedly increased due to the drug's intercalation into the interlayers, verifying the successful interlayer loading of drugs.

Next, the release profiles of RRX-001 and TTA-Q6 from LDH nanosheets at different pH values were investigated in vitro by measuring the UV spectra of the supernatant after nanocarrier removal, then the drug concentrations were obtained by UV spectra separation according to the standard curves (Fig. 2j and Supplementary Fig. 4). The loaded drugs are released fast within the first 6 h, then keep sustained release beyond, around 9.7 wt% RRX-001 and 6.4 wt% TTA-Q6 could be released from LDH carriers during 24 h under pH 6.5. The drug loading capacity is 18.2 wt% (7.2 wt% for TTA-Q6 and 11 wt% for RRX-001), while the encapsulation efficiency is 47.9 wt% (41 wt% for TTA-Q6 and 62 wt% for RRX-001), as calculated from UV spectra analyses of the drug solution before and after drug loading (Supplementary Fig. 5). Then, we investigated the degradability of LRT in phosphate-buffered saline (PBS) of varied pH values. The TEM images (Supplementary Fig. 6) indicate that LRT have significantly degraded for 72 h in acidic PBS (pH 6.5). By analyzing the concentration of $Zn^{2+}$ in the supernatant using ICP-OES, the degradation rate was calculated to be 57% in acidic PBS and 12% in a neutral PBS in 72 h (Fig. 2k). Such a pH-responsive degradation profile guarantees the sustained drug release from LRT in TME.

### LRT regulates the CD47 and CRT expressions of cancer cells

In order to regulate membrane proteins, the layered nanocarrier is expected to target the tumor stroma without being endocytosed by cancer cells, thus avoiding nanomedicine disintegration in lysosomes and prolong the effect of nanomedicine. First, the LRT phagocytic efficiency of Lewis lung carcinoma (LLC) cells was determined by flow cytometry. After co-incubation with the PI-labeled LRT, LLC cells were stained with DAPI to eliminate dead cells and avoid false-positive results from the cells stained with extracellularly released PI (Supplementary Fig. 7a). As depicted in Supplementary Fig. 7b, less than 10% of the LLC cells uptake LDH after co-incubation for 24 h, indicating that LLC can hardly uptake these layered materials most probably due to the well-designed submicron size and zeta potential of the LRT nanomedicine.

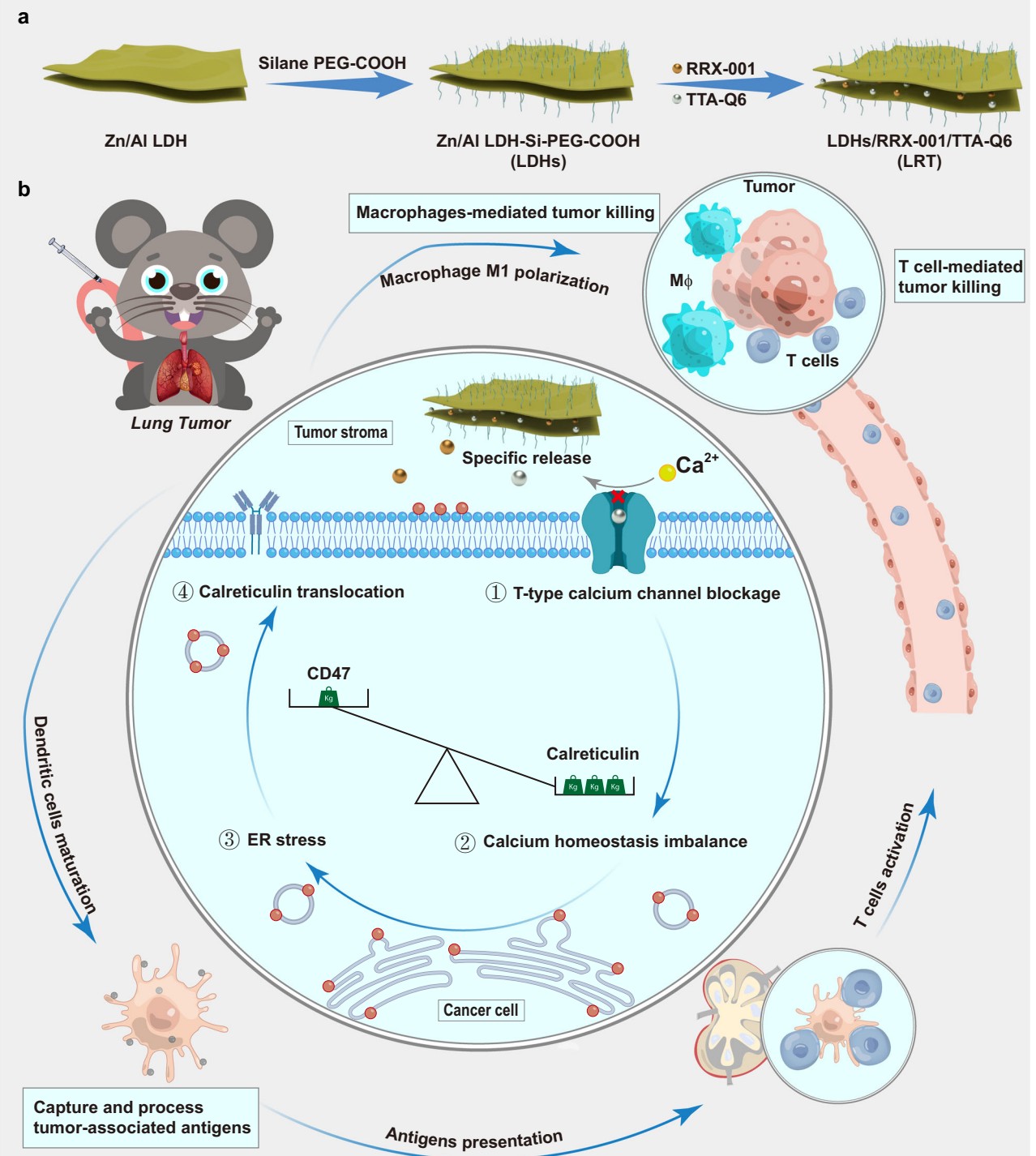

**Fig. 1 | Schematic illustration of synthetic procedures and therapeutic mechanisms of LRT nanomedicine. a** Schematic diagram of LDH nanosheet synthesis and the subsequent drug loading. **b** Mechanism of antitumor immune responses against orthotopic lung tumors by TTA-Q6/RRX-001 co-delivery. (1) T-type calcium influx channel blockage by TTA-Q6, (2) calcium homeostatic deficiency in parallel with the CD47 downregulation by RRX-001, (3) ER stress induction due to the calcium deficiency, (4) calreticulin translocation towards the cell membrane. MΦ represents macrophages.

Subsequently, we used permeable fluorescent calcium indicators Calbryte 520 and flow cytometry to analyze the inhibition effects of LRT on calcium influx. LDH, LDH/RRX-001 (LR), and LDH/TTA-Q6 (LT) were also set for comparison. As displayed in the cytometric results (Fig. 3a and Supplementary Fig. 8), the calcium fluorescence signal in the LLC cells treated with LT or LRT remain constant after calcium ion addition, indicating that the nanomedicine loaded with TTA-Q6 is capable of largely suppressing the cancer cell uptake of calcium from the extracellular environment. Besides, the intercellular calcium level of LLC cells was further investigated by confocal laser scanning microscopy (CLSM). According to Fig. 3b, calcium probe Fluo-4 fluorescent signals in LLC cells in 24 h of LRT treatment were substantially weaker than that in the control group, indicating the marked below-normal intracellular calcium levels, i.e., these LLC cells are under calcium homeostasis deficiency, which will lead to ER stress.

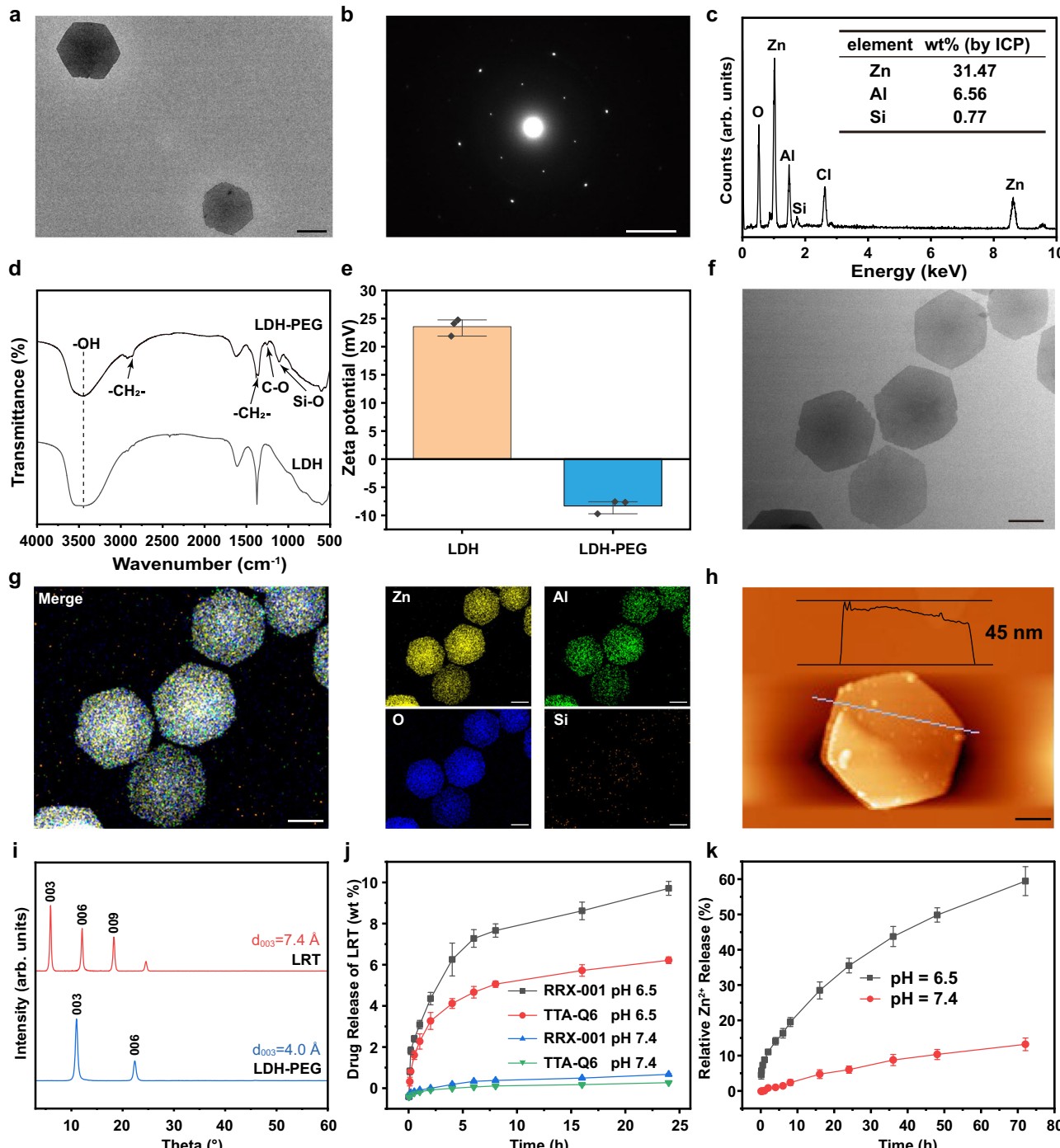

**Fig. 2 | Synthesis and characterization of LRT nanomedicines.** The representative TEM image of LDH nanosheets (**a**) and corresponding electron diffraction pattern (**b**). scale bars: 400 nm, 4/nm. $N = 3$ samples with similar results. **c** EDS profile of LDH sample after PEGylation and the elemental analyses by ICP-OES. $N = 3$ samples with similar results. **d** Representative Fourier-transform infrared spectra (FT-IR) of LDH before and after PEGylation. $N = 3$ samples with similar results. **e** Zeta potential variations in the preparation procedure of LDH nanosheets. The data were expressed as means ± SD ($N = 3$ independent experiments). The representative TEM image of LRT nanomedicines (**f**) and corresponding elemental mappings (**g**). scale bars: 400 nm. $N = 3$ samples with similar results. **h** AFM image of LRT nanosheets. Scale bar, 200 nm. **i** Powder XRD patterns of LDH nanosheets and LRT nanomedicines. **j** Release profiles of TTA-Q6 and RRX-001 from LRT in the phosphate buffer saline (pH 6.5 and 7.4). The data were expressed as means ± SD ($N = 3$ independent experiments). **k** Accumulated release profiles of Zn element from LRT in PBS of different pH values. The data were expressed as means ± SD ($N = 3$ independent experiments). Source data are provided as a Source Data file.

ER morphology change is a common indicator of ER stress, which was observed using transmission electron microscopy. According to the Bio-TEM images displayed in Fig. 3c, compared to the normal ER morphology of LLC cells in the control group, the ER of the LRT-treated group shows intensified edema, expansion, vacuolation of the cavity and degranulation, as noted by red arrows. In addition, ER stress is accompanied by protein misfolding in ER. Under the ER stress conditions, ER chaperone protein, BIP (binding immunoglobulin protein), binds to unfolded proteins and thereby renders ER stress transducers IRE1-α (inositol-requiring enzyme 1α) activated, and the ER stress marker CHOP (C/EBP homologous protein) upregulated[28]. According to the related protein expressions measured via western blot analysis

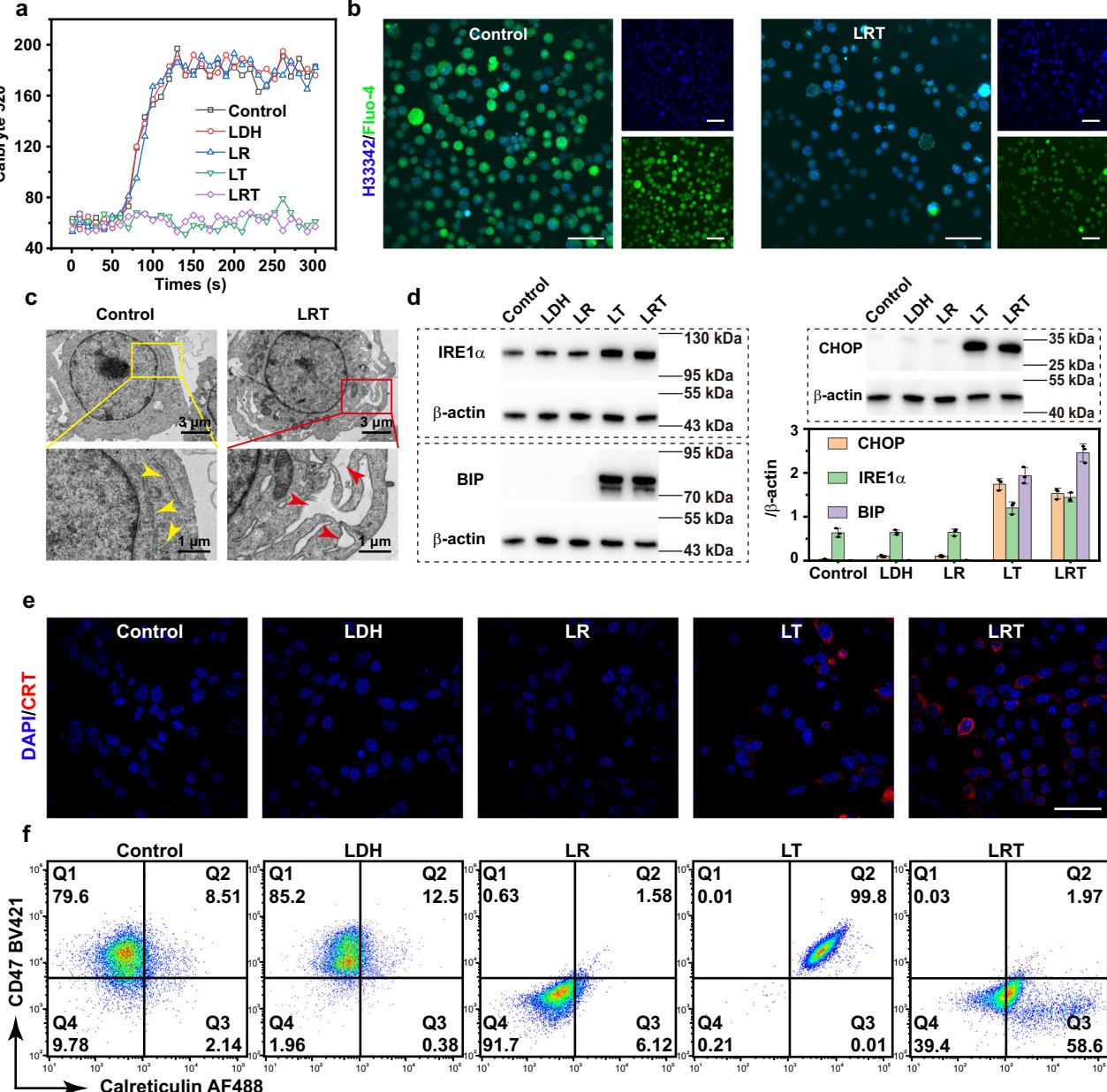

**Fig. 3 | In vitro CD47 and CRT regulations by LRT nanomedicine on the plasma membrane of LLC cells. a** Intracellular calcium flow analysis by flow cytometry of LLC cells after various treatments. $N = 3$ samples with similar results. **b** CLSM images of LLC cells after different treatments for 24 h. Fluo-4 (green) was used for cellular calcium observation and the nucleus were stained with Hoechst 33342 (blue). Scale bars, 50 μm. $N = 3$ samples with similar results. **c** Bio-TEM images of pristine LLC cells (control) and the LLC cells after the treatment with LRT nanomedicine for 16 h. Yellow arrows marks reveal the normal ER, while red ones indicate the ER under stress state. $N = 3$ samples with similar results. **d** Western blot analysis of ER stress-associated proteins of LLC cells after various treatments. The data were expressed as means ± SD ($N = 3$ samples). **e** CLSM images showing the CRT translocation of LLC cells treated with PBS, LDH, LR, LT or LRT. Blue and red fluorescence show DAPI-stained nucleus and CRT exposure on the cell surface, respectively. Scale bar = 50 μm. Cells were not permeabilized prior to immuno-fluorescence staining. $N = 3$ samples with similar results. **f** Flow cytometric analyses for CD47 and CRT regulations of LLC cells after various treatments, representative of 3 independent experiments. Source data are provided as a Source Data file.

(Fig. 3d), the BIP, CHOP and IRE1-α proteins are significantly upregulated in the LT/LRT-treated group, which implies the LT/LRT nanomedicines have induced ER stress in LLC cells. Subsequently, as a result of ER stress, the translocation of calreticulin to the cell surface as the "eat-me" signal was directly detected by CLSM in differently treated groups. As shown in Fig. 3e, significant red fluorescent signals can be observed in the LLC cells after the treatment with LT/LRT for 16 h, indicating that LRT nanomedicine successfully induces CRT exposure in the cell plasma membrane. Quantitative evaluations of CRT exposure as well as CD47 expression at the cell surface were made by flow cytometry (Fig. 3f and Supplementary Fig. 9). As expected, the CRT and CD47 expressions were respectively up- and downregulated in the LLC cells treated with LRT. These expression changes on cancer cell surface will activate the antitumor response of macrophages and stimulate DC maturation.

## In vitro activation of macrophages' antitumor response by LRT therapy

To assess the activation of macrophages' antitumor response by LRT therapy, the BMDMs were first isolated from bone marrow. As shown in

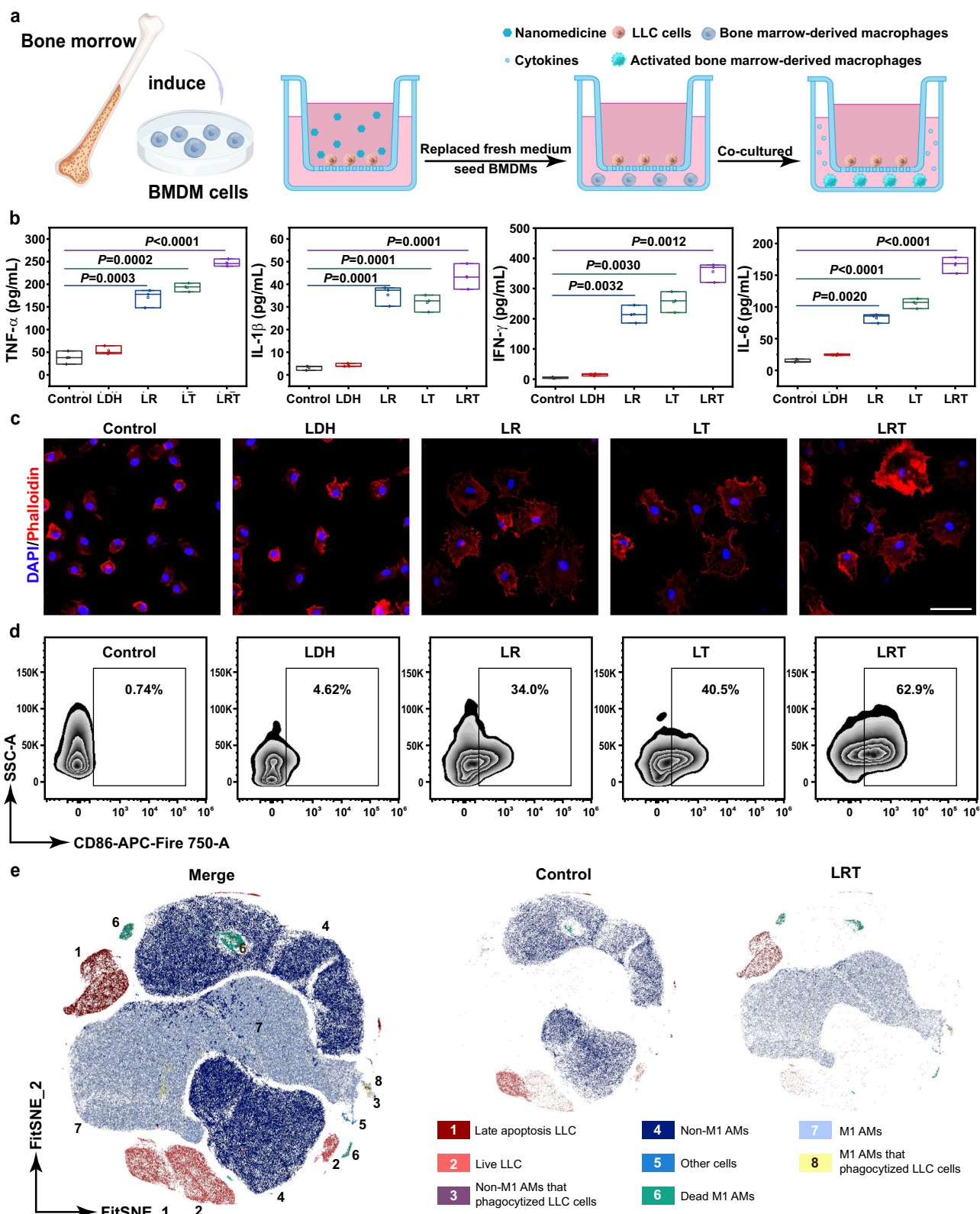

Supplementary Fig. 10, the isolated BMDMs have a purity of 97.3%. As illustrated in Fig. 4a, the LLC cells were seeded into the upper chamber of the transwell system followed by co-incubation with various nanomedicines, then the nanomedicines were removed and the BMDMs were seeded into the lower chamber for a 24 h co-incubation. Finally, the media in the transwell system were collected for cytokines detection, and the BMDMs in the lower chamber were fixed for morphology observation.

Cytokines are essential in the host defense against invading pathogens as the products of immune signaling activation. The secretion of cytokines and related immune response constitutes the fundamentals of host defense, and tumor necrosis factor-α (TNF-α),

**Fig. 4 | In vitro stimulation of immune response of macrophages incubated with LLC upon LRT nanomedicine treatment. a** Schematic illustration of BMDMs activation in a transwell system. LLC cells were placed in the upper chamber followed by various treatments for 16 h, then the media containing different nanoparticles were removed and BMDMs were cultured in the lower chamber. **b** Quantifications of the TNF-α, IL-1β, IL-6, and IFN-γ levels in the stimulated BMDMs suspensions. Data are expressed as means ± SD (*N* = 3 independent experiments). Boxplots show the distribution of expression with the center of the box representing the mean, the center line correspond to the median, upper and lower bounds representing 75% and 25% percentiles. All statistical significances were calculated via one-sided unpaired Student's *t* test. **c** CLSM images of BMDMs after being co-incubated with the LDH/LR/LT/LRT-treated LLC cells. BMDMs were stained with Tracker Red-594 phalloidin (red) and DAPI (blue) for cytoskeleton and nucleus imaging, respectively. Scale bar: 50 μm. *N* = 3 samples with similar results. **d** Flow cytometric analysis of BMDMs polarization following various treatments. **e** Fast interpolation-based t-SNE (Fit-SNE) representation of AMs and LLC cell landscape, representative of 3 independent experiments. The AMs pre-stimulated by LRT nanomedicine-treated LLC cells were co-cultured with CFSE-labeled LLC cells in V-bottom 96-well plate for AMs immunophenotype, tumor cell phagocytosis and killing effect flow cytometry analyses. Source data are provided as a Source Data file.

interleukin-1β (IL-1β), interferon-γ (IFN-γ), and interleukin-6 (IL-6) are the predominant cytokines of innate immunity in the protection against a majority of lung diseases[33]. it has been widely reported that M1 polarization of macrophages is essential in the antitumor immune response, and various kinds of cytokines secreted by macrophages play dominant roles in regulating intracellular communication and killing cancer cells[34]. Especially, IL-1β may induce IFN-γ production, while TNF-α and IFN-γ can kill tumor cells and inhibit tumor growth directly. To this end, ELISA assay was employed to detect various types of cytokines such as inflammatory IL-6, TNF-α, IFN-γ, and IL-1β, as the hallmarks of macrophage M1 polarization, in the supernatant as demonstrated in Fig. 4a. As shown in Fig. 4b, it is clear that the LRT nanoparticles-treated LLC cells could trigger the releases of all these kinds of cytokines by BMDMs, indicating the prevailing M1 polarization of BMDMs in the LRT group.

In addition to the secretion of cytokines, alterations in cell morphology provide direct evidence of the immunophenotyping of macrophages. Unlike the round and flattened morphology of M2-type macrophages, the M1-type ones display an elongated fusiform morphology for engulfing and digesting pathogens. After coculturing with the LLC cells treated by various nanomedicines in the transwell system, the BMDMs were fixed and stained with DAPI and Tracker Red-594-phalloidin to observe cytoskeleton and nucleus. It can be clearly observed from the CLSM images that the BMDMs have experienced marked morphological changes from the round one to spindle-like one, after the coincubation with LRT-treated LLC cells (Fig. 4c). This further demonstrates that LRT is capable of inducing extensive M1 polarization of BMDMs, which ensures the following immunoactivation-based lung tumor therapy. Besides, to further investigate the activation effect for BMDMs by LRT therapy, the activated BMDMs, as shown in Fig. 4a, were digested into single cells and stained with CD86 for polarization analyses by flow cytometry. As shown in Fig. 4d and Supplementary Fig. 11, increased amount of BMDMs stimulated by LRT-treated show CD86 positive signals, demonstrating that LRT therapy can activate the M1-type polarization of BMDMs. Additionally, the macrophage phagocytosis was also evaluated by flow cytometry. The stimulated BMDMs, as Fig. 4a, were co-incubated with the CFSE-labeled LLC cells in a V-bottom 96-well plate. The assessment of tumor cell phagocytosis by BMDMs was conducted in 24-h post-incubation. As illustrated in Supplementary Fig. 12, an enhanced amount of macrophages after stimulated by LRT-treated LLC cells exhibit CFSE signals compared to those in the control groups, indicating that LRT therapy have promoted the cancer cell phagocytosis of BMDMs.

In addition to BMDMs, there are abundant AMs in lung tumor tissues. As tissue-resident macrophages of lungs, AMs are the immune cells of the highest concentrations in healthy lungs responsible for the initiation or suppression of inflammatory responses, thus playing a crucial role in the development and growth of lung tumors[35–37]. Unlike the BMDMs, AMs originate from fetal monocytes and can self-maintain their numbers in vivo under steady-state conditions[38], performing different characteristics from BMDMs. Therefore, it is necessary to evaluate the activation of AMs by LRT therapy. In this study, we first isolated AMs by a well-established bronchoalveolar lavage approach,

and cell phenotype was further identified by flow cytometry. As shown in Supplementary Fig. 13, CD170+CD11c+ AMs cells have a purity of >95%, then these isolated AMs were cultured for proliferation for subsequent experiments.

To assess the immunophenotype, tumor cell phagocytosis and killing effects changes of AMs before and after LRT therapy, a multicolor flow cytometric experiment was designed to analyze the above parameters of AMs in parallel. First, LLC cells were pre-stained with cytoplasmic dye carboxyfluorescein succinimidyl ester (CFSE) to verify the phagocytic capacity of AMs, then AMs pre-activated by LRT-treated LLC cells were cocultured with the CFSE-labeled LLC cells in a V-bottom 96-well plate for 24 h. Next, the mixed cells were stained with CD170 and CD11c to label AMs, CD86 to analyze AMs polarization, and Annexin V and PI for apoptosis analysis of tumor cells. After quality control by FlowAi algorithms[39], the resulting high-dimensional dataset was visualized using the FitSNE algorithm[40], and meanwhile we used the unsupervised clustering and visualization algorithm FlowSOM[41] to group the cells (Supplementary Fig. 14a). The phenotype identity of each individual FlowSOM metacluster (MC) was defined by a heatmap showing the integrated median fluorescence intensity (MFI) values of each marker and visualization of the phenotypic variation across all cell subsets in FitSNE (Supplementary Fig. 14b–e). As revealed in Fig. 4e, the defined 8-cell subsets constitute dimensionality-reduction diagrams in different ratios. In detailed statistics shown in Supplementary Tables 2–4, AMs, which were pre-activated with LRT-treated LLC cells, were polarized predominantly to M1 phenotypes (96% of total AMs, MC 6, 7, 8), whereas the control group was not. Notably, a small fraction of AMs was dead after being polarized to M1 phenotypes (MC6, 3.5% of total AMs) because the activated AMs had lost their capability of proliferation to prevent excessive immune responses. Furthermore, LLC cells in the LRT group depicted 86% of late apoptotic cells (MC 1) and only 14% live cells (MC 2), indicating the potent tumor cell-killing effect of AMs in LRT group. Notably, slight LLC cell phagocytosis by AMs (MC 3 and 8, only 0.02% and 0.53% of total AMs) can be seen, demonstrating that AMs exert its antitumor function mainly through cell-killing rather than phagocytosis.

## In vitro immunity stimulation of dendritic cells by LRT therapy

While number of cell types have been reported to be capable of cross-presenting antigens, mature DCs are the most efficient among them in capturing, processing, and presenting antigens for lymphocyte activation in vivo. However, immature dendritic cells inhibit CD8+ T cells by producing TGF-β, expanding the Foxp3+ regulatory T (Treg) cells and expressing PD-L1. Therefore, inducing DCs maturation is critical for activating T cells' immune responses[11,12]. Since DCs located in the lungs and tumors are myeloid-derived, we first obtained DCs by inducing bone marrow in vitro and further purified them through flow sorting to remove CD11c+ macrophages. Then, as revealed in Supplementary Fig. 15, the purified bone marrow-derived dendritic cells (BMDCs) were obtained for subsequent experiments.

The secretion of cytokines is a reliable index for identifying DCs maturation because the matured DCs play a key immunomodulatory

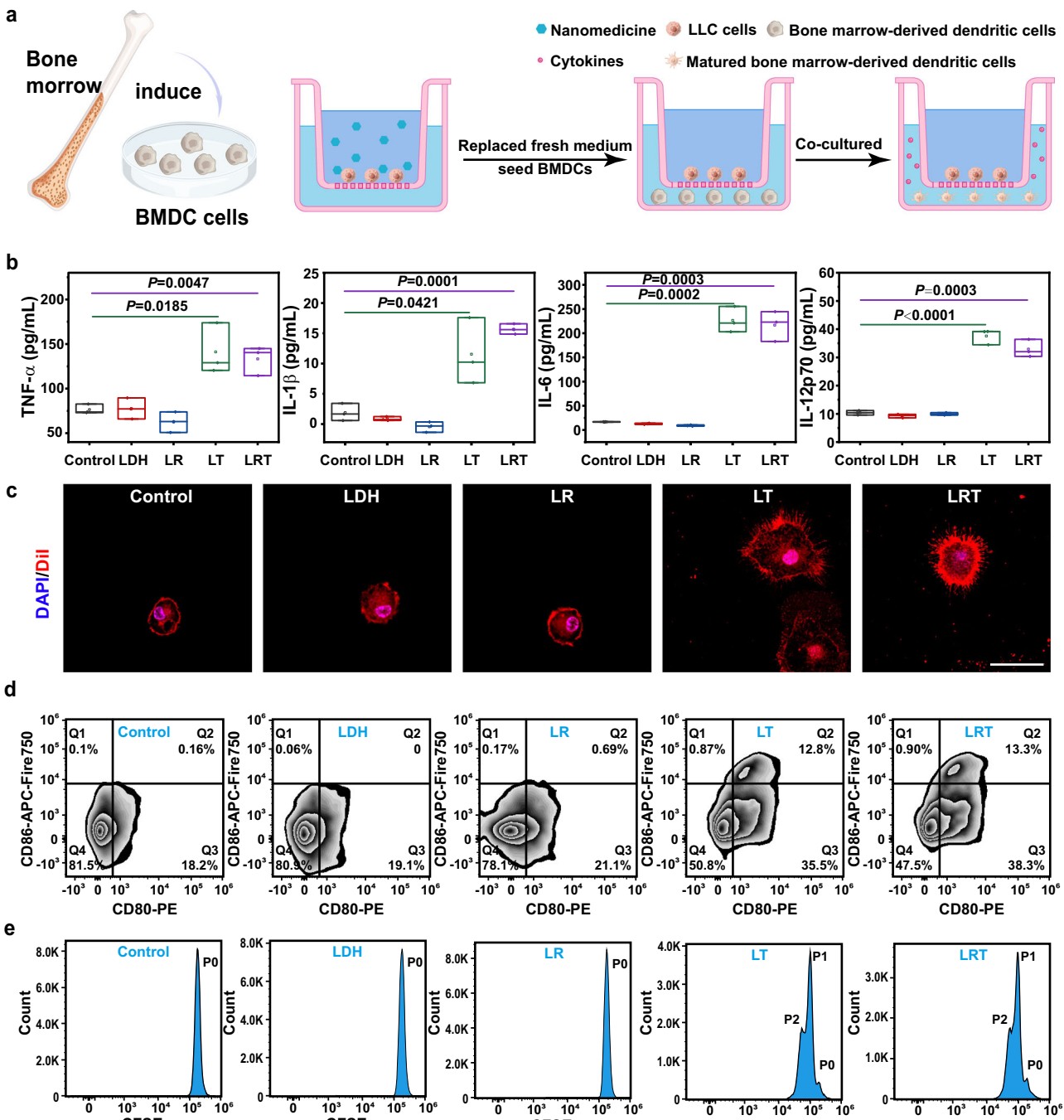

**Fig. 5 | In vitro BMDCs maturation induced by LLC upon LRT nanomedicine treatment. a** Schematic illustration of BMDCs stimulation in a transwell system. LLC cells were placed in the upper chamber followed by various treatments for 16 h, then the resultant media containing nanomedicines were removed and BMDCs were cultured in the lower chamber. **b** Quantifications of the secretion levels of TNF-α, IL-1β, IL-6, and IL-12p70 in the stimulated BMDCs suspensions. The data are expressed as means ± SD (N = 3 independent experiments). Boxplots show the distribution of expression with the center of the box representing the mean, the center line correspond to the median, upper and lower bounds representing 75%

and 25% percentiles. All statistical significances were calculated via one-sided unpaired Student's t test. **c** CLSM images of BMDCs after being co-incubated with LLC cells pre-treated with LDH/LR/LT/LRT. The cells were then stained with DiI (red) and DAPI (blue) for lightening plasma membrane and nucleus, respectively. Scale bar, 30 μm. N = 3 samples with similar results. **d** FACS plots of CD80 and CD86 expressions on BMDCs gated on CD11c⁺ cells, representative of 3 independent experiments. **e** Flow cytometric counts of CFSE-labeled T cells gated on CD3⁺ CD8⁺ cells for T cell proliferation analysis, representative of 3 independent experiments. Source data are provided as a Source Data file.

role in the tumor immunotherapy by secreting pro-inflammatory cytokines, such as IL12p70, a typical T helper type 1 (Th1) response promoter[42]. As revealed in Fig. 5a, BMDCs were cocultured with LLC cells after the treatment in the transwell system for 24 h. Then, the supernatant was collected for cytokine detection. From the Elisa results (Fig. 5b), BMDCs in LT and LRT groups secreted higher amounts

of IL-6, TNF-α, IL12p70, and IL-1β than in other groups, indicating the successful DCs maturation by upregulating CRT expression on cancer cell's surface.

As reflected by their name, the matured DCs grow branched projections. However, the immature DCs, also named veiled cells, possess abundant cytoplasmic "veils" rather than dendrites. Therefore,

surface protrusion is a crucial morphological sign of DC maturation. After the supernatant collection, BMDCs in the lower chamber were stained with plasma membrane dye DiI and nucleus dye DAPI for cell morphology observation. As displayed in CLSM image (Fig. 5c), BMDCs in the LT and LRT groups demonstrated a classic dendritic morphology. Comparatively, a round morphology corresponding to an immature state was observed in another group, further suggesting that LRT therapy could efficiently induce the DCs maturation. Such a morphology change of DCs enables their engulfing, processing and surface presenting for antigens. Therefore, the antigen phagocytosis and processing by DCs were then revealed by CLSM imaging (Supplementary Fig. 16). LRT-treated LLC cells were rapidly engulfed by DCs, which benefited from the easy recognition of CRT on the LLC cell surface by the matured DCs. After phagocytosis, the antigens in cancer cells will be processed and further presented to T cells, thereby facilitating T cell activation and proliferation.

The maturation and antigen presentation function of DCs were further confirmed through flow cytometric phenotypic and T cell proliferation analysis. As illustrated in Supplementary Fig. 17a, BMDCs were first co-incubated with LRT treated-LLC-OVA cells for 24 h, then co-cultured with CFSE-labeled CD8$^+$ T cells isolated from OT-I mouse spleen through flow cytometric sorting (Supplementary Fig. 17b, c). Then the mixed cells were collected and analyzed by flow cytometry (Supplementary Fig. 17d). Compared with the control group, BMDCs in the LRT group exhibited a higher upregulation in co-stimulatory molecules (CD80 and CD86) than in other groups (Fig. 5d), further suggesting the maturation of BMDCs in the group. Furthermore, using an in vitro OT-I CD8$^+$ T-cell priming assay, three distinct fluorescence intensity peaks respectively belonging to primary, and the first/second-filial generation in the LRT group indicate a significant proliferation of OT-I CD8$^+$ T cells. In contrast, only a single fluorescence intensity peak belonging to primary cells can be seen in the control group (Fig. 5e). These data suggest that LRT therapy could effectively activate BMDCs and trigger subsequent T-cell responses.

## In vivo orthotopic lung cancer immunotherapy by LRT treatment

Encouraged by the favorable performance of LRT-based combination therapy in vitro, we investigated its therapeutic efficacy in animal orthotopic lung tumor models. First, a orthotopic LLC lung cancer model was established by injecting LLC-Luc cell-containing Matrigel into the left lung of male C57BL/6J mice (six-week-old). Then, the distribution and tumor accumulation of LRT in LLC tumor-bearing mice were evaluated by tracking the fluorescence of indocyanine green (ICG)-labeled LRT using an IVIS spectrum imaging system in 8 days LLC post-injection. As presented in Fig. 6a, Supplementary Fig. 18a, b, the ICG fluorescence signal shows an extensive overlap with Luciferin signal in the tumor region, which peaks in 1 h post-injection, revealing a 9.4% ID•g$^{-1}$ of tumor accumulation by measuring the amount of Al element in main organs and tumor tissues using ICP-OES (Supplementary Fig. 18c). Compared with the rather low tumor accumulation of most of the reported nanomedicines (mostly below than 5% ID•g$^{-1}$)[43,44], the enrichment rate of LRT in the orthotopic lung tumors has been substantially elevated. Besides, the blood circulation half-life ($T_{1/2}$) of LRT was calculated to be approximately 27 min (Supplementary Fig. 19) according to a two-compartment model. The short half-life of blood circulation is due to the fact that large-sized nanomedicines are easily cleared by the liver and spleen, but at the same time, they are also more likely to accumulate in tumor tissues. This is one of the reasons why we chose to use such large-sized nanocarriers.

Next, the efficacy of LRT therapy was evaluated. In order to detect lung invasion of tumor cells more accurately, micro-CT were used to determine the development of orthotopic lung tumor upon the model establishment. As illustrated in Supplementary Fig. 20, the HE staining image shows a solid tumor in the left lung lobe in eight days LLC post-

injection, suggesting that the mice in situ lung cancer model was established successfully. Then, the C57BL/6J mice bearing tumors as confirmed by micro-CT imaging, were randomly divided into five groups ($N = 6$), including the control (PBS), LDH, LR, LT, and LRT groups. Then, the mice were intravenously injected with PBS/LDH/LR/LT/LRT according to groups on day 0/2/4, followed by micro-CT detecting once per week until a humane endpoint or the trial endpoint was reached, as shown in the scheme of animal treatments (Fig. 6b).

To better observe the lung tumor progression of mice in every group, lung three-dimensional reconstruction and corresponding CT transverse section images are depicted in Fig. 6c. Notably, the large low-density area of darker contrast in the transverse section images represents a healthy lung without invasion by tumor cells, while a high-density area of brighter contrast indicates the tumor-invaded lung. The 3D reconstruction model displays the overall morphology of lungs with or without the defect site which represents the tumor, as the tumor 3D morphology was difficult to reconstruct because the tumor border outside the lung matrix was almost non-definable due to the contrast absence. As illustrated in Fig. 6d, tumor progressions were characterized by determining the volumes of healthy lung matrix from the 3D reconstruction. All mice in the control group reached their humane endpoints after the treatment with PBS for 12 d, demonstrating the high growth rate of the orthotopic lung cancer and the resultant high degree of invasiveness. LDH inhibited tumor growth slightly compared to the control group, and the median survival extended to 14 d (Fig. 6e), which may benefit from the released high concentrations of $Zn^{2+}$ and $Al^{3+}$. In contrast, the volume changes of the healthy lungs in LR and LT groups imply that the tumor growths were first suppressed but then recovered in the time course of treatment and observation, leading to the diminished lung matrix volumes due to the tumor growths in the later stages. There are significant individual differences in the treatment effects of LR groups. Besides, relatively large-sized recurrent metastases in the right lung and small-sized recurrent metastases in the left lung can be observed as a special case in the LR group in the 3D reconstruction image, which is believed to be due to the lack of T cells response. LT groups exhibited slower tumor progression and longer median survival than the LR groups, which benefit from the activation of macrophages and T cells. However, the lack of CD47 regulation deteriorated the long-term anti-tumor efficacy of macrophages. To our delight, LRT showed a rapid and durable therapeutic effect, and 83% of mice remained alive after the treatment for as long as three months, confirming the excellent efficacy of the combined CD47 and CRT regulations.

In addition, the therapeutic efficacies of LRT and the combined administration of free TTA-Q6 + RRX-001 drugs were also evaluated to investigate the role of the LDH nanocarrier. Following the same schedule in Fig. 6b, the mice bearing orthotopic lung tumor were intravenously injected with PBS/LDH/TTA-Q6 + RRX-001/LDH + TTA-Q6 + RRX-001/LRT on day 0/2/4, followed by micro-CT detecting. As shown in Supplementary Fig. 21, the LRT performed a much higher therapeutic efficacy than administrating free drugs of LDH + TTA-Q6 + RRX-001, indicating that LRT features an elevated intra-tumoral enrichment rate compared to free small molecule drugs. In addition, the weights of the mice in LDH + TTA-Q6 + RRX-001 groups decreased much more significantly than in LRT groups, demonstrating good biosafety of LRT than simply administrating TTA-Q6 + RRX-001 owing to its favored pH responsiveness of drug release from LRT.

Furthermore, to show the contributions of AMs and DC-primed T cells in the anti-tumor effects of LRT in vivo, the immunotherapeutic outcomes of LRT on both wild-type and Rag-deficient mice were also assessed. As illustrated in Supplementary Fig. 21g, the Rag-deficient mice show a faster tumor progression and a shorter survival period than that of the wild-type mice, demonstrating that T cells are indispensable for durable tumor immunosuppression, as the innate macrophages possess no immune memory effect.

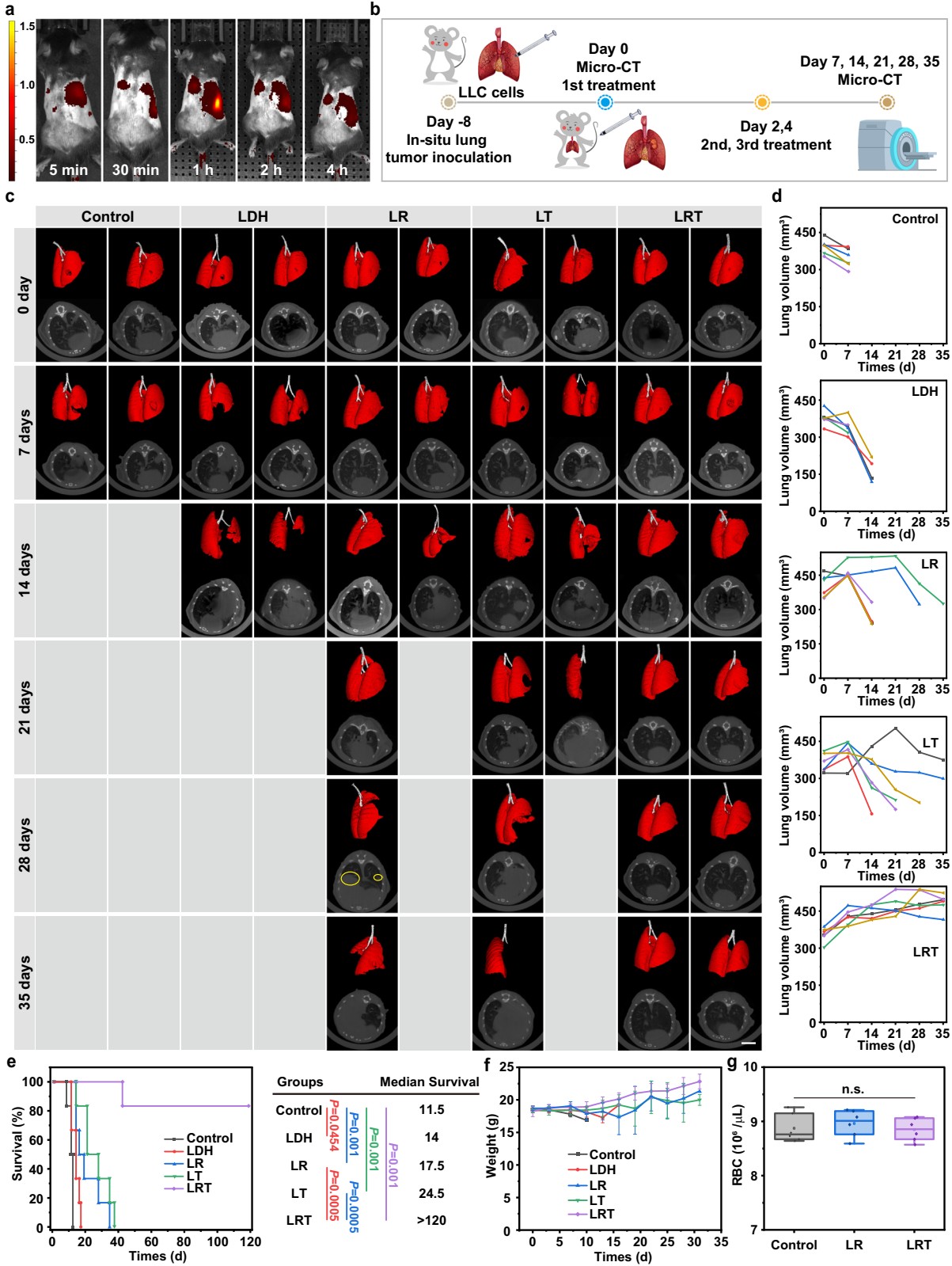

The weight of mice in the LRT group kept increasing after the treatment, showing excellent biosafety (Fig. 6f). Most importantly, the immunotherapy based on CD47 down-regulation may raise a concern about anemia because red blood cells also evade the macrophage phagocytosis response by high-expressed CD47, and the severe anemia caused by CD47 ICB therapy become a crucial issue for clinical usage in patients. Due to the targeting and specific-responsive

releasing of LRT, the red blood cell count kept at normal values after three dose administrations of LRT (Fig. 6g). A comprehensive study of the in vivo biocompatibility of LRT for short and long time periods was conducted by intravenously injecting the nanomedicine into healthy mice. First, the levels of Al elements in mouse feces and urine were quantified using ICP-OES to investigate LRT metabolism kinetics. The results, as depicted in Supplementary Fig. 22, indicate that

**Fig. 6 | In vivo immunotherapeutic outcomes by LRT nanomedicine treatments for orthotopic Lewis lung tumor model on C57BL/6 J mice. a** In vivo biodistribution evaluation of LRT nanomedicines. Color bar: radiant efficiency [$10^8$ p/sec/cm$^2$/sr/μW/cm$^2$]. Representative images of 3 animals with similar results. **b** Schematic illustration of the experimental procedure to assess the in vivo growth inhibition effects on tumors. **c** Representative lung 3-dimensional reconstruction and corresponding CT transverse section images from each group (6 animals) after varied treatments as indicated. Scale bar: 5 mm. **d** The healthy lung volumes of six animals per group quantized by computational extraction through 3D reconstruction. The Kaplan-Meier survival curves and median survival durations (**e**), and body weight fluctuation (**f**) in between of LLC tumor-bearing mice after varied

therapy. The data were expressed as means ± SD ($N$ = 6 animals), and p values were calculated using the log-rank test. **g** Red blood cell (RBC) counts of LLC tumor-bearing mice after three dose administrations of LRT nanomedicines, $N$ = 6 animals. Approximately 10 microliters of blood were collected from the tail-vein on day 5 for the counting. Statistical significances were calculated via unpaired one-sided Student's t test. Boxplots show the distribution of expression with the center of the box representing the mean, the center line corresponds to the median, upper and lower bounds representing 75% and 25% percentiles, and upper and lower whiskers extending to the largest value no further than 1.5 times interquartile range from bounds of box. Source data are provided as a Source Data file.

approximately 70% of LRT have been metabolized and excreted within a 7-day period through urine and feces, suggesting that most LRT can be efficiently cleared up from the body within a relatively short timeframe. In addition, to ascertain whether the accumulation of LRT in the liver and spleen will induce inflammatory cytokine responses in mice or not, LRT was administrated to healthy tumor-free mice via intravenous tail injections for three times every other day. Subsequently, serum samples were collected and concentrations of TNF-α and IL-12p70 were quantified through ELISA assay. The main organs were collected for H&E staining analyses. As demonstrated in Supplementary Fig. 23, no significant elevations in cytokine levels and no notable inflammatory responses in the organs can be observed among the tumor-free mice, thereby indicating that LRT will trigger no immune activation in healthy mice without tumors. Furthermore, a four-week period of observation revealed that the body weights of all mice increased gradually, with no discernible differences have been observed between groups (Supplementary Fig. 24a). Besides, hematological parameters (Supplementary Fig. 24b) and H&E staining of main organs (Supplementary Fig. 24c) harvested from mice in all groups exhibited no significant abnormity on day 28, indicating the satisfactory histocompatibility of LRT nanomedicine.

**Mechanism study of LRT treatment**

In addition to the activation of the immune system, the therapeutic effect of LRT against lung tumor may also be attributed to the direct killing effect of LRT on cancer cells. To assess the cytotoxicity of nanomedicine on tumor cells, we conducted a TUNEL assay to stain the tumors sections of mice after LDH or LRT treatment. As depicted in Supplementary Fig. 25, the negligible TUNEL positive signals of the tumor sections after LDH treatment suggest that zinc ions released from LDH are impotent to induce immunogenic cell death in cancer cells. Following a single LRT treatment, only a slight amount of tumor cells display TUNEL signals. Even after the LRT treatment for three times, the proportion of TUNEL-positive tumor cells remained at a low level. Therefore, the activation of the anti-tumor immune response, rather than the direct cytotoxic impact of the nanomedicine on tumor cells, is the main mechanism of cancer cell killing and tumor regression.

To investigate the underlying antitumoral immune mechanism by LRT treatment, immune cells and microvascular development in the LLC tumors were studied on day 0 before LRT therapy, and day 8 after tumor inoculation. As shown in the CD45/CD31 multiplex fluorescence immunohistochemistry image (Fig. 7a), abundant microvascularity and immune cell infiltration have been present in the untreated tumors, providing a rational basis for the enrichment of LRT nanomedicine in tumor tissue and cancer immunotherapy. Considering that calreticulin as an ER-resident chaperones is abundantly expressed in the ER of normal cells, CHOP, a typical ER-stress marker, was selected to track the calreticulin translocation from ER to plasma membrane. In one day of LRT treatment, the CHOP expression level was upregulated (Fig. 7b), demonstrating that LRT has successfully induced ER stress in cancer cells. Besides, CD47 was downregulated in LRT groups compared with LDH groups, and the combined CD47/

CHOP changes will lead to the activation of the immune system against cancer cells.

Subsequently, immune cells in LLC tumors were assessed after different treatments and immunophenotyping of leukocyte subpopulations was performed using a twenty-one-color flow cytometry panel. As shown in Fig. 7c–e and Supplementary Figs. 26, 23 different subpopulations of the CD45$^+$ cells among six groups are defined by x-shift algorithm. Compared with LDH groups, LR treatment effectively increased the populations of TAMs-M1 and reduced those of TAMs-M2, while the tumor-infiltrating matured DCs and CD8$^+$ T cells were not affected significantly, indicating that LR treatment has inhibited tumor growth mainly through the M1 polarization of the macrophages rather than the activation of CD8$^+$ T cells. The matured DCs and CD8$^+$ T cells infiltrated in tumors after LT treatment proliferated significantly compared with the LDH group, while the populations of TAMs-M2 and TAMs-M1 are respectively higher and lower than those in the LR group, though TAMs-M1 still proliferated compared with LDH groups. In addition, although the CD8$^+$ T cell population in LT group increased compared to LDH group, there are no significant differences in memory CD8$^+$ T cells (Ly6C$^{high}$CD8$^+$ T cells)[45], Th1-like memory CD4$^+$ T cells (Ly6C$^{high}$CD4$^+$ T cells) and Tfh-like memory CD4$^+$ T cells (Ly6C$^{low}$CD4$^+$ T cells)[46] between LDH and LT groups, which is essential for preventing tumor relapse or metastasis. The specific antitumor immune response modulation results from the high expression of CD47 in LT groups which prevents macrophage from exaggerated M1 polarization (Fig. 7b), then the increased TAM-M2 population inhibits the downstream immune responses. The above data explain the unsatisfactory treatment efficacy on lung cancer in both LR and LT groups. Out of all groups, the most intensive tumor-infiltrating CD8$^+$ T cells, Th1-like memory CD4$^+$ T cells as well as Tfh-like memory CD4$^+$ T in the LRT-8 days group have been found (Fig. 7e), resulting in much enhanced overall immune cell percentage (Fig. 7d), benefiting long-lasting therapeutic effects and immune memory protection. By profiling the tumor's immune microenvironment in 2 days post-LRT treatment, we observed a significant reduction in TAM-M2 populations compared to the control group, and the concurrent substantial population increases of TAM-M1 and matured DC. These findings suggest that the LRT nanomedicine can rapidly activate the antitumor functions of macrophages and DC within a short timeframe. Such a combined innate and adaptive immune response activation demonstrates potent and durable antitumor effects, explaining the best therapeutic efficacy of LRT treatment as can be observed in Fig. 6c. In addition to tumor immune microenvironment, cytokine secretion is also a typical indicator of immune responses. As shown in Supplementary Fig. 27, the TNF-α and IL-12p70 levels in the serum of LRT-treated mice are largely upregulated as the typical markers of antitumor immune response, while TGF-β indicative of an immunosuppressive environment is much downregulated, also evidencing the activated antitumor immune response in the LRT therapy.

Recently, the spatial distribution of immune cells and tumor cells has emerging as a novel biomarker for predicting immunotherapeutic responses. To further explore the mechanism of LRT therapy at the spatial level, the fluorescent multiplex

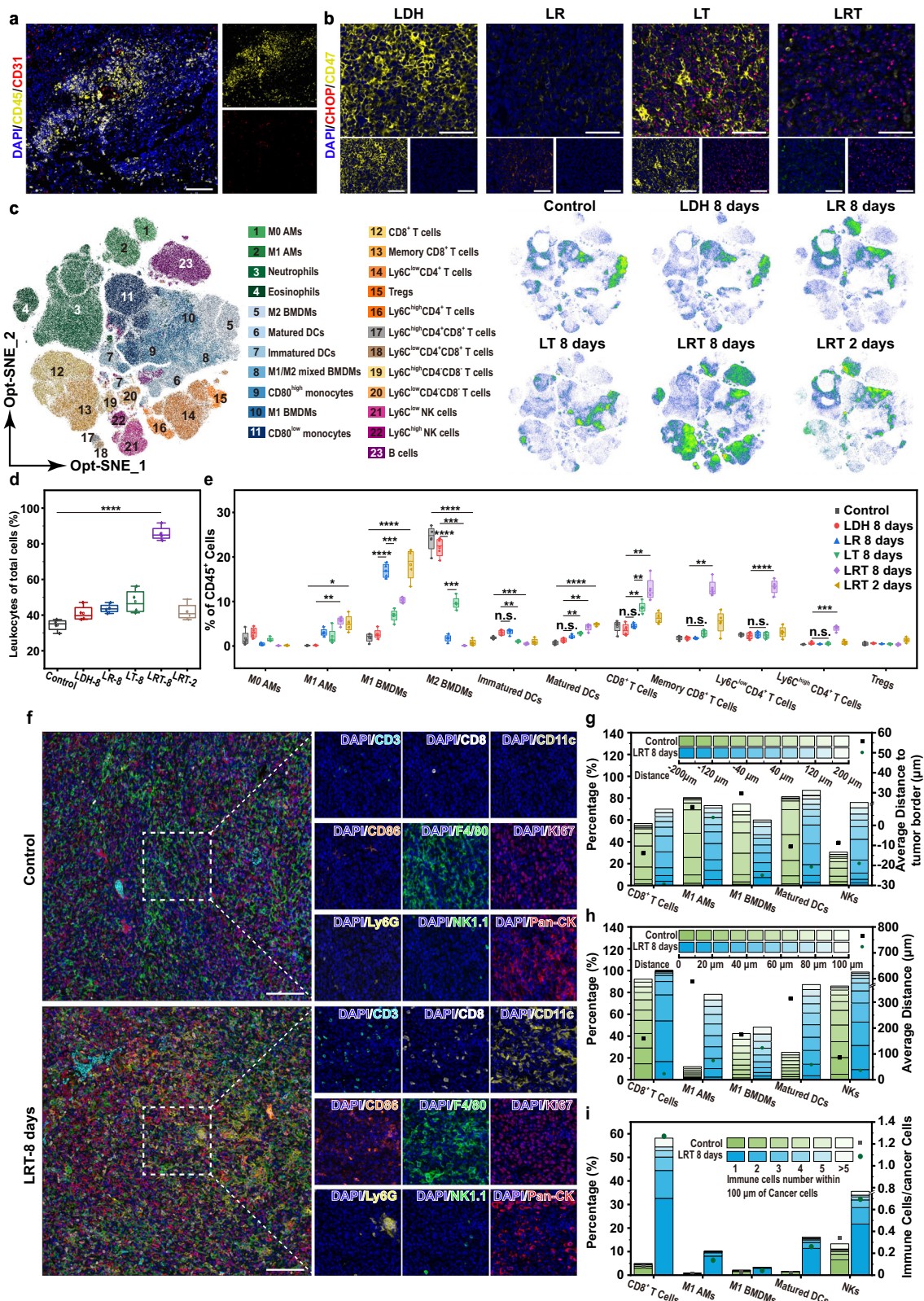

immunohistochemical analysis was performed on the tumor of the mice in the control and LRT 8-day groups. As shown in Fig. 7f, different cell types can be defined by differently co-expressed markers, such as CD8+ T cells (CD3+CD8+), M1 AMs (F4/80+CD11c+CD86+), M1 BMDMs (F4/80+CD11c-CD86+), matured DC (F4/80-CD11c+CD86+), NK cells (NK1.1+CD3-) as well as cancer cells (Pan-CK+Ki67+), then we used HALO platform to analyze the

locations of the defined immune cells from tumor border, the shortest distance between cancer cells and immune cells, as well as the number of immune cells surrounding cancer cells. From the infiltration results (Fig. 7g and Supplementary Fig. 28), the immune cells in LRT 8 days groups distributed more prominently within the tumor tissue, presenting a larger average infiltration depth than that of control groups benefiting the immune activation in TME, which is

**Fig. 7 | Mechanism study of antitumor immunotherapy by LRT nanomedicine. a** Polychromatic immunofluorescent staining images of tumor sections from the mice without any therapy showing DAPI (blue), CD45⁺ (yellow) and CD31⁺ (red) cells. Scale bars: 100 μm. Representative of 4 animals with similar results. **b** Representative polychromatic immunofluorescent staining images of tumor sections showing DAPI (blue), CHOP⁺ (red) and CD47⁺ (yellow) cells in different groups after varied treatments as indicated for 1 days. Scale bars: 50 μm. Representative of 4 animals with similar results. **c** High-dimensional data analysis on tumor-infiltrating leukocytes from different groups displaying x-shift clusters projected on two Opt-SNE dimensions. The overlay plot shows concatenated events from all 24 samples, while the density plots show the differential population distributions among representative individual samples ($N = 4$ animals). **d** Proportions of CD45⁺ leukocytes in the whole tumors of different groups. Box-plots show the distribution of expression with the center of the box representing the mean, the center line corresponds to the median, upper and lower bounds representing 75% and 25% percentiles, and upper and lower whiskers extending to the largest value no further than 1.5 times interquartile range from bounds of box. $N = 4$ animals. **e** Quantifications of various kinds of immune cells in x-shift clusters in percentages over total CD45⁺ cells after diverse treatments. Boxplots show the distribution of expression with the center of the box representing the mean, the center line corresponds to the median, upper and lower bounds representing 75% and 25% percentiles, and upper and lower whiskers extending to the largest value

no further than 1.5 times interquartile range from bounds of box. $N = 4$ animals. **f** Representative mIHC images from the lung tumor sections of the mice with/without LRT treatment for 8 days. Scale bars: 100 μm. Representative of 4 animals with similar results. **g** Immune cell infiltration analysis across tumor border. The columnar stacking diagrams towards the left $y$-axis show the distribution of immune cells at different locations of infiltration from tumor border. The scatter-plots towards the right $y$-axis represents the average infiltration depth. Representative of 4 animals with similar results. **h** The nearest neighbor analysis for each cancer cell by mIHC staining. The columnar stacking diagrams towards the left $y$-axis show the spatial distribution of immune cells of the shortest distance from cancer cells. The scatterplots towards the right $y$-axis represent the average shortest distance between cancer cells and immune cells. Representative of 4 animals with similar results. **i** Proximity analysis of immune cells towards each cancer cell by mIHC staining. The columnar stacking diagrams towards the left $y$-axis show the number and corresponding proportion of immune cells in the vicinity of cancer cells (within 100 microns away from that cancer cell center). The scatterplots towards the right $y$-axis represent the average numbers of the immune cells in the vicinity of cancer cells. Representative of 4 animals with similar results. *$p < 0.05$, **$p < 0.01$, ***$p < 0.001$, ****$p < 0.0001$, n.s., not significant. Statistical significance is calculated using unpaired one-sided Student's $t$ test. Source data are provided as a Source Data file.

particularly important for solid tumor immunotherapy. In contrast, although there are small amounts of M1-type macrophages, matured DC as well as CD8⁺ T cells, their migration mobilities are in turn reduced in the immunosuppressive microenvironment. These could explain, at least in part, the unsatisfactory clinical results of adaptive immunotherapy for solid tumors. Subsequently, the shortest distances between cancer cell and immune cell are shown in Fig. 7h and Supplementary Fig. 29, the most adjacent CD8⁺ T cells, M1 AMs, matured DCs and NK cells of cancer cells in LRT 8 days groups are mostly distributed within 100 microns of diameter away from the cancer cells, presenting the shortest average distance. In particular, more than 70% of cancer cells in control groups are located longer than 100 microns away from their most adjacent M1 AMs as well as matured DCs, this distance is too large to induce effective damage to cancer cells. In order to evaluate tumor treatment efficacy and prognosis more intuitively, proximity analysis was performed to count immune cells in the vicinity of cancer cells (Supplementary Fig. 30), and the number of immune cells within 100 micron away from the center of that cancer cell were recorded to evaluate the immune response (Fig. 7i). More than 50% of cancer cells in LRT 8 days group were surrounded by at least one immune cell(s) in the vicinity, showing the much higher immune cell/cancer cell ratios than the control group, further explaining the enhanced therapeutic effect of LRT. Of note, NK cells in LRT 8 days groups are located closer to cancer cells and in higher percentages than other immune cells except CD8⁺ T cells, probably due to the regulatory effect of low-expressed CD47 on NK cells[47].

### Long-term immune memory effects

From Fig. 7e, abundant memory T cells can be found in tumor microenvironment after LRT therapy, which indicates that the immune system may have established the long-term antitumor immune memory. To evaluate the immune memory effects generated by LRT treatment, mice that survived on day 90 after the first LRT treatment were rechallenged with LLC cells (Fig. 8a), with age- and sex-matched naive mice being injected with the same number of LLC cells as controls. As shown in the followed-up lung CT scans (Fig. 8b), obvious LLC cellular infiltration in lung tissue can be observed in the naive mice after the LLC cells injection. Compared with the control group, half of the mice that survived after LRT therapy did not show evidence of tumor formation over time, while the other half were observed to bear small tumor or inflammatory nodules in lung tissues, which, however, were later found to be spontaneously regressed over time. All

rechallenged mice survived until the end of the experiment (Fig. 8c), demonstrating the long-term anti-tumor immunological memory by the LRT therapy which successfully protected mice from tumor recurrence. In addition, weights of the mice in the LRT group decreased significantly earlier than controls (Fig. 8d), this could be due to side effects of the anti-tumor immune response.

To verify the underlying mechanism of the immunological memory responses, peripheral-blood lymphocytes were collected 7 days after the rechallenge from the retro-orbital venous plexus and evaluated the IFN-γ production by CTLs with ELISpot assay. Typically, when exposed to the same but secondary antigen, immune system will be activated spontaneously and rapidly eliminate any cells that express this antigen on their surface by IFN-γ-secreting T cells. As shown in Fig. 8e, a large number of spots formed in the LRT groups, indicating T cells were activated after LLC rechallenge, establishing a strong antitumor immune response by the primary LRT therapy. Memory T cells as primary lymphocytes during the secondary immune response were also analyzed by flow cytometry after being challenged with LLC cells for 31 days (the spleens of the control group were collected at their humane end points). Compared to the control group, the percentages of memory T cells in the spleen, including central memory ($T_{CM}$, CD62L⁺CD44⁺) and effector memory T cells ($T_{EM}$, CD62L⁻CD44⁺), increased significantly in the LLC cell rechallenged group (Supplementary Fig. 31, Fig. 8f, g), demonstrating that LRT therapy has effectively established antitumor immune memory.

## Discussion

Large numbers of immunosuppressive macrophages and dendric cells are present in lung tumor microenvironment, which play a role in suppressing the antitumor effects of T cell therein resulting in the disappointing efficacy of immunotherapy. Herein, we report a sub-micron-sized and layer-structured pH-responsive nanomedicine LRT, in which a T-type calcium channel inhibitor, TTA-Q6, and a CD47 small molecule inhibitor, RRX-001, were loaded in between the interlayer, for macrophage and dendritic activation, finally reversing immunosuppression in lung tumors (Fig. 1, Supplementary Figs. 32, 33). LRT treatment in vitro has been demonstrated to inhibit calcium influx and subsequently induce ER stress, resulting in the calreticulin upregulation and CD47 protein down-regulation in cancer cells. The LRT-treated cancer cells successfully induced M1-type polarization of macrophages and DC maturation, and finally activated anti-tumor immune response of T cells. The

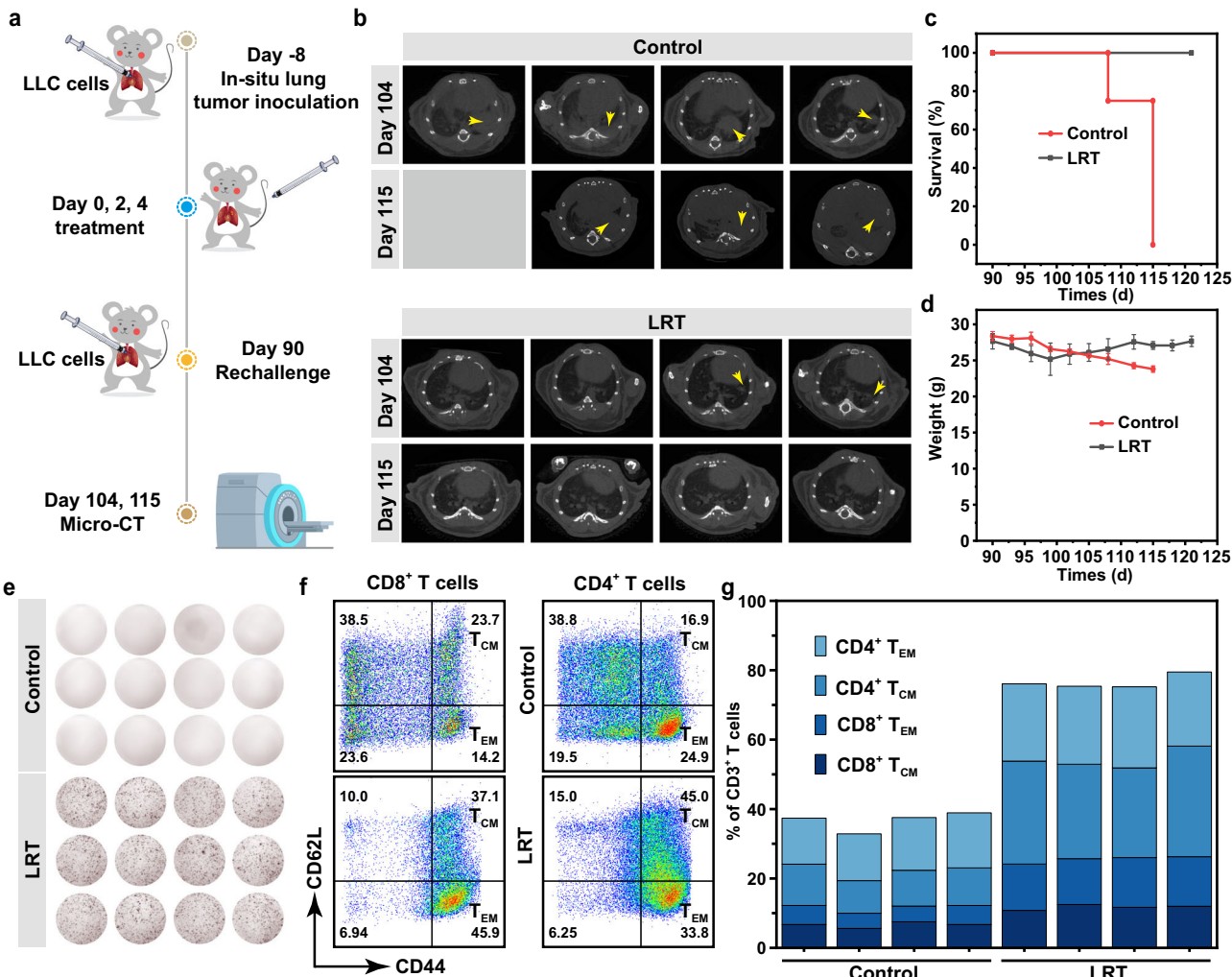

**Fig. 8 | Long-term immune memory effects. a** Schematic illustration of the experiment design to assess the immunological memory response triggered by LRT therapy. **b** Representative CT transverse section images of four biologically independent animals from each group after varied treatments as indicated. Kaplan−Meier survival curves (**c**) and body weight fluctuation curves (**d**) of mice. The data were expressed as means ± SD ($N$ = 4 animals). **e** IFN-γ spot-forming cells from peripheral blood lymphocytes in four animals per groups determined by the ELISpot assay on day 7 after the rechallenge. Blood samples (200 μL) were collected from the ophthalmic veins. Flow dot plots (**f**) and statistical data of $T_{EM}$ and $T_{CM}$ (**g**) in the splenic lymphocytes analyzed by flow cytometry, representative of 4 animals with similar results.

intravenous administration of LRT nanomedicine exhibits that 5/6 mice bearing orthotopic lung tumor could be fully cured by complete tumor eradications, and could spontaneously recover without further treatments once rechalleged with LLC cancer cells, evidencing the rapid and sustained immunotherapeutic effect of LRT treatment. Further mechanism study reveals abundant M1-type macrophages, matured DC as well as CD8+ T cells and memory T cells in tumor microenvironment after LRT therapy, demonstrating that the excellent anti-tumor efficacy of LRT therapy has been achieved by activating not only innate immune system, but also adaptive immune system. More importantly, long-term immunological memory has been established by LRT therapy. We suggest this treatment strategy as a therapeutic modality for lung cancer innate/adaptive immunotherapy.

## Methods
### Ethical statement
This research complies with all relevant ethical regulations. Experiments were performed in agreement with the Animal Care Ethics Commission of Shanghai Tenth People's Hospital, Tongji University

School of Medicine (ID: SHDSYY-2018-Z0026, SHDSYY-2022-P0050 and SHDSYY-2023-Z0026-3).

### Chemical and reagents
$Zn(NO_3)_2 \cdot 6H_2O$, $Al(NO_3)_3 \cdot 9H_2O$, and NaCl were obtained from Sinopharm Chemical Reagent Co.. Dimethyl sulfoxide (DMSO, 99%) was provided by Tokyo Chemical Industry. Silane-PEG-COOH was purchased from Pengsheng Biotechnology. RRX-001, TTA-Q6 and cell-counting kit-8 were bought from MedChemExpress. Phalloidin-Tracker Red-594 and 4′, 6-diamidino-2-phenylindole (DAPI) were acquired from Beyotime. Propidium Iodide was purchased from Acros. RMPI 1640, GlutaMAX and penicillin/streptomycin were purchased from Gibco.

### Synthesis of Zn/Al LDH-PEG-COOH
$Zn(NO_3)_2 \cdot 6H_2O$ (0.396 g), $Al(NO_3)_3 \cdot 9H_2O$ (0.25 g), and urea (5 g) were homogenized into deionized water (100 mL) by magnetic stirring under argon protection for 30 min, followed by heating at 100 °C to allow the reaction for 18 h. Then the obtained precipitate was washed with DI water thrice and followed decarbonated by

treating it with a salt-acid mixed solution (1 M NaCl and 3.3 mM HCl) for 12 h at ambient temperature. After washed with DI water thrice, the obtained LDH and silane-PEG-COOH (50 mg) were homogenized in deionized water (100 mL) and reacted at 100 °C. After the reaction for 4 h, the suspension was centrifuged, and the obtained precipitate was washed with ethanol thrice to obtain Zn/Al LDH-PEG-COOH.

### Preparation of RRX-001/TTA-Q6-Zn/Al LDH-PEG

The RRX-001 (10 mg) and TTA-Q6 (10 mg) were homogenized in DI water (5 mL) with 1 mg CTAC dissolved, then stirred for 30 min. Subsequently, Zn/Al LDH-PEG-COOH (50 mg) was suspended and magnetic stirred for 12 h. Then the suspension was centrifuged, and the obtained precipitate was further washed with deionized water three times to obtain RRX-001/TTA-Q6-Zn/Al LDH-PEG.

### Characterization

TEM image, element mapping and EDS profile were obtained on a JEM-2100F microscope (JEOL). SEM image was acquired on Quattro S (ThermoFisher Scientific). XRD patterns were obtained on the Ultima IV X-ray diffractometer (Rigaku). FT-IR spectra were recorded on Nicolet iS 10 (Thermo Scientific). Zn and Al concentrations in the solution were determined by Agilent 715 inductively coupled plasma optical emission spectrometer (ICP-OES) (Agilent Technologies).

### Degradation of the nanocarrier and release of TTA-Q6/RRX-001

A 10 mg LRT was dispersed in PBS (10 mL, pH = 6.5/7.4) at 37 °C. An aliquot (0.5 mL) was extracted from the solution at different times (5, 10, 30 and 60 min; 2, 4, 6, 8, 16, and 24 h) and centrifuged. The released drug at varied time points was calculated according to the UV-Vis spectrum. Then the supernatant was used for determining the concentrations of released Zn/Al elements by ICP-OES.

### Evaluations of the drug loading capacity and encapsulation efficiency

The suspension before and after drugs loading was collected and diluted to 1/100 fold. Then the UV spectra of the suspension after dilution were measured and calculated the drug loading capacity and encapsulation efficiency.

### Cell culture

Cell Bank/Stem Cell Bank, Chinese Academy of Sciences provided Lewis lung carcinoma cell line LLC (Catalog number: TCM 7). Luciferase-tagged LLC cell line was provided by iCell Bioscience Inc (Catalog number: iCell-0078a). LLC-OVA cells were generated by stable transfection with chicken egg ovalbumin (OVA). These cell lines were cultured on 25 cm² cell culture flasks containing DMEM with the addition of 10% FBS and 1% penicillin/streptomycin. DC2.4 cells (Catalog number: U30099) were obtained from YoBiBiotech Co., Ltd., which were cultured in RPMI 1640 medium with the addition of 10% FBS and 1% penicillin/streptomycin.

### In vitro cellular uptake of LRT

14 mg of EDC, 6 mg of NHS, and 4 mg of LRT were dissolved in 1 mL of MES buffer (0.1 M, pH = 6.0). The mixture was stirred for 15 min at room temperature. After that, 1 mg of propidium iodide was dissolved in 0.1 mL PBS buffer (0.1 M, pH = 7.4), then added to the above solution, followed by reacting for 30 min. After dialysis, the PI-conjugated LRT was obtained. LLC cells were seeded into a six-well microplate with a density of $10^5$/well and cultured overnight. Then, PI-conjugated LRT was added with a 200 μg/mL concentration. After co-incubation for 2, 4, 8, and 24 h, LLC cells were washed three times with PBS and then stained with DAPI solution, followed by testing by a flow cytometer (BD Fortessa X20).

### Calcium flow analysis by flow cytometry

LLC cells were incubated in a 6-well plate. After being treated by LDH, LR, LT, and LRT with the concentration of 100 μg/mL for 12 h, cells were washed and redispersed in PBS, followed by the addition of Calbryte 520 (2 μM) and incubation for 30 min at 37 °C. Then the cells were washed with saline twice. Baseline data was acquired in the first 60 s by Fortessa X20. Then, calcium nitrate solution was added (final concentration of 1 mM), and data acquisition was continued.

### Calcium observation by CLSM

LLC cells were incubated in a confocal dish. After being treated by LDH, LR, LT, and LRT with the concentration of 100 μg/mL for 24 h, cells were washed, stained with Fluo-4 AM at a concentration of 1 μM and incubation for 30 min at 37 °C. Finally, the cells were washed with PBS twice, stained with H33342 and observation by Olympus FV1000.

### Bio-TEM observation

LLC cells were incubated in 25 cm² cell culture flasks overnight and then treated with LRT (100 μg/mL) for another 16 h. These cells were then collected by cell scraper and fixed by glutaraldehyde (2.5%) at room temperature to make ultrathin sections for observation.

### Western blotting

LLC cells were cultured in 25 cm² cell culture flasks overnight, then treated with LDH, LR, LT, and LRT with 100 μg/mL concentration in DMEM medium for 16 h. Next, wash the cells twice with pre-chilled PBS on crushed ice, then lyse the cells by adding ice-cold RIPA containing PMSF (1 mM). After complete cell lysis, collect the cell lysate into pre-chilled centrifuge tubes using a cell scraper. Subsequently, sonicate the cells three times for 5 seconds each at a sonication power of 300 W, using a probe with a 6 mm diameter. All of these sample-handling steps should be performed on crushed ice. Following this, centrifuge the samples at 12,000 rpm for 10 min at 4 °C, and use the resulting supernatant for subsequent experiments. After the standard procedures for western blotting, the proteins were transferred to polyvinylidene difluoride membranes. Then, the membranes were incubated with primary antibodies against anti-$\beta$-Actin (Cell Signaling Technology, catalog number: 3700S, clone number: 8H10D10, 1:1000 dilution), anti-BIP (Cell Signaling Technology, catalog number: 3177T, clone number: C50B12, 1:1000 dilution), anti-CHOP (Cell Signaling Technology, catalog number: 2895T, clone number: L63F7, 1:1000 dilution), and anti-IRE1$\alpha$ (Cell Signaling Technology, catalog number: 3294T, clone number: 14C10, 1:1000 dilution) overnight, followed by incubation with anti-rabbit IgG HRP-linked (Cell Signaling Technology, catalog number: 7074S, 1:1000 dilution) or anti-mouse IgG HRP-linked (Cell Signaling Technology, catalog number: 7076S, 1:1000 dilution) secondary antibodies for visualization with an imaging system.

### Immunofluorescence imaging of calreticulin

LLC cells were cultured in a confocal dish overnight and then co-incubated with LDH, LR, LT, and LRT with 100 μg/mL concentration in DMEM medium for 16 h. Then the cells were washed with PBS three times and fixed with 4% paraformaldehyde at room temperature for 30 min. After that, the cells were incubated with a blocking buffer (5% rabbit serum dissolved in PBS) for 60 min. Then, anti-calreticulin (Cell Signaling Technology, catalog number: 12238S, clone number: D3E6, 1:400 dilution) and those cells were co-incubated overnight at 4 °C and washed with PBS thrice. Subsequently, anti-rabbit IgG (H + L), F(ab')₂ Fragment (Alexa Fluor® 647 Conjugate) (Cell Signaling Technology, catalog number: 4414S, 1:1000 dilution) were added into a confocal dish and incubated at room temperature for 2 h followed by washing with PBS thrice. At last, those cells were stained with DAPI solution and observed by Olympus FV1000.

### CD47 and calreticulin expression analysis by flow cytometry

LLC cells were incubated in a 6-well plate. After being treated with LDH, LR, LT, and LRT with the concentration of 100 μg/mL for 16 h, cells were washed and digested into single-cell suspension with Accutase. Afterward, the single-cell suspension was incubated with anti-CD47-BV421 (Biolegend, catalog number: 127527, clone number: miap301, 1:25 dilution) and anti-calreticulin-AF488 (Cell Signaling Technology, catalog number: 62304S, clone number: D3E6, 1:50 dilution) for 30 min at 4 °C. Subsequently, the cells were washed with PBS containing 2% FBS twice and analyzed by Fortessa X20.

### Bone marrow derived macrophages isolation and culture

Bone marrow derived macrophages were isolated according to an established method[48] and were cultured in RMPI 1640 medium containing 10% FBS, penicillin (100 units/mL), streptomycin (100 μg/mL), M-CSF (20 ng/mL) and IL-4 (10 ng/mL).

### Stimulation of bone marrow derived macrophages

LLC cells were cultured in the upper chamber of the transwell system, then co-incubated with LDH, LR, LT, and LRT with the concentration of 50 μg/mL in DMEM medium for 16 h. After that, the fresh RPMI 1640 medium was replaced, and the bone marrow derived macrophages were seeded into the lower chamber of the transwell system, followed by culturing for another 24 h.

### Phagocytosis of tumor cells by bone marrow derived macrophages

The stimulated bone marrow derived macrophages were harvested by Accutase, and the untreated LLC cells were stained by CFSE. Subsequently, the bone marrow derived macrophages and LLC cells mentioned above were co-incubated in a V-bottom 96-well plate at a ratio of 10:1 (BMDMs: LLC). After culturing for 24 h, the cells were digested into single cells and stained with anti-F4/80-BV421 (Biolegend, catalog number: 123132, clone number: BM8, 1:160 dilution) and anti-CD11b-BV650 (Biolegend, catalog number: 101259, clone number: M1/70, 1:270 dilution), following by flow cytometric analyzing.

### Analyze cytokine concentrations of BMDMs

The supernatant of stimulated BMDMs was collected, and mouse TNF-α (MULTI SCIENCES, EK282), IL-1β (MULTI SCIENCES, EK201BHS), IFN-γ (MULTI SCIENCES, EK280) and IL-6 (Valukine™, VAL604G) ELISA kits quantified the cytokines released into the supernatant with a standard protocol.

### Phalloidin staining of the stimulated BMDMs

The stimulated BMDMs were washed with PBS thrice and fixed with 4% paraformaldehyde at room temperature for 30 min. After that, the cells were permeabilized by washing twice with PBS/0.1% saponin, followed by staining with Tracker Red-594-phalloidin and DAPI. After being washed thrice with PBS/0.1% saponin, the cells were observed by Olympus FV1000.

### Polarization analysis of BMDMs

The stimulated BMDMs were washed with PBS thrice and digested into single cells. After blocking with TruStain FcX™ PLUS (Biolegend, catalog number: 156604, clone number: S17011E, 1:200 dilution), the cells were resuspended in 100 μL stain buffer (2% FBS, 10 μL Brilliant stain buffer plus, 5 μL True-Stain Monocyte Blocker) containing anti-F4/80-BV421 (Biolegend, catalog number: 123132, clone number: BM8, 1:160 dilution), anti-CD11b-BV650 (Biolegend, catalog number: 101259, clone number: M1/70, 1:270 dilution) and anti-CD86-APC-Fire 750 (Biolegend, catalog number: 105046, clone number: GL-1, 1:160 dilution), and incubated for 30 min. The cells

were then washed with stain buffer (2% FBS) twice. Finally, the cells were resuspended in stain buffer (2% FBS) containing PI (1 μg/mL) and analyzed by flow cytometer.

### Polarization, killing effect, and swallowing function analysis of AMs

Primary alveolar macrophages were isolated according to an established method[49] and were cultured in RMPI 1640 medium containing 10% FBS, penicillin (100 units/mL), streptomycin (100 μg/mL), β-Mercaptoethanol (50 μM) and GM-CSF (20 ng/mL). The fresh medium was replaced every two days and the primary alveolar macrophages would double every 7–10 days. Alveolar macrophages were used for subsequent experiments when expanded to a sufficient number.

LLC cells were digested into single cells and stained with CFSE (2 μM) at 37 °C for 5 min. After being washed with PBS/2% FBS three times, those LLC cells and the stimulated AMs (same method as stimulating BMDMs) were co-incubated in a V-bottom 96-well plate at a 1:10 ratio. After being cultured for 24 h, the mixed cells were digested into single cells with Accutase and washed with PBS/2% FBS twice.

Subsequently, the single-cell suspension was incubated with TruStain FcX™ PLUS (Biolegend, catalog number: 156604, clone number: S17011E, 1:200 dilution) in a Laminar Wash 96-well plate (Curiox Biosystems Co., Ltd, Seoul, South Korea) to reduce non-specific binding to FcRs for 15 min, followed by staining with ANNEXIN V-PE in binding buffer for 30 min. Then, the cells were washed by laminar flow using the Curiox Laminar Wash System HT2000 for seven cycles. After that, the cells were resuspended in 50 μL stain buffer (2%FBS, 10 μL Brilliant stain buffer plus, 5 μL True-Stain Monocyte Blocker) containing anti-CD170-APC (Biolegend, catalog number: 155507, clone number: S17007L, 1:80 dilution), anti-CD11c-PE-Fire640 (Biolegend, catalog number: 161103, clone number: QA18A72, 1:330 dilution), anti-CD86-APC-Fire 750 (Biolegend, catalog number: 105046, clone number: GL-1, 1:160 dilution), and incubated for 30 min. The cells were then washed with the laminar flow for twelve cycles using the Curiox Laminar Wash System HT2000. Finally, the cells were resuspended in stain buffer (2% FBS) containing PI (1 μg/mL) and transferred to a U-bottom 96-well plate for acquisition in SONY ID7000 (5-laser; 355, 405, 488, 561, and 637 nm). The multi-spectrum data were unmixed by SONY ID7000 software. All staining processes mentioned above were on crushed ice. Data was analyzed with the Flowjo V10.8.1.

### Isolation and stimulation of dendritic cells

Dendritic cells were isolated from the bone marrow of about male 6-week-old C57BL/6J mice according to an established method[50]. LLC cells were cultured in the upper chamber of the transwell system, then co-incubated with LDH, LR, LT, and LRT with the concentration of 100 μg/mL in DMEM medium for 16 h. Following that, the BMDCs were seeded into the lower chamber of the transwell system and cultured for another 24 h in a fresh RPMI 1640 medium.

### Analyze cytokine concentrations of BMDCs

The supernatant of stimulated BMDCs was collected, and mouse TNF-α (MULTI SCIENCES, EK282), IL-1β (MULTI SCIENCES, EK201BHS), IL-6 (Bio-Techne Valukine™, VAL604G) and IL-12p70 (MULTI SCIENCES, EK212HS) ELISA kits quantified the cytokines released into the supernatant with a standard protocol.

### DC Cells' morphology observation

The stimulated BMDCs were fixed with 4% paraformaldehyde at room temperature for 30 min. After that, the cells were stained with DiI for 30 min at room temperature. After being washed thrice with PBS, the cells were stained with DAPI solution and observed by Olympus FV1000.

## Co-incubation of DC2.4 cells and the stimulated LLC cells

LLC cells were cultured in a 6-well plate, then co-incubated with LDH, LR, LT, and LRT with a concentration of 100 μg/mL in DMEM medium for 16 h. The cells were then incubated with anti-calreticulin (Cell Signaling Technology, catalog number: 12238S, clone number: D3E6, 1:400 dilution) and stained with anti-rabbit IgG (H + L), F(ab')₂ Fragment (Alexa Fluor 647 Conjugate) (Cell Signaling Technology, catalog number: 4414S, 1:1000 dilution) secondary antibody. Then, the cells were stained with Hoechst 33342 and washed twice with PBS. Subsequently, DC2.4 cells labeled with CFSE were added to the 6-well plate and co-incubated with those LLC cells for 2, 4, 6, 16, and 24 h and observed by Olympus FV1000.

## Flow cytometric analysis for maturation of BMDCs and proliferation of CD8⁺ T cells

Single-cell suspension of the spleen from OT-I mouse (Shanghai Model Organisms Center, Inc.) was prepared by gentle mashing and passing through a 70 μm mesh cell strainer. Subsequently, the cells were incubated with TruStain FcX™ PLUS (Biolegend, catalog number: 156604, clone number: S17011E, 1:200 dilution) on crushed ice for 15 min and stained with anti-CD8α-BV570 (Biolegend, catalog number: 100740, clone number: 53-6.7, 1:160 dilution) for 30 min. After being washed twice with PBS/2% FBS, the single-cell suspension was stained using PI. Finally, CD8⁺ PI⁻ T cells were sorted by SONY MA900. Then, the sorted T cells were stained with CFSE and set aside.

LLC-OVA cells were cultured 6-well plate, then co-incubated with LRT with the concentration of 100 μg/mL in DMEM medium for 16 h. Next, the fresh RPMI 1640 medium was replaced, and BMDCs were seeded into a 6-well plate and cocultured for another 24 h. After that, the stimulated BMDCs were digested into single cells with Accutase and washed with PBS/2% FBS twice. Subsequently, the stimulated BMDCs and the CD8⁺ T cells stained with CFSE were co-incubated in a U-bottom 96-well plate at a 1:5 ratio (protect from light).

After being cultured for 36 h, the mixed cells were digested into single cells with Accutase and washed with PBS/2% FBS twice. Then, the single-cell suspension was incubated with TrueStain FcX™ PLUS in a Laminar Wash 96-well plate (Curiox Biosystems Co., Ltd, Seoul, South Korea) to reduce nonspecific binding to FcRs for 15 min, followed by staining with anti-CD3ε-AF700 (Biolegend, catalog number: 152316, clone number: 500A2, 1:200 dilution), anti-CD8α BV570 (Biolegend, catalog number: 100740, clone number: 53-6.7, 1:160 dilution), anti-CD11c-PE-Fire 640 (Biolegend, catalog number: 161103, clone number: QA18A72, 1:330 dilution), anti-CD86-APC-Fire 750 (Biolegend, catalog number: 105046, clone number: GL-1, 1:160 dilution) and anti-CD80-PE (Biolegend, catalog number: 104707, clone number: 16-10A1, 1:270 dilution) for 30 min. Then, the cells were washed by laminar flow using the Curiox Laminar Wash System HT2000 for twelve cycles. Finally, the cells were resuspended in stain buffer (2% FBS) and transferred to a U-bottom 96-well plate for acquisition in SONY ID7000 (5-laser; 355, 405, 488, 561, and 637 nm). The multi-spectrum data were unmixed by SONY ID7000 software. All staining processes mentioned above were on crushed ice. Data were analyzed with the Flowjo V10.8.1.

## Orthotopic lung cancer model

Male C57BL/6J mice (6 weeks old) were purchased from Shanghai SLAC Laboratory Animal Co., Ltd. Mice were placed in lateral recumbency and fixed on the operating table after being anesthetized by gas anesthesia with isoflurane. The fur of the left chest was removed using a depilatory cream. After the left chest was swabbed with 75% alcohol, a small incision of about 5 mm was made at about 1.5 cm above the costal arch of the left anterior axillary line. The subcutaneous tissue was exposed to the chest wall, 500 μL of the cell suspension (about $4 \times 10^5$ cells) was mixed with 500 μL of Matrigel (high concentration), and

50 μL of LLC cells and Matrigel suspension were injected into the left lung of the mouse with an insulin injection needle before the rapid coagulation, and the needle was inserted about 3 mm. After the injection, the needle was withdrawn for 5 s, then the incision was closed with 3 M Vetbond™ tissue adhesive. After about eight days, the orthotopic lung tumor was analyzed by Micro-CT (Quantum GX, PerkinElmer). The maximal authorized tumor burden was 1500 mm³ and was not exceeded during the course of the study. Criteria for euthanasia were labored breathing, a 20% weight loss one week, and severe hypothermia ( < 32 °C). These mice were housed in ventilated stainless-steel cages under standard conditions (light: 12 h light/dark cycle, ambient temperature: 25 ± 2 °C, humidity: 60 ± 10%), which were fed with pellet food ad libitum and sterilized water.

## In vivo Biodistribution of LRT nanomedicine

An orthotopic lung cancer model was established using LLC-Luc cells. An 8 mg LRT was suspended in 10 mL ICG solution (20 μg/mL) and co-incubated overnight. After being washed three times with PBS, the obtained LRT-ICG was further suspended in 2 mL PBS. Orthotopic lung tumor-bearing mice were injected with PBS containing LRT-ICG (200 μL). At various time intervals (5 min, 0.5, 1, 2, and 4 h) after the mice were injected, the in vivo ICG fluorescence and bioluminescence imaging was performed using the in vivo imaging system (IVIS) Spectrum (PerkinElmer) with Living Image software. In addition, d-luciferin potassium salts (150 mg/kg) were injected intraperitoneally for 8 min before in vivo imaging.

## In vivo therapy by LRT

Thirty orthotopic lung tumor-bearing were randomly divided into five groups ($N = 6$), administered intravenously with PBS, LDH, LR, LT, or LRT (40 mg/kg) every other day for a total of three injections. The mice were imaged every week to monitor tumor spread by micro-CT imaging. Mice were sacrificed when the disease progressed, as per animal care requirements. In addition, to highlight the advantages of LRT over the TTA-Q6 and RRX-001 combination treatment, thirty orthotopic lung tumor-bearing were randomly divided into five groups ($N = 6$ animals), administered intravenously with PBS, LDH, RRX-001 + TTA-Q6, LDH + RRX-001 + TTA-Q6, or LRT (40 mg/kg) every other day for a total of three injections. The dosage of RRX-001 and TTA-Q6 were calculated according to the drug loading capacity of LRT. To show the relative contribution of AMs and DC-primed T cells in the anti-tumor effects of LRT in vivo, the Rag deficient mice were used to establish orthotopic lung cancer model and treated with LRT with the same method.

## In vivo safety evaluation of LRT nanomedicine

Healthy 6-week-old male C57BL/6J mice were randomly assigned into three groups and intravenously injected with PBS and LRT (dose: 20 and 40 mg/kg) for 28 days. Whole blood and serum were collected at predetermined time nodes for systematic blood routine examination and blood biochemistry analysis. Then, the prime organs were dissected and processed by hematoxylin and eosin (H&E) staining for scientific histological evaluation. In addition, 6-week-old mice were intravenously administered with 40 mg/kg LRT. Then, the urine and faeces at 6, 12, 24, 48, 72, 96, 120, 144, and 168 h were collected and treated with aqua regia. The Al element in urines and faeces were measured by ICP-OES and calculated metabolism rate. Moreover, the LRT was administered to tumor-free mice (6-week-old male C57BL/6J) via intravenous tail injections three times, with injections occurring every other day. On the eighth day, the serum and major organs were collected, then the concentrations of TNF-α and IL-12p70 in serum were measured through ELISA assay, and the major organs performed hematoxylin and eosin (H&E) staining.

## Flow cytometry analysis of immune cells from the tumor microenvironment

Orthotopic lung tumors were harvested from mice in different groups and split in half. Then half was fixed with 4% paraformaldehyde and used for a subsequent multiplex immunohistochemistry experiment. In contrast, the other was further cut into small pieces followed by digesting with the Mouse Tumor Tissue Dissociation Kit, gentleMACS C tubes, and the gentleMACS™ Octo Dissociator and filtering with 30 μm MACS SmartStrainers (Miltenyi Biotec). Subsequently, the single-cell suspensions were centrifuged and resuspended with RBC lysis buffer (BD Biosciences). After being co-incubated for 5 min at room temperature, the single-cell suspensions were treated with an equal volume of PBS/10% FBS buffer and centrifuged and washed with PBS buffer twice.

Then, about $2 \times 10^6$ cells were suspended in 30 μL PBS buffer and co-incubated with 20 μL PBS buffer containing Zombie UV (Biolegend, catalog number: 423107, 1:400 dilution) in a Laminar Wash 96-well plate (Curiox Biosystems Co., Ltd, Seoul, South Korea). After 30 min, the cells were washed seven times with laminar flow using Curiox Laminar Wash System HT2000. Subsequently, the cells were co-incubated with TruStain FcX™ PLUS (Biolegend, catalog number: 156604, clone number: S17011E, 1:200 dilution) to reduce nonspecific binding to FcRs for 15 min. During the incubation, the antibody cocktail containing anti-CD45-BUV395 (BD Bioscience, catalog number: 564279, clone number: 30-F11, 1:320 dilution), anti-F4/80-BV421 (Biolegend, catalog number: 123132, clone number: BM8, 1:160 dilution), anti-Ly6C-PB (Biolegend, catalog number: 128013, clone number: HK1.4, 1:1600 dilution), anti-CD8α-BV570 (Biolegend, catalog number: 100740, clone number: 53-6.7, 1:160 dilution), anti-NK1.1-BV605 (Biolegend, catalog number: 108739, clone number: PK136, 1:200 dilution), anti-CD11b-BV650 (Biolegend, catalog number: 101259, clone number: M1/70, 1:270 dilution), anti-CD206-BV711 (Biolegend, catalog number: 141727, clone number: C068C2, 1:160 dilution), anti-Ly-6G-BV785 (Biolegend, catalog number: 127645, clone number: 1A8, 1:80 dilution), anti-CD49b-AF488 (Biolegend, catalog number: 108913, clone number: DX5, 1:400 dilution), anti-CX3CR1-Percp-Cy5.5 (Biolegend, catalog number: 149009, clone number: SA011F11, 1:1250 dilution), anti-CD80-PE (Biolegend, catalog number: 104707, clone number: 16-10A1, 1:270 dilution), anti-CD64-PE-Dazzle 594 (Biolegend, catalog number: 139319, clone number: X54-5/7.1, 1:40 dilution), anti-CD11c-PE-Fire 640 (Biolegend, catalog number: 161103, clone number: QA18A72, 1:330 dilution), anti-CD127-PE-Cy5 (Biolegend, catalog number: 135015, clone number: A7R34, 1:130 dilution), anti-Siglec-F-PE-Cy7 (Biolegend, catalog number: 155528, clone number: S17007L, 1:270 dilution), anti-MHC-II-PE-Fire 810 (Biolegend, catalog number: 107667, clone number: M5/114.15.2, 1:625 dilution), anti-CD25-APC (Biolegend, catalog number: 101909, clone number: 3C7, 1:70 dilution), anti-CD3ε-AF700 (Biolegend, catalog number: 152316, clone number: 500A2, 1:200 dilution), anti-CD86-APC-Fire 750 (Biolegend, catalog number: 105046, clone number: GL-1, 1:160 dilution) and anti-CD4-APC-Fire 810 (Biolegend, catalog number: 100479, clone number: GK1.5, 1:670 dilution) were prepared in FACS buffer (1× PBS, 2 mM EDTA, and 0.5% FBS, 20% Brilliant stain buffer plus, 10% True-Stain Monocyte Blocker) following the best concentration determined by antibody titration. Next, the cells were stained with an antibody cocktail for 30 min and then washed by laminar flow using the Curiox Laminar Wash System HT2000 for twelve cycles. Finally, the cells were resuspended in stain buffer (2% FBS/PBS) and transferred to a U-bottom 96-well plate for acquisition in SONY ID7000 (5-laser; 355, 405, 488, 561, and 637 nm). The multi-spectrum data were unmixed by SONY ID7000 software. All staining processes mentioned above were on crushed ice. Data was analyzed with the Flowjo V10.8.1.

## Multiplex immunohistochemistry

After fixed with 4% paraformaldehyde, the tumors were dehydrated and embedded in paraffin. After slice preparation, the two micrometers of FFPE sections were dewaxed and rehydrated. Opal dye (Akoya) was used for multiplex IHC. In the first round antigen was retrieved with a pressure cooker (EDTA pH = 8.0) at 100 °C for 3 min. Slides were cooled to room temperature (RT), washed with water (1 time, 3 min) and incubated with $H_2O_2$ (3%) for 15 min, then washed with TBST (2 times, 3 min). Subsequently, slides were washed and blocked with blocking buffer (Cwbio) for 10 min. The primary antibody, anti-CD11c (CST, catalog number: 97585S, clone number: D1V9Y, 1:600 dilution), was incubated at RT for 30 min. Slides were washed and an HRP-conjugated secondary antibody was incubated at RT for 10 min. Opal 570 dye (1:50 dilution) was applied for 10 min after washing with TBST two times. Then the slide was boiled for 3 min in EDTA to remove antibodies from the previous round and washed with water (one time, 3 min) as well as TBST (two times, 3 min). This was repeated five more times using the following antibodies, anti-F4/80 (CST, catalog number: 70076S, clone number: D2S9R, 1:450 dilution, Opal 520), anti-CD86 (CST, catalog number: 19589S, clone number: E5W6H, 1:300 dilution, Opal 620), anti-CD3ε (CST, catalog number: 78588S, clone number: E4T1B, 1:400 dilution, Opal 480), anti-CD8α (Abcam, catalog number: ab217344, clone number: EPR21769, 1:1500 dilution, Opal 780) and anti-pan-CK (Abcam, catalog number: ab7753, clone number: C-11, 1:300 dilution, Opal 690). Nucleus were stained with DAPI (PerkinElmer). Ready-to-use secondary antibodies anti-rabbit IgG (ZSGB-BIO, catalog number: PV-6001, undiluted) or anti-mouse IgG (ZSGB-BIO, catalog number: PV-6002, undiluted) were used. Another adjacent section was stained with anti-Ly6G (CST, catalog number: 87048S, clone number: E6Z1T, 1:50 dilution, Opal 570), anti-Ki67 (Abcam, catalog number: ab16667, clone number: SP6, 1:400 dilution, Opal 690), and anti-NK1.1 (CST, catalog number: 39197S, clone number: E6Y9G, 1:300 dilution, Opal 520) using the same protocol. The fluorescence signals from two slides were merged and analyzed with HALO platform. In addition, the CD47/CHOP panel was stained with anti-CD47 (Elabscience, catalog number: E-AB-40365, 1:50 dilution, Opal 570) and anti-CHOP (Elabscience, catalog number: E-AB-70087, 1:1200 dilution, Opal 690). The CD45/CD31 panel were stained with anti-CD45 (CST, catalog number: 70257S, clone number: D3F8Q, 1:200 dilution, Opal 520) and anti-CD31 (CST, catalog number: 77699S, clone number: D8V9E, 1:200 dilution, Opal 570).

## Tunel assay

The tumor tissue sections prepared in the previous section were dewaxed and rehydrated. Then the sections were stained with Tunel assay according to the instructions provided by the supplier (Elabscience, E-CK-A321). Finally, the sections were stained with DAPI and scanned by Pannoramic MIDI II.

## Immune memory affects evaluation

Orthotopic lung cancer models were established on the cured and age- and sex-matched naive C57BL/6J mice following the same protocol and analyzed by micro-CT (Skyscan 1176 CMOS, Bruker). On day 7, blood samples (200 μL) were collected from the ophthalmic veins and the lymphocytes were isolated using density gradient centrifugation (lymphocyte separation medium). Subsequently, the lymphocytes were seeded at $5 \times 10^5$ cells/well in IFN-γ ELISPOT 96-well plate and incubated for 24 h. ELISpot assays were performed according to IFN-γ ELISpot Kit instruction (DAKEWE, 2210002). At the humane end points or the end of the experiment, the spleens of mice were collected and processed into a single-cell suspension. Then the cells were stained with Zombie UV (Biolegend, catalog number: 423107, 1:400 dilution), anti-CD45-BUV 395 (BD Bioscience, catalog number: 564279, clone number: 30-F11, 1:320 dilution), anti-CD3ε-AF700 (Biolegend, catalog number: 152316, clone number: 500A2, 1:200 dilution), anti-CD4-APC-Fire

810 (Biolegend, catalog number: 100479, clone number: GK1.5, 1:670 dilution), anti-CD8α-BV570 (Biolegend, catalog number: 100740, clone number: 53-6.7, 1:160 dilution), anti-CD44-PE (Biolegend, catalog number: 103024, clone number: IM7, 1:20 dilution) and anti-CD62L-BV421 (Biolegend, catalog number: 104436, clone number: MEL-14, 1:160 dilution) followed as previously mentioned.

### Reporting summary
Further information on research design is available in the Nature Portfolio Reporting Summary linked to this article.

## Data availability
The authors declare that all data needed to support the finding of this study are presented in the Article, Supplementary information and Source data file. A reporting summary for this article is available as a Supplementary Information file. Source data are provided with this paper.

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

## Acknowledgements

The authors acknowledge Mr. Xinhui Cui in Sony Biotechnology Inc. for his assistance of spectrum flow cytometry and flow cytometry cell sorting. The mouse in Figs. 1, 6b, 8a and Supplementary Fig. 32, the bone in Figs. 4a, 5a, the lung and CT icon in Figs. 6b, 8a and Supplementary Fig. 25 were designed by Freepik (http://www.freepik.com). We thank the website www.figdraw.com for some cartoon figures preparation. This work was financially supported by the CAMS Innovation Fund for Medical Sciences (Grant No. 2021-I2M-5-012, J.S.), National Natural Science Foundation of China (Grant No. 21835007, J.S.), Key Research Program of Frontier Sciences, Chinese Academy of Sciences (Grant No. ZDBS-LY-SLH029, J.S.) and Basic Research Program of Shanghai Municipal Government (Grant No. 21JC1406000, J.S.).

## Author contributions

Y.G., P.H., and J.S. designated the idea of this work. Y.G. and Q.B. synthesized the nanomedicine and performed in vitro and in vivo experiments. Y.G. wrote the whole manuscript. P.H. and J.S. supervised the project, revised the manuscript, and commented on it.

## Competing interests

The authors declare no competing interests.
