## [Peer Review File · Nature Communications]

Nanomedicine-based co-delivery of a calcium channel inhibitor and a small molecule targeting CD47 for lung cancer immunotherapyEditorial Note #1: Parts of this Peer Review File have been redacted as indicated to remove third-party material where no permission to publish could be obtained.

Editorial Note #2: The schematics in the figures on pages 15, 64, 65, 79, 81 and 87 in this Peer Review File are designed by Freepik www.freepik.com

REVIEWER COMMENTS

Reviewer #1 (Remarks to the Author): with expertise in cancer immunology, CD47

Guo et al, in this manuscript, described a Zn-Al LDH-based pH-responsive nanomedicine for the co-delivery of a calcium channel inhibitor and a CD47 inhibitor. The hypothesis of this study is that loading therapeutic payloads to nanomedicine would lead to improvements in either or all of the following categories such as therapeutic dosage window, overall toxicity, tumor-targeting drug enrichment, or therapeutic efficacy. However, utilizing two well-established compounds as therapeutic payloads, the authors compared LRT to the vehicle- or single payload-loaded LDH nanomaterial but failed to demonstrate the advantages of the LDH-based nanomedicine over, for example, systematically administered small compound. In addition, while it is interesting the authors demonstrated some in vitro phenotypic changes of alveolar macrophages and dendritic cells, the authors left out monocyte-derived macrophages in their in vitro assays. Monocyte-derived macrophages are the most abundant phagocytes in the LLC tumor tissue, according to the authors' own data.

Major:

It is important to note that comparing LRT with vehicle LDH or single-payload-loaded LR or LT is not a fair comparison as it is almost predictable that LRT will achieve better therapeutic efficacy. The authors should have instead addressed the crucial question of to what extent do the LRT nanomedicines surpass the TTA-Q6 and RRX-001 combination treatment.

Additionally, the authors need to demonstrate increased phagocytosis of tumor cells by phagocytes after LRT treatment, especially the advantage over TTA-Q6 and RRX-001 combination treatment. Chemotaxis assay does not necessarily equal active tumor cell removal by

myeloid phagocytes.

The authors focused on tissue-resident alveolar macrophages as the effector for their LLC tumor model. However, according to Fig.6, CD11b+ F4/80+ tumor associated macrophages are undoubtedly the most dominant tumor-infiltrating myeloid effector cells. It is unclear why the authors chose not to investigate the in vitro mechanism of action using bone marrow derived macrophage.

It will be beneficial for the authors to provide solid justification for using Ly6C as the sole master memory T cells marker.

Line 349 -353. There was no evidence of the tumor-tissue targeted accumulation of LRT, but it was largely sequestered by liver and spleen, which negates the merit of this design. The authors should address if the concern that simply administrating TTA-Q6 and RRX-001 may achieve better therapeutic efficacy than LRT.

Minor:

The reviewer has concerns about the accuracy of the flow cytometry data presented in the manuscript. Based on limited flow cytometry scatter plots shown in this manuscript, the data were poorly compensated. For examples: in Sup.18a and Sup.23, the compensation of CD45/Zombie UV appears to be suboptimal; similar case for CD86/CD80 in sup.13d.

Can the authors explain the rationale for choosing only male mice for in vivo efficacy validation in Fig. 5?

Line 233, 246, macrophages mediate tumor cell removal majorly via antibody-dependent cell phagocytosis (ADCP), instead of ADCC.

Fig.3a-c, it seems that there is no difference in LR/LT/LRT treatment in inducing chemotaxis, as well as cytokine production.

Reviewer #2 (Remarks to the Author): with expertise in nanotechnology for cancer therapy

This study reports a combinational immunotherapeutic strategy against lung tumor by establishing a sub-micro-sized LDH-based co-delivery system for dual small-molecular drugs to activate the antitumor effect of TAM, which subsequently induces DC maturation, and further presents antigens to T cells. Although the novelty of the LDH-based co-delivery system in this work is limited, the experiments are well organized with lots of data. This manuscript could be accepted for publication provided that the authors addressed the following issues:

Specific comments:

1. The organization of introduction should be better improved. Why is the sub-micron-sized layered double hydroxide nanosheet (> 800 nm) chosen as the nanocarrier? What is the advantage of the drug release in tumor stroma? In addition, how about the therapeutic efficiency of TTA-Q6 and RRX-001 released in tumor stroma, compared with the drug delivery into tumor cells?
2. The authors claim that the EPR effect is the driving force for the accumulation of the co-delivery system in the lung. However, the EPR effect of nanoparticles with the size larger than 800 nm should be less effective.
3. Generally, Zn^{2+} ions are toxic, and can induce immunogenic cell death (ICD) effect. What is the dose of Zn^{2+} ? The effect of Zn^{2+} on tumor cells should be investigated.
4. The average topographic height of LRT nanosheets is approximate 40 nm by AFM, which can't indicate the layered structure.
5. The drug loading capacity and encapsulation efficiency should be provided.
6. The language of the manuscript should be polished.

Reviewer #3 (Remarks to the Author): with expertise in calcium, ER stress, cancer therapy

In this work, Guo et al. show the generation of the so-called LRT nanomedicine, consisting of a layered double hydroxide (LDH)-based co-delivery system loaded with RRX-001 (an inhibitor of CD47, downregulating CD47 on tumor cell membrane) and TTA-Q6 (a specific T-

type calcium channel inhibitor, preventing calcium ion uptake by tumor cells), and its effect on the innate/adaptive immune system in lung cancer. They show evidence for the following reasoning: LRT nanomedicine interacts with the tumor cell leading to the inhibition of calcium influx and subsequent induction of ER stress, resulting in calreticulin (CRT) upregulation and CD47 downregulation in cancer cells. This would lead to the generation of M1-type alveolar macrophages and dendritic cell maturation, and finally to activated anti-tumor response of T cells. This view is supported by both in vitro and in vivo approaches. The study is of interest, really exhaustive, and rather complex, in the sense that many cell types and biochemical processes are expected to be affected by the LRT nanomedicines. However, authors focus only in some of those cells and processes that fix the above mechanism of action. In this regard, the extensive study carried out seems to be a little skewed by a complicated process in which one step influences a later one in individual cells, but it could be entirely plausible that the earlier processes are occurring at the same time in many different cell types with perhaps opposite effects. As mentioned above, the study is exhaustive and of great interest, but some results seem to be overestimated and interpreted in a rather biased way. Other results are not clear enough (some regarding appropriate controls and comparison between treatments), and the pieces of evidence shown to propose the claimed mechanism of action, as well as regarding an apparent higher presence of the nanomedicines in the tumor, is rather weak. Additional evidence and further discussion should be needed to support authors' claims.

Some specific points to be dealt with are listed below:

1. According to Fig. S6b, authors claim that LLC cells hardly take up the nanomedicines (less than 10%). Appropriate controls are missing in these experiments. However, in the tumor microenvironment (TME) as well as in the bloodstream, plenty of different cells, especially phagocytes could take up these nanomedicines and destroy them. How authors deal with this possibility? The experiment shown in Fig. S6 refers to LDH phagocytosis. Does this refer to empty nanomedicines? What about LRT? The evidence for a pH dependency to accumulate in the TME is not clear.
2. Fig. 2. Assuming that LLC cells do not take up significant amounts of nanomedicines, authors suggest that LRT treatment inhibits LLC cancer cell calcium uptake, leading to ER stress, CD47 downregulation and CRT upregulation. There, it seems that the induction of ER stress is the critical and triggering event in LRT action. The decrease in CD47 cell surface

expression is also obtained through LR treatment (Fig. 2g) without the onset of ER stress (Fig. 2, d-e). In order to show translocation of CRT to the cell surface, cells could be stained after permeabilization and without permeabilization to show the distinct subcellular localization. Alternatively, this differential subcellular localization could be checked by flow cytometry. Western blots regarding CRT and CD47 expression under different treatments could also be helpful.

3. Either L, LT or LRT nanomedicines promote apparently activation of alveolar macrophages (AMs), release similar amounts of cytokines (Fig. 3c-d), and induced a similar killing of LLC cells as assessed by measurements of the intensity of Hoechst 33342 into the supernatant (Fig. S9). These latter measurements could be further supported by more specific ways to detect cell death, and the claimed killing could be due to other ways different from the putative regulatory events on CD47 and CRT expression. AMs could kill tumor cells through different ways and they could be tested by the use of the corresponding specific inhibitors or by targeting specifically the proteins involved.

4. Where are shown the cell death percentages mentioned in lines 281-285? Is there any difference between the distinct nanomedicines?

5. In Figure 4, LT and LRT apparently promote similar effects. In this regard, LR, LT and LRT induce similar immune response stimulation (Fig. 3). Thus, what does LRT bring to LR or LT? Authors should highlight the advantages of LRT compared to LT or LR.

6. It is not clear what the tissue distribution shown in Figure S14c refers to. Does this refer to LRT nanomedicines? If so, what happens with empty nanomedicines and the other LR and LT nanomedicines? Authors claim that "(Figure S14c) a high accumulation rate of these nanoparticles in the tumor region" (lines 348-349). However Figure S14c shows a rather poor accumulation in the tumor, especially as compared to liver and spleen, and being in a similar range to other tissues. This also raises some concerns about liver and spleen toxicity, and therefore some additional data and discussion on this aspect should be included.

7. Authors indicate that the nanodrugs had a rather short half-life (27 min) (Figure S15); is the same for all nanomedicines shown here? If so, and treatments involve only three injections every other day, it is expected that very little amount of nanomedicines reach the tumor. Because the in vivo assays show remarkable antitumor activity following LRT treatment (Fig. 5), both effective and durable responses, it is suggested the LRT induces rapidly responses that were not present only following an ER stress, or LT and LR

treatments. Fig. 6d shows a similar percentage of CD45+ leukocytes in the tumors of different treatments, but LRT-8 promotes a very high presence of CD45+ leukocytes (over 85%) in the tumor. How many LRT jabs were given in LRT-8 in comparison to LRT-2, and to the other treatments (LDH, LR, LT)? Why do you choose this timing?

8. There is some variability in the schedules followed for the in vivo experiments. It is not clear enough what schedule was followed in each experiment. This should be specified and any variation could be commented and justified. Apparently, authors conducted exhaustively in vivo experiments with different nanomedicine treatments, but the results sometimes refer to all the treatments and sometimes to only LRT treatments. In this regard, a question lies in whether different treatments in vivo could lead to similar results, such as those seen in vitro. Authors claim a very nice mechanism of action in the Discussion section, but this is based basically in correlative data, but causative approaches should be achieved in order to further demonstrate this mechanism.

9. Can authors discuss or make a comparison between the LRT treatment and the action of CD47 and calcium channel inhibitors, used separately or in combination, in cancer? What makes LRT a better option for cancer treatment than the use of CD47 and calcium channel inhibitors separately? Concerns about toxicity, safety and tumor accumulation should be critical, and these aspects are not clear at the moment. Safety could be a major issue if drugs were accumulated mainly in liver and spleen.

10. A scheme summarizing the putative mechanism of action suggested by the authors for LRT, together with the pieces of evidence supporting each step, could be very helpful.

Reviewer #4 (Remarks to the Author): with expertise in lung immunology, cancer

In this manuscript Guo et al report the anti-tumor effects of a nanomedicine delivering system combining T-type calcium channel inhibitor (TTA-Q6) and a small molecule inhibitor of CD47 (RRX-001). They observed antitumor effects of this nanomedicine (LRT) in a mouse LLC lung cancer model, and they claim such antitumor effects are mediated via LRT-induced endoplasmic reticulum stress in LLC cells and the transport of calreticulin onto the tumor cell membrane, which activates anti-tumor functions of alveolar macrophages as well as antigen presentation by dendritic cells that favors anti-tumor adaptive T cell response and memory. While this is an interesting potential anti-cancer therapeutic strategy, the key

conclusions in this study are not convincingly supported by current experimental data. The reviewer has the following major and minor comments:

Major comments:

1. The authors observed an anti-tumor effect of LRT in vivo. They also observed AMs and DCs stimulation in vitro by LT/LR/LRT-treated LLC cells. However, the significance of AM activation and DC maturation in this in vivo antitumor effect remains speculative and are not formally addressed with experiments. In this regard, the authors need to analyze the phenotypic and functional changes in AMs and DCs in tumor-bearing lung tissues, before and after nanomedicine treatment.

2. In Figure 3a,c and Figure S9, the effects of LR, LT and LRT on AM activation are similar. Does this mean a redundant effects of RRX-001 and TTA-Q6 in this culture system, or alternative mechanisms are involved in AM activation? The authors need to use CRT blocking or equivalent experiment to show the roles of CRT upregulation in AM activation (Figure 3). Similarly, the impact of CRT upregulation on DC maturation (Figure 4) needs to be experimentally addressed.

3. The authors need also to exclude direct tumor cell cytotoxic effects of nanomedicine.

4. The authors claim that activated AMs can kill tumor cells directly via ADCC. However, in Figure 3e no exogenous antibodies were supplemented to the co-culture system. The authors need to clarify whether AMs kill LLC cells via ADCC.

5. In Figure 4e, the authors need to show T cell proliferation in all the five groups of DCs (Figure 4b,c). This will help to link the maturation of DCs to T cell activation.

6. The authors indicate that the in vivo anti-tumor effects of LRT involve anti-tumor effects of both activated AMs and T cells primed by DCs. They need to show the relative contribution of AMs and DC-primed T cells in the anti-tumor effects of LRT in vivo. To do this, the authors can use Rag deficient mice to show the roles of anti-tumor AMs in the

absence of T cells.

7. "As shown in Figure S19, the TNF- α and IL-12p70 levels in the serum of LRT-treated mice are largely upregulated as the typical markers of antitumor immune response". High concentrations of systemic inflammatory cytokines in serum may cause damage to the body. Can LRT treatment induce these inflammatory cytokines in mice without tumors?

Minor comments:

1. Some conclusions are based on speculation but not experimental evidence and should be revised. For example: line 74 "...activates the antitumor effects of TAMs, which subsequently induces DC maturation" (is DC maturation directly related to/caused by TAM activation?), line 426 "because the CD47 downregulation will not induce significant DC maturation.", line 444 "...subsequently the CD8+ T cells will migrate to tumor sites for killing tumor cells...".

2. English writing and typos should be revised. For example, line 94 "clinical"?, line 524 "lymphopenia"

Reviewer #1: Guo et al, in this manuscript, described a Zn-Al LDH-based pH-responsive nanomedicine for the co-delivery of a calcium channel inhibitor and a CD47 inhibitor. The hypothesis of this study is that loading therapeutic payloads to nanomedicine would lead to improvements in either or all of the following categories such as therapeutic dosage window, overall toxicity, tumor-targeting drug enrichment, or therapeutic efficacy. However, utilizing two well-established compounds as therapeutic payloads, the authors compared LRT to the vehicle- or single payload-loaded LDH nanomaterial but failed to demonstrate the advantages of the LDH-based nanomedicine over, for example, systematically administrated small compound.

In addition, while it is interesting the authors demonstrated some *in vitro* phenotypic changes of alveolar macrophages and dendritic cells, the authors left out monocyte-derived macrophages in their *in vitro* assays. Monocyte-derived macrophages are the most abundant phagocytes in the LLC tumor tissue, according to the authors' own data.

Major:

1. It is important to note that comparing LRT with vehicle LDH or single-payload-loaded LR or LT is not a fair comparison as it is almost predictable that LRT will achieve better therapeutic efficacy. The authors should have instead addressed the crucial question of to what extent do the LRT nanomedicines surpass the TTA-Q6 and RRX-001 combination treatment.

Response: Thanks for your constructive and helpful suggestions. To address your concerns, we redesigned and conducted animal experiments to assess the therapeutic effects by administrating LRT and free TTA-Q6 + RRX-001 combination on orthotopic lung tumor. As shown in **Supplementary Figure 18**, the LRT performed favorably much better therapeutic efficacy than TTA-Q6 + RRX-001. The advantages of LRT over the combined TTA-Q6 + RRX-001 treatment is reasoned as follows:

(1) **LRT performed higher tumor enrichment efficiency than small molecular drugs.** As shown in **Supplementary Figure 15**, about 9.4% $\text{ID}\cdot\text{g}^{-1}$ of LRT has accumulated at the tumor tissue. This accumulation efficiency is relatively high

compared to other nanomedicines published in literatures (mostly below than 5% ID•g⁻¹). (*Nature Reviews Materials*, 2016, 1(5): 1-12.) In fact, the tumor accumulation of small molecular drugs is also low. Even the currently available and most efficient tumor-targeted antibody-drug conjugates (ADC) have less than 0.1% tumor accumulation. (*Clinical Cancer Research*, 2011, 17(20): 6389-6397; *Journal of Cellular Physiology*, 2019, 234(5): 5628-5642.) Therefore, compared to simply administrating TTA-Q6 and RRX-001, LRT still offers a higher tumor enrichment advantage than others, leading to the better therapeutic efficacy of LRT than LDH+RRX-001+TTA-Q6.

(2) **LRT showed higher biosafety due to its better pH-responsiveness than the free-drug combination.** As well-known, immunotherapies based on CD47 blockade often result in severe anemia in patients since red blood cells also express high amounts of CD47 protein to prevent phagocytosis by immune cells in the bloodstream. This is a major factor preventing the clinical application of CD47 ICB therapies. (*New England Journal of Medicine*, 2018, 379(18): 1711-1721.) Owing to the favored pH responsiveness, LRT only release RRX-001 and TTA-Q6 in tumor microenvironment, guaranteeing its biosafety. As shown in **Supplementary Figure 18 h**, the body weight of the mice after administrating TTA-Q6 and RRX-001 decreased significantly, indicating that TTA-Q6 and RRX-001 combination treatment will cause considerable side effects.

Supplementary Figure 18. *In vivo* immunotherapeutic outcomes by various treatments for orthotopic Lewis lung tumor model on C57BL/6J mice ($N = 6$). (a) Representative lung 3-dimensional reconstruction and corresponding CT transverse section images from each group after varied treatments as indicated. (d, f) Quantifications of the healthy lung volumes by computational extraction through 3D reconstruction. The body weight fluctuation (h) and Kaplan-Meier survival curves (i) in between of LLC tumor-bearing mice after varied therapy.

In summary, compared with the administration of free drugs TTA-Q6 + RRX-001, LRT performed a markedly enhanced therapeutic efficacy.

2. Additionally, the authors need to demonstrate increased phagocytosis of tumor cells by phagocytes after LRT treatment, especially the advantage over TTA-Q6 and RRX-001 combo-treatment. Chemotaxis assay does not necessarily equal active tumor cell removal by myeloid phagocytes.

Response: Thank you for your insightful and valuable suggestions. Following your

suggestion, we evaluated the effect of LRT or TTA-Q6 and RRX-001 combination treatment on macrophage phagocytosis. Following your recommendations, we employed a transwell system to activate bone-marrow derived macrophages (BMDMs) using LRT- or TTA-Q6+RRX-001-treated LLC cells. Subsequently, these activated BMDMs were co-cultured with CFSE-labeled LLC cells in a V-bottom 96-well plate. The assessment of tumor cell phagocytosis by BMDMs was conducted immediately after a 24-hour period. As illustrated in **Supplementary Figure 9**, macrophages stimulated by LRT-treated LLC cells present almost the same phagocytic efficiency to those in the TTA-Q6 + RRX-001. Consistent with our response to Question 1, it is important to emphasize that the LDH nanocarrier offers the advantages of efficient drug accumulation within tumors, thereby mitigating side effects *in vivo*.

Supplementary Figure 9. Flow cytometric analyses of phagocytosis of tumor cells by macrophages after different treatment.

3. The authors focused on tissue-resident alveolar macrophages as the effector for their LLC tumor model. However, according to Fig.6, CD11b⁺F4/80⁺ tumor associated macrophages are undoubtedly the most dominant tumor-infiltrating myeloid effector cells. It is unclear why the authors chose not to investigate the *in vitro* mechanism of action using bone marrow derived macrophage.

Response: We thank the Reviewer for this insightful comment. The intricate nature of

the tumor immune microenvironment, such as the diversified array of macrophage types, including bone marrow-derived macrophages (BMDMs), alveolar macrophages (AMs), and interstitial macrophages (IMs), make the mechanistic investigations rather complicated (*Frontiers in Immunology*, 2022, 13: 827719). It has been reported that alveolar macrophages are responsible for the initiation and early stage development of lung cancer (*Nature*, 2021, 595(7868): 578-584.; *Nature Immunology*, 2023, 24(3): 423-438.; *Nature Immunology*, 2023, 24(3): 423-438; *Nature Communications*, 2023, 14(1): 143; *Cancer Cell*, 2023, 41(7): 1261-1275). Therefore, the alveolar macrophages were chosen to investigate the *in vitro* mechanism of action in our first draft. Indeed, as you pointed out, bone marrow-derived macrophages are more prevailing in the lung tumor immune environment than AMs. Therefore, to further address your concern, we reassessed the effects of LRT treatment on BMDMs, and the results have been selectively included in **Figure 3**, and accordingly, a part of the data in the original version of Figure 3 have been moved in the supporting information.

Figure 3. *In vitro* stimulation of immune response of BMDMs incubated with LLC upon LRT nanomedicine treatment. (a) Schematic illustration of BMDMs activation in a transwell system. LLC cells were placed in the upper chamber followed by various treatments for 16 h, then the medium containing different nanoparticles were removed and BMDMs were cultured in the lower chamber. (b) Quantifications of the TNF- α , IL-1 β , IL-6 and IFN- γ levels in the stimulated BMDMs suspensions. Data are expressed as means \pm SD ($N = 3$). (c) CLSM images of BMDMs after being co-incubated with the LDH/LR/LT/LRT-treated LLC cells. BMDMs were stained with Tracker Red-594 phalloidin and DAPI for cytoskeleton and nuclei imaging, respectively. Scale bar: 50 μ m. (d) Flow cytometry characterization of macrophage phagocytosis of cancer cells. (e) Flow cytometric analysis of macrophage polarization following various treatments.

4. It will be beneficial for the authors to provide solid justification for using Ly6C as

the sole master memory T cells marker.

Response: We thank the Reviewer for this insightful question. It is important to clarify that memory T cells were not the primary focus in designing our flow cytometric panel; instead, we chose to utilize the Ly6C marker to identify monocytes. Interestingly, following dimensionality reduction (**Figure 6c**), we observed a distinct division within the CD8⁺ T cell population based on the Ly6C marker (Cluster 12 and 13).

Figure 6c. High-dimensional data analysis on tumor-infiltrating leukocytes from different groups displaying x-shift clusters projected on two Opt-SNE dimensions. The overlay plot shows concatenated events from all 24 samples, while the density plots show the differential population distributions among representative individual samples.

Upon conducting an extensive review of existing literature, we uncovered abundant evidence supporting the use of Ly6C as a marker for identifying memory T cells (*Immunity*, 2011, 35(4): 633-646; *Immunology*, 2019, 157(1): 3-12; *Nature Immunology*, 2020, 21(4): 412-421; *Nature Immunology*, 2002, 3(6): 558-563; *The Journal of Experimental Medicine*, 2000, 192(4): 557-564; *Immunity*, 2013, 38(4): 805-817). This finding underscores the importance of dimensionality reduction analysis over conventional flow cytometry analysis.

In adherence to the 3Rs principles (Replacement, **Reduction**, and Refinement) of animal welfare, we chose not to replicate the experiment using widely accepted markers (CD44 and CD62L) to identify memory T cells. A tumor re-challenge experiment was

conducted in treated mice to further substantiate the presence of immunological memory post-LRT treatment. As depicted in **Figure 7b**, mice from the treated group exhibit spontaneous tumor regression upon tumor re-challenge, further validating the favored immune memory. To further support our findings, we evaluated the shifts in the proportion of memory T cells within mouse spleens using the established markers CD44 and CD62L. In comparison to the control group, the LLC cells in the re-challenged group display significantly elevated percentages of memory T cells in the spleen, including central memory (TCM, CD62L⁺CD44⁺) and effector memory T cells (TEM, CD62L⁻CD44⁺), as illustrated in **Figure 7f** and 7g. This provides additional support for the establishment of anti-tumor immune memory through LRT therapy, thereby further supporting the conclusions presented in **Figure 6c**.

Figure 7. Long-term immune memory effects. (b) Representative CT transverse section images of four biologically independent animals from each group after varied treatments as indicated. (f) Representative flow dot plots in the splenic lymphocytes analyzed by flow cytometry.

In summary, although the use of Ly6C for identifying memory T cells was not our initial intention, it does not compromise the scientific validity of the conclusions. The confirmed applicability of Ly6C for memory T cell identification, coupled with our subsequent experimental verification, collectively contribute to the validity and applicability of our conclusions. We extend our heartfelt gratitude for your thoughtful and professional review.

5. Line 349 -353. There was no evidence of the tumor-tissue targeted accumulation of LRT, but it was largely sequestered by liver and spleen, which negates the merit of this design. The authors should address if the concern that simply administrating TTA-Q6 and RRX-001 may achieve better therapeutic efficacy than LRT.

Response: Thanks for your constructive and helpful suggestion. As shown in **Figure 5a**, LRT accumulates more significantly in orthotopic lung tumors than in normal lung tissue, demonstrating its passive targeting capability towards tumor tissues, though nanomedicines will be inevitably concentrated in the liver and spleen (*Biomaterials*, 2018, 174, 41-53; *Nature Reviews Materials*, 2023, 8(4): 282-300.). As shown in **Supplementary Figure 15c**, approximately 9.4% ID•g⁻¹ of LRT accumulated within the tumor tissue, significantly higher than other reported nanomedicines (mostly below than 5% ID•g⁻¹, *Nature Reviews Materials* (2016, 1(5): 1-12). In contrast, even the most efficient tumor-targeted antibody-drug conjugates (ADC) presents lower than 0.1% tumor accumulation (*Clinical Cancer Research*, 2011, 17(20): 6389-6397; *Journal of Cellular Physiology*, 2019, 234(5): 5628-5642). So simply administrating TTA-Q6 and RRX-001 will not achieve better therapeutic efficacy than LRT.

Figure 5a. *In vivo* biodistribution evaluation of LRT nanomedicines. **Supplementary Figure 15c.** Distribution of Al element in main organs (hearts, livers, spleens, lungs, and kidneys) and tumors of mice at 1 h points after LRT administration.

To further address your concern, we have carefully evaluated the therapeutic efficacy of both LRT and TTA-Q6+RRX-001. As demonstrated in **Supplementary Figure 18**, LRT exhibit enhanced therapeutic effectiveness when compared to TTA-Q6

and RRX-001 alone, which can be attributed to two main reasons presented in the responses to Question 1.

Supplementary Figure 18. *In vivo* immunotherapeutic outcomes by various treatments for orthotopic Lewis lung tumor model on C57BL/6J mice ($N = 6$). (a) Representative lung 3-dimensional reconstruction and corresponding CT transverse section images from each group after varied treatments as indicated. (d, f) Quantifications of the healthy lung volumes by computational extraction through 3D reconstruction. The body weight fluctuation (h) and Kaplan-Meier survival curves (i) in between of LLC tumor-bearing mice after varied therapy.

Minor:

6. The reviewer has concerns about the accuracy of the flow cytometry data presented in the manuscript. Based on limited flow cytometry scatter plots shown in this manuscript, the data were poorly compensated. For examples: in Sup.18a and Sup.23, the compensation of CD45/Zombie UV appears to be suboptimal; similar case for

Response: We greatly appreciate your insightful suggestion. We apologize that the original compensation values in **Supplementary Figure 14d (S13d in the original version)** are wrong and now have been appropriately rectified in the revised supporting information.

Supplementary Figure 14d. Gating strategy for DCs maturation and T cells proliferation flow cytometric analysis.

In **Supplementary Figure 23a**, the compensation value for BUV395 against Zombie UV appears slightly larger. Due to the large variations in the proportions of dead cells across different samples, identifying a compensation value for both BUV395 and Zombie UV can be challenging. Consequently, during the compensation adjustment process, our primary objective was to accurately discriminate the Zombie UV⁻ BUV395⁺ cell population from others. This approach primarily impacted the Zombie UV⁺ cells, thereby having minimal influence on our intended target analysis group (Zombie UV⁻ BUV395⁺ cell population). This data processing method is not uncommon in literatures (*Cytometry A*, 2013, 83(6): 530-2; *Cytometry A*, 2015, 87(12): 1067-9; *Cells Tissues Organs*, 2023, 212(1): 84-95.). By a thorough assessment, we have effectively calibrated the compensation values for each fluorescence channel within the Zombie UV⁻ BUV395⁺ cell population (**Figure R1**), thereby avoiding any

adverse impact on subsequent analyses.

Figure R1. The compensation of CD45⁺Zombie⁻ cells in Supplementary Figure 23a.

In **Supplementary Figure 28**, the cells under consideration were fixed prior to the acquisition, resulting in significant autofluorescence from a fraction of cells, which, however, is not our target population and thus does not significantly influence subsequent analyses. Our detailed examination has optimally rectified compensation values. As depicted in **Figure R2**, any attempt to increase or decrease the compensation value would lead to erroneous fluorescence spillover.

Figure R2. Schematic diagram of adjusting the compensation value.

We wish to present our sincere gratefulness once more for your thorough and professional review, which has markedly consolidated the conclusions of our manuscript.

7. Can the authors explain the rationale for choosing only male mice for *in vivo* efficacy validation in Fig. 5?

Response: Thanks for the comment! In our initial experiments, we consulted relevant literature to establish *in situ* lung tumor model (*Cancer Science*, 2020, 111(3): 951-61). In this literature, only male mice were used, so we also conducted our evaluations on the treatment effects of LRT only on male mice. To mitigate potential variability arising from sex-related factors, it is our standard practice in nanomedicine research to use mice of single-sex for animal experiments (*Nature Communications*, 2022, 13(1): 110; *Nature Communications*, 2021, 12(1): 3393). Of course, the sexuality in animal experiments may matter in the investigations, and we plan to explore the impact of sexuality on the efficacy of nanomedicines in our future work. Thank you again for your professional and insightful comments.

8. Line 233, 246, macrophages mediate tumor cell removal majorly *via* antibody-dependent cell phagocytosis (ADCP), instead of ADCC.

Response: Thanks for your professional comment and sorry for our mistake. In the revised manuscript, related mistakes have been corrected.

9. Fig.3a-c, it seems that there is no difference in LR/LT/LRT treatment in inducing chemotaxis, as well as cytokine production.

Response: Thanks for your comment. Indeed, differences in the macrophage chemotaxis and cytotoxicity results were not prominent enough. Previous researches have established that both the downregulation of CD47 in cancer cells and the upregulation of CRT can induce M1-type polarization of macrophages (*Proceedings of the National Academy of Sciences*, 2012, 109(17): 6662-6667; *Trends in Immunology*, 2018, 39(3): 173-184; *Cancer Cell*, 2023, 41(6):1014-1016; *Nature Immunology*, 2022, 23(4), 487–500).

In our endeavor to qualitatively investigate the potential of LR/LT/LRT-treated cancer cells to effectively trigger an anti-tumor response in macrophages, we intentionally employed high doses of nanomedicines and nanomedicines-treated cancer cells in the experimental setup (**Figure 3a-c in the original manuscript**).

Figure 3. (a) Chemotaxis rate of AMs towards LLC cells induced by the LDH/LR/LT/LRT nanosheets treatments. (b) Schematic illustration of AMs activation in a transwell system. LLC cells were placed in the upper chamber followed by various treatments for 16 h, then the medium containing different nanoparticles were removed and AMs were cultured in the lower chamber. (c) Quantifications of the TNF- α , IL-1 β , IL-6 and IFN- γ levels in the stimulated AMs suspensions. Data are expressed as means \pm SD ($N = 3$).

This approach ensured the activations of all macrophages , thus minimizing the discrepancies among the three experimental groups (more than 99%, **Table S2**). Notably, it is important to highlight that LRT has the capability to concurrently enhance CRT expression and diminish CD47 expression in cancer cells. This dual modulations aids in preventing immune evasion by cancer cells through the *in vivo* upregulation of CD47.

Table R1. Overview of the response of the question of Reviewer 1.

Order	Question	Response	Where
1	The advantages of LRT over small molecular drugs single usage.	[x] Explanation [x] Supplementary experiments [x] Modify text	Line 390-399 in the manuscript Pages 8 and 23 in the Supplementary Information Pages 1-3 in this Response
2	Evaluate the increased phagocytosis of tumor cells by phagocytes after LRT.	[x] Explanation [x] Supplementary experiments [x] Modify text	Line 239-244 in the manuscript Pages 17 in the Supplementary Information Pages 3-4 in this Response
3	Why choose AMs not BMDMs in the experiment in vitro ?	[x] Explanation [x] Supplementary experiments [x] Modify text	Line 205-254 in the manuscript Pages 17 in the Supplementary Information Pages 4-6 in this Response

Order	Question	Response	Where
4	The justification for using Ly6C as the sole master memory T cells marker.	[x] Explanation [ ] Supplementary experiments [ ] Modify text	Pages 7-8 in this Response
5	Tumor accumulation of LRT and the therapeutic efficacy of LRT and small molecular drugs.	[x] Explanation [x] Supplementary experiments [x] Modify text	Line 390-399 in the manuscript Pages 8 and 23 in the Supplementary Information Pages 9-10 in this Response
6	The compensations in some flow cytometric results.	[x] Explanation [ ] Supplementary experiments [x] Modify figures	Figure 4d and Supplementary Figure 14 Pages 11-13 in this Response

Order	Question	Response	Where
7	The reason of choosing only male for investigating in vivo efficacy validation.	[x] Explanation [ ] Supplementary experiments [ ] Modify text	Pages 13 in this Response
8	The wrong statement of ADCC.	[ ] Explanation [ ] Supplementary experiments [x] Modify text	Line 265-267 in the manuscript Pages 14 in this Response
9	Why the cytokine production and chemotaxis are same in LR/LT/LRT group?	[x] Explanation [ ] Supplementary experiments [ ] Modify text	Pages 14-15 in this Response

Reviewer #2: This study reports a combinational immunotherapeutic strategy against lung tumor by establishing a sub-micro-sized LDH-based co-delivery system for dual small-molecular drugs to activate the antitumor effect of TAM, which subsequently induces DC maturation, and further presents antigens to T cells. Although the novelty of the LDH-based co-delivery system in this work is limited, the experiments are well organized with lots of data. This manuscript could be accepted for publication provided that the authors addressed the following issues:

Specific comments:

1A. The organization of introduction should be better improved. Why is the sub-micron-sized layered double hydroxide nanosheet (> 800 nm) chosen as the nanocarrier? What is the advantage of the drug release in tumor stroma?

Response: We thank the Reviewer for this insightful comment. The selection of nanosheets with sub-micron dimensions was based on the operational mechanism of calcium channel inhibitors. As illustrated in **Figure R3**, calcium inhibitors can only enter calcium channels from outside the cell, following by forming stable coordination complexes with the protein components constituting the channel to prevent calcium ions enter the cells, and **the calcium inhibitor can not enter the calcium channels from inside the cell.**

In light of this, a larger-sized drug carrier was purposely chosen to prevent the internalization by cancer cells. This strategy ensures the external release of calcium channel inhibitors, thereby maximizing their efficacy.

[FIGURE REDACTED]

Figure R3. (a) Schematic diagram of calcium ion channel flow. (b) Mechanism of action of calcium ion channel inhibitors. (*Nature*, 576(7787), 492-497)

To clarify the issue, we revised the *Introduction* in the revised manuscript in a more reasonable way.

1B. In addition, how about the therapeutic efficiency of TTA-Q6 and RRX-001 released in tumor stroma, compared with the drug delivery into tumor cells?

Response: We appreciate your insightful comment. To evidence the size-effect of intracellular internalization of nanoparticles outlined in response to Question 1A, we also prepared smaller-sized nanosheets (<100 nm) through fragmentation the larger ones by sonication. Illustrated in **Figure R4**, these smaller-sized nanomedicines can be rapidly internalized by cancer cells, leading to much diminished intracellular calcium depletion within cancer cells in comparison with their larger counterparts. As exemplified in **Figure 2**, calcium depletion underlies induced CRT regulation and subsequent immune activation, while the smaller-sized nanomedicines cannot induce it (**Figure R4c**). Therefore, larger-sized (>800 nm) nanomedicine was fabricated and used

in this study. .

Figure R4. (a) TEM image of the LRT-small (the LDH nanocarrier was prepared by sonicating large-sized nanosheets.), scale bar: 100 nm. (b) LLC cell uptake flow cytometric analysis treated with 200 $\mu\text{g}/\text{mL}$ LRT-small. (c) CLSM images of LLC cells after different treatments for 24 h. Fluo-4 was used for cellular calcium observation. Scale bars, 50 μm .

2. The authors claim that the EPR effect is the driving force for the accumulation of the co-delivery system in the lung. However, the EPR effect of nanoparticles with the size larger than 800 nm should be less effective.

Response: Thanks for your professional question. In fact, the mechanism of nanomaterial accumulation in tumors is extremely complicated and has not yet been definitively determined. (*ACS Nano*, 2015, 9(7): 6655–6674; *Biomaterials*, 2018, 174, 41-53). Although several literatures reported that the EPR effect is less significant for nanoparticles larger than 800 nm, researches have also proposed different mechanisms

of nanomaterial's accumulation in tumors from the EPR effect, such as the transportation of endothelial cells (*Nature Materials*, 2020, 19(5): 566–575).

In this work, the *in vivo* investigations did reveal a considerable accumulation of nanoparticles larger than 800 nm within tumors (**Supplementary Figure 15** and **Figure 5a**). Nonetheless, the precise mechanism of nanoparticle's penetration into tumors remains to be clarified.

Figure 5a. *In vivo* biodistribution evaluation of LRT nanomedicines. **Supplementary Figure 15.** (a) *In vivo* 2D bioluminescence of LLC-Luc cells. (b) Quantification of the fluorescence intensity at the left lung area of LLC tumor-bearing mice taken at preset time points post *i.v.* injection of LRT. (c) Distribution of Al element in main organs (hearts, livers, spleens, lungs, and kidneys) and tumors of mice at 1 h points after LRT administration. Data are expressed as mean \pm SD ($N = 3$).

To avoid any uncertainty, we have deleted the description about nanomedicine's accumulation in tumors through EPR effect in the revised manuscript, and revised the original statement to read: "As illustrated in **Scheme 1**, mild acidity-responsive and laterally micro-sized Zn-Al LDH was chosen as the drug carrier to load and deliver two kinds of small-molecule drugs, calcium channel inhibitor, TTA-Q6, and CD47 inhibitor,

RRX-001" (Line 78-80 in the revised manuscript).

3. Generally, Zn^{2+} ions are toxic, and can induce immunogenic cell death (ICD) effect. What is the dose of Zn^{2+} ? The effect of Zn^{2+} on tumor cells should be investigated.

Response: We sincerely appreciate your concern regarding the potential toxicity of Zn^{2+} ions. Nevertheless, zinc has been recognized to be an important element for human body, and their toxicity, as reflected by the LC_{50} values, is highly dependent on its concentration/dosage (about 1 mM for cancer cells, *Nano Research*, 2009, 2(11):882-890). A previous study of our research group, as illustrated in **Figure R5**, evaluated the toxicity evaluation of zinc ions, which reveals that even at a concentration of 600 μ M, zinc ions maintained a non-toxic profile toward cancer cells (Yuedong Guo, Ping Hu* and Jianlin Shi*, *ACS Nano*, 2021, 15(10): 16286–16297).

[FIGURE REDACTED]

Figure R5. Viability assays of cancer cells after treated with different concentrations of Cu^{2+} , Fe^{2+} or Zn^{2+} for 24 h.

In the current study, the cumulative Zn^{2+} dosage within the tumor is approximately 105.5 μ M (calculated based on **Supplementary Figure 15c**), which is ineffective to trigger immunogenic death from the above **Figure R5**. Furthermore, as shown in **Figure 1k**, the gradual degradation of LRT was observed at pH 6.5, facilitating the slow release of Zn^{2+} . This implies that the actual concentration of zinc ions within the tumor is most probably much lower than the theoretical maximum value of 105.5 μ M, due to the continuous metabolism out of the body through vascular system.

Figure 1k. Accumulated release profiles of Zn element from LRT in PBS of different pH values, $N = 3$.

Furthermore, **Figures 3c** and **4b** reveal that cancer cells treated with LDH fail to activate macrophages and dendritic cells (DC), indicating that the Zn^{2+} released from LDH is inadequate to induce immunogenic death.

Figure 3c. Quantifications of the TNF- α , IL-1 β , IL-6 and IFN- γ levels in the stimulated AMs suspensions. Data are expressed as means \pm SD ($N = 3$). **Figure 4b.** Quantifications of the secretion levels of TNF- α , IL-1 β , IL-6, and IL-12p70 in the stimulated BMDCs suspensions. The data are expressed as means \pm SD ($N = 3$).

To further address your concern, we undertook supplementary cell and animal experiments to further examine the cytotoxicity of zinc ions. Initially, varied concentrations of LDH were co-incubated with cancer cells for 24 hours, and subsequently, CCK-8 staining was conducted to assess any changes in cell viability. As depicted in **Figure R6**, the viability of cancer cells upon treated with LDH remains essentially unchanged.

Figure R6. Viability assays of cancer cells after treated with different concentrations of LDH for 24 h

Furthermore, a TUNEL assay was applied to stain tumor tissue sections obtained from mice treated with LDH. Supplementary Figure 22 demonstrates that cancer cells in the LDH-treated group exhibit negligible apoptotic signals similar to the control group.

Supplementary Figure 22. TUNEL staining of tumor sections after LDH treatment three times. Scale bars: 100 µm.

4. The average topographic height of LRT nanosheets is approximate 40 nm by AFM, which can't indicate the layered structure.

Response: We are extremely grateful to the comment of the Reviewer and apologize for inaccurate descriptions. We have rectified the claim by providing a more suitable and accurate description: "The average topographic height of LRT nanosheets was determined to be approximate 40 nm by atomic force microscopy (AFM) (**Figure 1h**)". (Line 125-126, Page 6)

5. The drug loading capacity and encapsulation efficiency should be provided.

Response: We appreciate your question. The overall drug loading capacity was 18.2 wt% (7.2 wt% for TTA-Q6 and 11 wt% for RRX-001), while the overall encapsulation efficiency was 47.9 wt% (41 wt% for TTA-Q6 and 62 wt% for RRX-001). These values were obtained by UV spectra analysis of the drug solution, both before and after drug loading (**Supplementary Figure 5**).

Supplementary Figure 5. UV absorbance of the supernatant before and after drug loading.

6. The language of the manuscript should be polished.

Response: Thanks for your kind suggestion. The language has been carefully polished as far as possible to improve the readability and clarity. We hope the revised expressions are clear and accurate enough now.

Table R2. Overview of the response of the question of Reviewer 2.

Order	Question	Response	Where
1	Why choose a nanocarriers larger than 800 nm?	[x] Explanation [x] Supplementary experiments [x] Modify text	Line 74-83 in the manuscript Pages 19-21 in this Response
2	The EPR effect of the large-sized nanosheet.	[x] Explanation [ ] Supplementary experiments [x] Modify text	Line 80-83 in manuscript Pages 21-23 in this Response
3	The effect of zinc ions on tumor cells.	[x] Explanation [x] Supplementary experiments [x] Modify text	Line 440-449 in manuscript Pages 26 in the Supplementary Information Pages 23-25 in this Response

Order	Question	Response	Where
4	The imprecise description about AFM image.	[ ] Explanation [ ] Supplementary experiments [x] Modify text	Line 125-126 in manuscript Pages 25 in this Response
5	The drug loading capacity and encapsulation efficiency.	[x] Explanation [x] Supplementary experiments [x] Modify text	Line 145-148 in manuscript Pages 26 in this Response
6	The language of the manuscript should be polished.	[ ] Explanation [ ] Supplementary experiments [x] Modify text	Multiple places in manuscript Pages 26 in this Response

Reviewer #3: In this work, Guo et al. show the generation of the so-called LRT nanomedicine, consisting of a layered double hydroxide (LDH)-based co-delivery system loaded with RRX-001 (an inhibitor of CD47, downregulating CD47 on tumor cell membrane) and TTA-Q6 (a specific T-type calcium channel inhibitor, preventing calcium ion uptake by tumor cells), and its effect on the innate/adaptive immune system in lung cancer. They show evidence for the following reasoning: LRT nanomedicine interacts with the tumor cell leading to the inhibition of calcium influx and subsequent induction of ER stress, resulting in calreticulin (CRT) upregulation and CD47 downregulation in cancer cells. This would lead to the generation of M1-type alveolar macrophages and dendritic cell maturation, and finally to activated anti-tumor response of T cells. This view is supported by both *in vitro* and *in vivo* approaches. The study is of interest, really exhaustive, and rather complex, in the sense that many cell types and biochemical processes are expected to be affected by the LRT nanomedicines. However, authors focus only in some of those cells and processes that fix the above mechanism of action. In this regard, the extensive study carried out seems to be a little skewed by a complicated process in which one step influences a later one in individual cells, but it could be entirely plausible that the earlier processes are occurring at the same time in many different cell types with perhaps opposite effects. As mentioned above, the study is exhaustive and of great interest, but some results seem to be overestimated and interpreted in a rather biased way. Other results are not clear enough (some regarding appropriate controls and comparison between treatments), and the pieces of evidence shown to propose the claimed mechanism of action, as well as regarding an apparent higher presence of the nanomedicines in the tumor, is rather weak. Additional evidence and further discussion should be needed to support authors' claims.

Some specific points to be dealt with are listed below:

1. According to Fig. S6b, authors claim that LLC cells hardly take up the nanomedicines (less than 10%). Appropriate controls are missing in these experiments. However, in the tumor microenvironment (TME) as well as in the bloodstream, plenty of different cells, especially phagocytes could take up these nanomedicines and destroy them. How

authors deal with this possibility? The experiment shown in Fig. S6 refers to LDH phagocytosis. Does this refer to empty nanomedicines? What about LRT? The evidence for a pH dependency to accumulate in the TME is not clear.

Response: We thank the Reviewer for the kind and insightful comments. The control in **Supplementary Figure 7b** has been added into the revised supporting information following your suggestion.

Supplementary Figure 7. (a) Gating strategy for flow cytometry LLC cells phagocytosis analysis. (b) Flow cytometry analysis of phagocytosis of LRT by LLC cells.

Indeed, immune cells like macrophages will uptake nanomedicines, contributing to the relatively lower accumulation rate of nanomedicines within tumors (*Accounts of Chemical Research*, 2013, 46(3): 761-769.). Although it is currently not feasible to entirely prevent macrophages from phagocytosing nanomedicines, we can make efforts to minimize this impact. To address this, we modified the surface of the LDH nanocarrier with PEG, a widely employed method to enhance nanomedicine biocompatibility and reduce immune cell phagocytosis (*Nature Biomedical Engineering*, 2017, 1(2): 0029.). As highlighted in **Supplementary Figure 15**, approximately $9.4\% \text{ ID}\cdot\text{g}^{-1}$ of LRT accumulated in tumor tissue, significantly higher than other nanomedicines reported in previous literatures, which is mostly below $5\% \text{ ID}\cdot\text{g}^{-1}$ (*Nature Reviews Materials*, 2016, 1(5): 1-12). As demonstrated in **Figure 1j**,

LRT releases a majority of loaded drugs in 10 hours under a mildly acidic environment similar to that of the tumor. This indicates significant drug release from LRT before potential phagocytosis by macrophages. Hence, even though macrophages in the tumor microenvironment may uptake LRT to a large extent, the therapeutic efficacy of the small molecule drug released from LRT is still evident thanks to the significant LRT accumulations in tumors.

Figure 1j. Release profiles of TTA-Q6 and RRX-001 from a nanocarrier in the phosphate buffer saline (pH 6.5 and 7.4), $N = 3$.

In addition, we sincerely apologize for the error in the Figure caption, which has been rectified in the revised version, and the caption for **Supplementary Figure 7** now accurately describes the phagocytosing rate of LRT, rather than LDH, by LLC cells. All the descriptions provided in the Discussion section of the manuscript and the Method section of the supporting information refer to LRT. The error in the caption has been rectified.

Additionally, we apologize for any confusion caused by our unclear description of pH dependency for LRT accumulation in the tumor microenvironment. The related descriptions have been rewritten to clarify that LRT releases drugs in a pH-responsive manner, rather than accumulates in tumor in a pH-dependent manner.

2. Fig. 2. As summing that LLC cells do not take up significant amounts of nanomedicines, authors suggest that LRT treatment inhibits LLC cancer cell calcium

uptake, leading to ER stress, CD47 downregulation and CRT upregulation. There, it seems that the induction of ER stress is the critical and triggering event in LRT action. The decrease in CD47 cell surface expression is also obtained through LR treatment (Fig. 2g) without the onset of ER stress (Fig.2, d-e). In order to show translocation of CRT to the cell surface, cells could be stained after permeabilization and without permeabilization to show the distinct subcellular localization. Alternatively, this differential subcellular localization could be checked by flow cytometry. Western blots regarding CRT and CD47 expression under different treatments could also be helpful.

Response: Thanks for your kind comments. Based on your suggestions, in our supplementary experiments, the cells with or without LRT treatment were fixed, then CRT was stained through a standard immunofluorescence staining procedure either with or without permeabilization. As shown in **Figure R7**, there no significant difference between control and LRT groups after the cells were permeabilized, owing to the intrinsic characteristics of CRT. In fact, although a part of CRT will translocate from ER to plasma membrane when the cells are under ER stress, most CRT remain attached at ER. Therefore, regardless of the cells being under ER stress or not, there will be a persistent CRT positive signal in the cells after permeabilization. According to the recommendations provided by the official website of *Cell Signaling Technology* and the experimental protocols in most literatures (*Chemical Science*, 2023, 14(6):1461-1471), CRT staining of non-permeabilized cells is a favored choice for detecting ER stress and calreticulin translocation. Following a non-permeabilizing procedure, as shown in **Figure 2f**, significant red fluorescent signals can be observed in the LLC cells after the treatment with LT/LRT for 16 h, indicating that LRT nanomedicine has successfully induced CRT exposure in the cell plasma membrane.

Figure R7. CLSM images of the CRT immunofluorescence staining for the cells after different treatments, scale bar: 50 μm .

In addition, the regulations of CRT and CD47 expressions on cancer cells were also detected by flow cytometry. As shown in **Figure 2f**, the LLC cells without permeabilization after LRT treatment show stronger CRT and weaker CD47 signals than control group, clearly demonstrating the changes in the levels of CRT and CD47 localized at plasma membrane.

Figure 2. *In vitro* CD47 and CRT regulations by LRT nanomedicine on the plasma membrane of LLC cells. (f) CLSM images showing the CRT translocation of LLC cells treated with PBS, LDH, LR, LT or LRT. Blue and red fluorescences show DAPI-stained nucleus and CRT exposure on the cell surface, respectively. Scale bar, 50 μm . Cells were not permeabilized prior to immunofluorescence staining. (g) Flow cytometric analyses for CD47 and CRT regulations of LLC cells after various treatments.

3. Either L, LT or LRT nanomedicines promote apparently activation of alveolar macrophages (AMs), release similar amounts of cytokines (Fig. 3c-d), and induced a similar killing of LLC cells as assessed by measurements of the intensity of Hoechst 33342 into the supernatant (Fig. S9). These latter measurements could be further

supported by more specific ways to detect cell death, and the claimed killing could be due to other ways different from the putative regulatory events on CD47 and CRT expression. AMs could kill tumor cells through different ways and they could be tested by the use of the corresponding specific inhibitors or by targeting specifically the proteins involved.

Response: Thanks for your professional and constructive suggestion. In addition to Hoechst 33342 method, the tumor killing effect of macrophages was also evaluated by flow cytometry. As shown in **Figure 3e**, LLC cells in LRT group show 86% of late apoptotic cells (MC 1) and only 14% live cells (MC 2), indicating the potent tumor cell-killing effect of AMs in LRT group.

Figure 3e. Fast interpolation-based t-SNE (Fit-SNE) representations of AMs and LLC cell landscape, representative of 3 independent experiments. The AMs pre-stimulated by LRT nanomedicine-treated LLC cells were co-cultured with CFSE-labeled LLC cells in V-bottom 96-well plate for AMs immunophenotype, tumor cell phagocytosis and killing effect flow cytometry analyses.

As you kindly noted, the AMs could kill tumor cells through different ways. To further confirm the mechanism of cancer cell death by AMs, we stimulated macrophages with LRT-treated LLC cells and then co-cultured these activated macrophages with Hoechst 33342-labeled LLC cells in a 96-well V-bottom plate coated with IFN-gamma, TNF-alpha, IL-1beta and IL-6 antibodies which is to eliminate the potential killing effect of cytokines on the cancer cells. As shown in the **Figure R8**, in 24 hours of co-incubation, the cell death rate of the LRT + antibody group is not much

different from the control group, and is significantly lower than that of LRT alone group without antibody coating. So, the AMs, activated by the LLC cells featuring the lowered CD47 but elevated CRT expressions on cell surface, can kill cancer cells by secreting cytokines. This strongly supports that the observed cell death in **Figure 3e** is due to cytokines secreted by M1-type macrophages.

Figure R8. Killing effects on LLC cells by the stimulated macrophages, the cells were co-incubated in the V-bottom 96-well plate with/without coated with IFN-gamma, TNF-alpha, IL-1beta and IL-6 antibodies. Data are expressed as means \pm SD ($N = 4$).

4. Where are shown the cell death percentages mentioned in lines 281-285? Is there any difference between the distinct nanomedicines?

Response: Thanks for your question. The detailed percentages of cell death are outlined in **Table S2** in the Supplementary Information. Since the death percentages given in the main text was calculated on LLC cells alone, while the death ratio in **Table S2** was obtained based on the total amounts of LLC and AMs, which may cause misleading. Therefore, based on the original data in Table S2, we now present two separate Tables, Table S2.1 and 2.2, respectively for AMs and LLC cells.

Table S2.1 The percentages of LLC cells in Table S2.

Groups	Late apoptosis LLC cells	Live LLC cells
Control	7.22%	92.78%
LDH	8.87%	91.13%

LR	85.14%	14.86%
LT	84.36%	15.64%
LRT	85.85%	14.15%

As shown in **Table S2**, no significant differences in death percentages among the LR/LT/LRT groups can be observed. Previous researches suggest that CD47 downregulation or CRT upregulation occurring in cancer cells would lead to M1-type macrophage polarization (*Proceedings of the National Academy of Sciences*, 2012, 109(17): 6662-6667; *Trends in Immunology*, 2018, 39(3): 173-184; *Cancer Cell*, 2023, 41(6):1014-1016; *Nature Immunology*, 2022, 23(4): 487–500.). To qualitatively assess whether LR/LT/LRT-treated cancer cells can effectively trigger an anti-tumor response in macrophages or not, we employed a high-enough quantity of LR/LT/LRT-treated cancer cells to stimulate alveolar macrophages (AMs). This approach ensures the activation of all available macrophages due to the extensive stimulations. Subsequently, the excessive AMs of post-stimulation were co-cultured with LLC cells (AMs: LLCs=10:1). In terms of cell death percentages, no significant variations can be observed among the three experimental groups, owing to the elimination of all LLC cells by the excess AMs. Notably, in the revised manuscript, we re-evaluate the effects of LR, LT and LRT on macrophages activation at a lowered dosage of nanomedicines. It can be found that in this case, LRT induces significantly stronger activation effect on macrophages than LR and LT, further demonstrating the above perspective (**Figure 3b and d**).

5. In Figure 4, LT and LRT apparently promote similar effects. In this regard, LR, LT and LRT induce similar immune response stimulation (Fig. 3). Thus, what does LRT bring to LR or LT? Authors should highlight the advantages of LRT compared to LT or LR.

Response: We greatly appreciate your thorough review. Similar to our response to **Question 4**, LT and LRT groups performed similar effects in activating DC cells with each other (**Figure 4**), while LR/LT/LRT groups showed similar immune response stimulation among each other of macrophages (**Figure 3**).

Compared to LR and LT, LRT can simultaneously upregulate CRT and downregulate CD47 levels in cancer cells *in vitro*, as shown in **Figure 2**, which is effective in preventing cancer cells from evading the immune system, though the present rather short-term *in vitro* experiments may be not convincing enough to fully understand this intricate immune activation process.

Further, as illustrated in **Figure 5c and d**, LRT yielded a more pronounced therapeutic effect against lung cancer in comparison to LR and LT *in vivo*, owing to the combined regulations on CRT and CD47 expressions, highlighting its attractive performance.

Figure 5. (c) Representative lung 3-dimensional reconstruction and corresponding CT transverse section images from each group after varied treatments as indicated. Scale bar: 5 mm. (d) Quantifications of the healthy lung volumes by computational extraction through 3D reconstruction.

6. It is not clear what the tissue distribution shown in Supplementary Figure 14c refers to. Does this refer to LRT nanomedicines? If so, what happens with empty nanomedicines and the other LR and LT nanomedicines? Authors claim that “(Supplementary Figure 14c) a high accumulation rate of these nanoparticles in the tumor region” (lines 348-349). However, Supplementary Figure 14c shows a rather poor accumulation in the tumor, especially as compared to liver and spleen, and being in a similar range to other tissues. This also raises some concerns about liver and spleen toxicity, and therefore some additional data and discussion on this aspect should be included.

Response: We appreciate your valuable comment. As this comment contains several sub-comments, each of them will be addressed in detail in the following:

(1) It is not clear what the tissue distribution shown in Supplementary Figure 14c refers to. Does this refer to LRT nanomedicines? If so, what happens with empty nanomedicines and the other LR and LT nanomedicines?

We apologize for not labeling clearly. Yes, **Supplementary Figure 15c** illustrates the tissue distribution of LRT nanomedicines. Since the drug was encapsulated only within the interlayers of LDH rather than on the outer surface, the nanomedicine's surface charge or morphology is thought to be less affected by drug loading. The existing literature has demonstrated that the type of drugs loaded in the carriers will not affect its body distribution (*Accounts of Chemical Research*, 2013, 46(3): 622-631.). Thus, we only evaluated the tissue distributions of the most typical LRT nanomedicine, in accordance with the common practice in nanomedicine studies (*Nature Nanotechnology*, 2023, DOI: 10.1038/s41565-023-01447-7; *Nature Nanotechnology*, 2021,16(11): 1271–1280; *Nature Communications*, 2022, 13(1): 2834).

(2) Authors claim that “(Supplementary Figure 14c) a high accumulation rate of these nanoparticles in the tumor region” (lines 348-349). However, Supplementary Figure 14c shows a rather poor accumulation in the tumor, especially as compared to liver and spleen, and being in a similar range to other tissues.

We apologize for the imprecise wording in our previous description. "the high accumulation rate" refers to the fact that LRT exhibits a greater tumor accumulation rate than other nanomedicines reported in the literatures. To be more accurate, we have revised the original text as follows: " As presented in **Figure 5a, and Supplementary Figure 15a, b**, the ICG fluorescence signal show an extensive overlap with Luciferin signal in the tumor region, which peaks in 1 h post-injection, revealing a 9.4% $ID \cdot g^{-1}$ of tumor accumulation by measuring the amount of Al element in main organs and tumor tissues using ICP-OES (**Supplementary Figure 15c**). Compared with the rather low tumor accumulation of most of the reported nanomedicines (mostly below than 5 % $ID \cdot g^{-1}$), the enrichment rate of LRT in the orthotopic lung tumors has been substantially elevated." (Line 346-352) Additionally, as far as lung tissue being concerned in this study, LRT displays enhanced accumulation in orthotopic lung tumors, highlighting the effective passive targeting of LRT to tumor tissues.

(3) This also raises some concerns about liver and spleen toxicity, and therefore some additional data and discussion on this aspect should be included.

We greatly appreciate your insightful concern. **According to your suggestion, additional data and discussion about liver and spleen toxicity had been added in the revised manuscript and supporting information.** We evaluated the metabolic behavior of LRT *in vivo* by collecting and measuring the amount of Al elements in mouse feces and urine using ICP-OES. The results, as depicted in **Supplementary Figure 19**, indicate that approximately 70% of LRT was excreted by metabolism within a 7-day period through urine and feces, manifesting that most LRT can be efficiently cleared up from the body within a relatively short timeframe.

Supplementary Figure 21. Accumulated AI (in faeces and urine) excretion out of the mice body after the administration of LRT for different durations.

Moreover, we further investigated whether LRT accumulated in liver and spleen can trigger an immune response or not. To this end, we administered LRT to tumor-free mice *via* intravenous tail injections on three separate occasions, with the injections being spaced every other day. Subsequently, serum samples were collected on the eighth day, and concentrations of TNF- α and IL-12p70 were quantified through ELISA assay. As demonstrated in **Supplementary Figure 20a**, no significant elevations in cytokine levels can be observed in the tumor-free mice, thereby indicating that LRT can hardly trigger immune response in mice without tumors.

Supplementary Figure 20a. TNF- α and IL12p70 levels in sera from the mice without tumors after LRT treatment. The LRT group were treated with LRT three times and the sera were harvested on day 8. $N=4$. Statistical significances were calculated *via* Student's t test.

Additionally, major organs of these treated mice were hematoxylin and eosin (H&E)-stained. **Supplementary Figure 20b** reveals no significant injuries to major organs, thus further substantiating the safety of the LRT treatment.

Supplementary Figure 20b. H&E staining of major organs (heart, liver, spleen, lung, and kidney) harvested from the mice without tumor. The LRT group were treated with LRT three times and the organs were harvested on day 8. Scale bar, 100 μ m.

The inherent safety of LRT treatment can be attributed to the pH-responsive release nature of the cargos from LRT nanomedicine. As depicted in **Figure 1j**, the encapsulated drugs within LRT are exclusively released within mildly acidic environments, leading to the critical expression upregulation of CRT and downregulation of CD47 in cancer cells for effective immune activation. Importantly, as the drugs remain encapsulated in LRT in normal tissues, LRT treatment will not induce significant secretions of inflammatory cytokines in neutral normal organs and tissues.

Figure 1j. Release profiles of TTA-Q6 and RRX-001 from a nanocarrier in the phosphate buffer saline (pH 6.5 and 7.4), $N = 3$.

Finally, we also explored the potential long-term toxicity of LRT, as illustrated in **Supplementary Figure 21**. A four-week observation period reveal a gradual increase in the body weights of all mice, with no significant differences observed among the groups (**Supplementary Figure 21a**). Furthermore, hematological parameters (**Supplementary Figure 21b**) and H&E staining of main organs (**Supplementary Figure 21c**) harvested from mice across all groups exhibit no significant abnormalities on day 28, thus validating the favorable histocompatibility of LRT nanomedicine.

Supplementary Figure 21. *In vivo* safety evaluation of LRT nanomedicine. (a) Time-dependent body-weight curves of normal mice after treatment with PBS or PBS containing different doses of LRT nanosheets during an observation period of 28 days. Statistical significances were calculated *via* Student's t test. n.s., not significant. (b) Hematological parameters of normal mice after various treatments on day 28. (c) H&E staining of major organs (heart, liver, spleen, lung, and kidney) harvested from normal mice after different treatments on day 28. Scale bar, 200 μm.

7. Authors indicate that the nanodrugs had a rather short half-life (27 min) (Supplementary Figure 15); is the same for all nanomedicines shown here? If so, and treatments involve only three injections every other day, it is expected that very little amount of nanomedicines reach the tumor. Because the *in vivo* assays show remarkable antitumor activity following LRT treatment (Fig. 5), both effective and durable responses, it is suggested the LRT induces rapidly responses that were not present only following an ER stress, or LT and LR treatments. Fig. 6d shows a similar percentage of CD45⁺ leukocytes in the tumors of different treatments, but LRT-8 promotes a very high presence of CD45⁺ leukocytes (over 85%) in the tumor. How many LRT jabs were

given in LRT-8 in comparison to LRT-2, and to the other treatments (LDH, LR, LT)?
Why do you choose this timing?

Response: We are extremely grateful to the Reviewer's careful review of our paper. The following is our detailed responses to each point of the comment:

(1) Authors indicate that the nanodrugs had a rather short half-life (27 min) (Supplementary Figure 15); is the same for all nanomedicines shown here?

As pointed out by the Reviewer, the half-life of LRT was approximately 27 minutes, consistent with most reports on submicron-sized inorganic particles (*Nature Communications*, 2022, 13(1): 2834; *Nature Communications*, 2021, 12(1): 3393). Although the half-lives of LR/LT/LDH were not directly observed, it's worth noting that the drug encapsulation within the interlayers of LDH will not significantly affect the half-life due to minimally changed surface properties and overall morphology of the nanomedicines by drug loading (*Accounts of Chemical Research*, 2013, 46(3): 622-631). Consequently, the half-life of these nanomedicines would likely remain mostly unaffected. A large number of previous literatures have also proved that a majority of drug carriers and drug delivery systems share the similar half-lives from tens of minutes to several hours in blood stream, and the half-life assessment on one most representative nanomedicine is a common practice in nanomedicine (*Nature Nanotechnology*, 2023, DOI: 10.1038/s41565-023-01447-7; *Nature Nanotechnology*, 2021, 16(11): 1271–1280; *Nature Communications*, 2022, 13(1): 2834).)

(2) If so, and treatments involve only three injections every other day, it is expected that very little amount of nanomedicines reach the tumor.

The administered dosage of LRT nanomedicine is 40 mg/kg, which is relatively large in the realm of nanomedicines (*Nature Communications*, 2022, 13(1): 2834). Moreover, significantly the 9.4% ID•g⁻¹ tumor enrichment rate of LRT is higher than most reported values of nanomedicines (*Nature Reviews Materials*, 2016, 1(5): 1-12). The *in vivo* treatment shows significant inhibition on tumor growth (**Figure 5b**), proving that this dosage and the robust tumor accumulation rate have resulted in

effective treatment against the lung tumor.

(3) Because the *in vivo* assays show remarkable antitumor activity following LRT treatment (Fig. 5), both effective and durable responses, it is suggested the LRT induces rapidly responses that were not present only following an ER stress, or LT and LR treatments. Fig. 6d shows a similar percentage of CD45⁺ leukocytes in the tumors of different treatments, but LRT-8 promotes a very high presence of CD45⁺ leukocytes (over 85%) in the tumor. How many LRT jabs were given in LRT-8 in comparison to LRT-2, and to the other treatments (LDH, LR, LT)? Why do you choose this timing?

As noted by the Reviewer, LRT induced rapid responses that are not achievable through ER stress alone or individual LT- or LR-only treatment. It is widely acknowledged that elevating CRT in cancer cells is often combined with the downregulation of CD47 expression in cancer immunotherapies, because cancer cells tend to evade macrophage phagocytosis by increasing CD47 expression on cell surface (*Science Translational Medicine*, 2010, 2(63): 63ra94). In the case of LT treatment, only CRT is upregulated in cancer cells, leading to the non-sustained activation of macrophages (**Figure 6e**). On the other hand, only LR-treatment solely triggers CD47 downregulation in cancer cells, failing to activate DC cell maturation for effective antigen presentation (**Figure 6e**). In contrast, LRT treatment achieves the dual effects of upregulating CRT expression and downregulating CD47 expression in cancer cells. This integrated therapeutic approach result in much enhanced therapeutic outcomes compared to the separate LR-only or LT-only treatment.

As demonstrated in **Figure 6d**, the LRT-8 group demonstrates upregulated expression of CD45⁺ leukocytes within the tumor compared to the LR and LT groups. This enhancement is attributed to the notable infiltration of T cells in the LRT-8 group, resulting in the increased proportion of immune cells within the tumor. The absence of DC and T cell activation in the LR group results in minimal changes in the T cell population, thereby maintaining a relatively low immune cell proportion similar to that of the control group (**Figure 6e**). While the LT group exhibits considerable DC cell maturation, and a corresponding increase in the proportion of M2-type BMDMs,

leading to the immune-suppressive modulation of the tumor immune microenvironment, and the resultantly lowered T cell count and insignificant infiltration of T cells compared to the LRT group.

For better clarity and precision, we have provided the numbers of administrations for each group in **Table R3**. To prevent potential ambiguity, we have updated the labels of the groups in Figure 6 to LDH-8 days, LR-8 days, and LT-8 days.

Table R3 The number of administrations and time durations of each group.

Groups	Number of administrations	Time durations
LDH	3	8 days
LR	3	8 days
LT	3	8 days
LRT-8	3	8 days
LRT-2	1	2 days

Considering the prolonged period of T cell activation, we selected day 8 to monitor the tumor immune microenvironments among the groups, and to noticeably observe the changes in the tumor immune landscape. In contrast, innate immune activation is much faster, therefore a two-day time duration was used in the LRT treatment group, to serve as a control in comparison to the LRT-8 day group. **Figure 6e** reveals that the LRT- has triggered anti-tumor activity of macrophages within two days, while the anti-tumor immune response of T cells became activated in eight days. This evidences the effect and advantage of the combined activations of both innate and adaptive immunity by LRT treatment.

Figure 6. Mechanism study of antitumor immunotherapy by LRT nanomedicine. (c) High-dimensional data analysis on tumor-infiltrating leukocytes from different groups displaying x-shift clusters projected on two Opt-SNE dimensions. The overlay plot shows concatenated events from all 24 samples, while the density plots show the differential population distributions among representative individual samples. (d) Proportions of CD45⁺ leukocytes in the whole tumors of different groups. (e) Quantifications of various kinds of immunocells in x-shift clusters in percentages over total CD45⁺ cells after diverse treatments.

8. There is some variability in the schedules followed for the *in vivo* experiments. It is not clear enough what schedule was followed in each experiment. This should be specified and any variation could be commented and justified. Apparently, authors conducted exhaustively *in vivo* experiments with different nanomedicine treatments, but the results sometimes refer to all the treatments and sometimes to only LRT treatments. In this regard, a question lies in whether different treatments *in vivo* could lead to similar results, such as those seen *in vitro*. Authors claim a very nice mechanism of action in the Discussion section, but this is based basically in correlative data, but causative approaches should be achieved in order to further demonstrate this mechanism.

Response: We are extremely grateful to the Reviewer for this valuable suggestion and apologize for any confusion caused by our unclear description. The following is the point-by-point response to each comment:

(1) There is some variability in the schedules followed for the *in vivo* experiments. It is not clear enough what schedule was followed in each experiment. This should be specified and any variation could be commented and justified.

In the revised manuscript, the schedules of *in vivo* experiments have been clearly marked, such as in **Figure 5c-e**. The schedules of each experiment *in vivo* is depicted in **Figure R9**.

Figure R9. The schedule followed in each experiment *in vivo*.

(2) Apparently, authors conducted exhaustively *in vivo* experiments with different nanomedicine treatments, but the results sometimes refer to all the treatments and sometimes to only LRT treatments. In this regard, a question lies in whether different treatments *in vivo* could lead to similar results, such as those seen *in vitro*.

Thank you for the comment. **Figure 5** and **6** give the extensive experimental evaluations of all nanomedicines. Through flow cytometry, we assessed the tumor immune microenvironment changes by all nanomedicine treatments, which demonstrates that LRT can induce a comprehensive activation of the anti-tumor immune response. Nevertheless, though the *in vitro* treatment can be studied as extensively as possible, but the treatments *in vivo* were selectively scheduled and arranged based on the outcomes of *in vitro* treatment. *In vitro* treatments of ineffective or less effective outcomes would not be adopted for the following long-lasting and high-cost *in vitro* treatments, which is the common practice in the nanomedicine studies.

(3) Authors claim a very nice mechanism of action in the Discussion section, but this is based basically in correlative data, but causative approaches should be achieved in order to further demonstrate this mechanism.

Thanks for your constructive feedback. To further address this concern, two additional experiments were designed to verify two important conclusions mentioned in the Discussion section.

To show the roles of CRT upregulation in macrophages activation and DCs maturation, the transwell system was pre-coated with CRT antibodies. LLC cells were then seeded into the upper chamber and treated with LT for 16 hours. After removing the spent medium, macrophages or DCs were seeded into the lower chamber, as depicted in **Figures 3a or 4a**. Fresh medium supplemented with CRT antibodies was added. Following a 24-hour co-incubation, the supernatant was collected to measure IL-6 concentrations, a common activation marker for both macrophages and DCs, using an ELISA assay. As indicated in **Figure R10**, macrophages in the antibody-supplemented group secreted lower levels of IL-6 compared to the group without antibody-supplemented, confirming that CRT upregulation plays a significant role in macrophage activation. Similar results and conclusions were observed in the DC experiments, as shown in **Figure R11**.

Concentrations of IL-6 secreted by the macrophages (**Figure R10**) and DCs (**Figure R11**) stimulated with LT-treated LLC cells.

In addition, to show the important role of T cells for lung tumor immune therapy by LRT, we assessed the immunotherapeutic effects of LRT on both wild-type and Rag1-deficient mice. First, the orthotopic lung tumors were established in Rag1 deficient and wild-type C57/BL/6J mice through injecting LLC cells *in situ*. After three LRT administrations, the tumor growth behavior was evaluated by Micro-CT imaging. To better observe the lung tumor progression of mice in every group, lung 3-dimensional reconstruction and corresponding CT transverse section images are depicted in **Supplementary Figure 18a**, and the **Supplementary Figure 18f and g** are the lung volume calculated from the 3D reconstruction. As shown in **Supplementary Figure 18**, the Rag1^{-/-} mice show the faster cancer progression and the shorter survival time than wild-type mice, fully demonstrating the import role of T cells by LRT therapy.

Supplementary Figure 18. *In vivo* immunotherapeutic outcomes by various treatments for orthotopic Lewis lung tumor model on C57BL/6J mice ($N = 6$). (a) Representative lung 3-dimensional reconstruction and corresponding CT transverse section images from each group after varied treatments as indicated. (f, g) Quantifications of the healthy lung volumes by computational extraction through 3D reconstruction. The body weight fluctuation (h) and Kaplan-Meier survival curves (i) in between of LLC tumor-bearing mice after varied therapy.

9. Can authors discuss or make a comparison between the LRT treatment and the action of CD47 and calcium channel inhibitors, used separately or in combination, in cancer? What makes LRT a better option for cancer treatment than the use of CD47 and calcium channel inhibitors separately? Concerns about toxicity, safety and tumor accumulation should be critical, and these aspects are not clear at the moment. Safety could be a major

issue if drugs were accumulated mainly in liver and spleen.

Response: Thanks for your constructive and helpful suggestions. The following is the point-by-point response to each comment:

(1) Can authors discuss or make a comparison between the LRT treatment and the action of CD47 and calcium channel inhibitors, used separately or in combination, in cancer? What makes LRT a better option for cancer treatment than the use of CD47 and calcium channel inhibitors separately?

To address your concern, we redesigned and conducted animal experiments to assess the therapeutic effects of LRT and the combined TTA-Q6+RRX-001 administration in *in situ* lung tumor. As shown in **Supplementary Figure 18**, the LRT show higher therapeutic efficacy than administrating TTA-Q6 + RRX-001. The advantages of LRT over the combined TTA-Q6+RRX-001 treatment are as follows:

(1) **Enhanced Tumor accumulation:** LRT demonstrates a notably higher tumor enrichment efficiency than conventional small molecular drugs. As shown in **Supplementary Figure 15**, approximately 9.4% ID•g⁻¹ of LRT accumulated within the tumor tissue, significantly higher than other reported nanomedicines (mostly below than 5% ID•g⁻¹), as documented in *Nature Reviews Materials* (2016, 1(5): 1-12). In contrast, even the most efficient tumor-targeted antibody-drug conjugates (ADC) presently available exhibit less than 0.1% tumor accumulation (*Clinical Cancer Research*, 2011, 17(20): 6389-6397; *Journal of Cellular Physiology*, 2019, 234(5): 5628-5642). Thus, the efficient intra-tumor enrichment of LRT is responsible for its enhanced therapeutic effectiveness compared to LDH+RRX-001+TTA-Q6.

(2) **Improved Biosafety through pH Responsiveness:** LRT's favorable pH responsiveness contributes to its enhanced biosafety profile. It is well-established that immunotherapies based on CD47 blockade can result in severe anemia among patients, primarily due to the high CD47 expression on red blood cells that shields them from immune cell phagocytosis in circulation. This limitation poses a substantial barrier to the clinical application of CD47 ICB therapies (*New England Journal of Medicine*,

2018, 379(18): 1711-1721). The pH-responsive behavior of LRT ensures the selective release of RRX-001 and TTA-Q6 within the tumor microenvironment, consequently minimizing potential side effects. This is corroborated by the observations in **Supplementary Figure 18**, where the administration of TTA-Q6 + RRX-001 led to a significant decrease in mice body weight, indicative of pronounced adverse effects.

Supplementary Figure 18. *In vivo* immunotherapeutic outcomes by various treatments for orthotopic Lewis lung tumor model on C57BL/6J mice ($N = 6$). (a) Representative lung 3-dimensional reconstruction and corresponding CT transverse section images from each group after varied treatments as indicated. (d, f) Quantifications of the healthy lung volumes by computational extraction through 3D reconstruction. The body weight fluctuation (h) and Kaplan-Meier survival curves (i) in between of LLC tumor-bearing mice after varied therapy.

(2) Concerns about toxicity, safety and tumor accumulation should be critical, and these

aspects are not clear at the moment. Safety could be a major issue if drugs were accumulated mainly in liver and spleen.

As shown in **Figure 5a**, LRT accumulates more efficiently in orthotopic lung tumors than in normal lung tissue, demonstrating its passive targeting effect to tumor tissues, which minimized the concerns about the safety of nanomedicine. Currently, it's inevitable that nanomedicines concentrate largely in the liver and spleen, and the development of non-hepatic drug delivery systems remains a challenge, necessitating persistent efforts from the academic community (*Biomaterials*, 2018, 174, 41-53; *Nature Reviews Materials*, 2023, 8(4): 282-300).

Thank you for your insightful reminder for the toxicity of LRT. In the revised manuscript, additional experiment results for the safety of LRT have been added, including metabolism, short toxicity as well as long-term toxicity. The detailed discussions have been provided in the response to **Questions 6(3)**.

10. A scheme summarizing the putative mechanism of action suggested by the authors for LRT, together with the pieces of evidence supporting each step, could be very helpful.

Response: We thank the Reviewer for this valuable suggestion. Based on the framework of original **Scheme 1**, we have integrated more corroborative evidences for each step within the mechanism diagram. This modified comprehensive mechanism diagram is visually depicted below (**Supplementary Figure 29**).

Supplementary Figure 29A. Schematic diagram of the LRT therapy and relative evidence supporting each step: cancer cells part.

Supplementary Figure 29B. Schematic diagram of the LRT therapy and relative evidence supporting each step: immune system part.

Table R4. Overview of the response of the question of Reviewer 3.

Order	Question	Response	Where
1	The fate of nanomedicines in vivo .	[x] Explanation [ ] Supplementary experiments [x] Modify text and figure	Line 398-399 in the manuscript Pages 16 in the Supplementary Information Pages 29-31 in this Response
2	The further CRT and CD47 evaluation by CLSM, WB or Flow cytometer.	[x] Explanation [x] Supplementary experiments [ ] Modify text	Pages 21-33 in this Response
3	The detail ways that AMs kill tumor cells.	[x] Explanation [x] Supplementary experiments [ ] Modify text	Pages 34-35 in this Response

Order	Question	Response	Where
4	Where are the cell death percentages?	[x] Explanation [ ] Supplementary experiments [x] Modify text	Pages 13 in the Supplementary Information Pages 35-36 in this Response
5	The advantages of LRT compared to LT or LR.	[x] Explanation [ ] Supplementary experiments [x] Modify text	Pages 11-12 in the manuscript Pages 36-37 in this Response
6	The tumor accumulation and biosafety of LRT.	[x] Explanation [x] Supplementary experiments [x] Modify text	Line 411-424 in the manuscript Pages 24-25 in the Supplementary Information Pages 38-43 in this Response

Order	Question	Response	Where
7	The fate of nanomedicines in vivo and the time point of some experiment.	[x] Explanation [ ] Supplementary experiments [x] Modify text	Line 346-352 in the manuscript Pages 44-47 in this Response
8	The schedules followed for the in vivo experiments and the causative approaches should be achieved.	[x] Explanation [x] Supplementary experiments [x] Modify text	Line 390-405, 522 in the manuscript Pages 23 in the Supplementary Information Pages 47-51 in this Response
9	The therapeutic efficacy of LRT and small molecular drug and the biosafety of LRT.	[x] Explanation [x] Supplementary experiments [x] Modify text	Line 390-399 in the manuscript Pages 23 in the Supplementary Information Pages 51-54 in this Response

Order	Question	Response	Where
10	A scheme of mechanism and related evidence should be provided.	[ ] Explanation [ ] Supplementary experiments [x] Modify text and figure	Line 582-583 in the manuscript Pages 31-32 in the Supplementary Information Pages 54-56 in this Response

Reviewer #4: In this manuscript Guo et al report the anti-tumor effects of a nanomedicine delivering system combining T-type calcium channel inhibitor (TTA-Q6) and a small molecule inhibitor of CD47 (RRX-001). They observed antitumor effects of this nanomedicine (LRT) in a mouse LLC lung cancer model, and they claim such antitumor effects are mediated *via* LRT-induced endoplasmic reticulum stress in LLC cells and the transport of calreticulin onto the tumor cell membrane, which activates anti-tumor functions of alveolar macrophages as well as antigen presentation by dendritic cells that favors anti-tumor adaptive T cell response and memory. While this is an interesting potential anti-cancer therapeutic strategy, the key conclusions in this study are not convincingly supported by current experimental data. The reviewer has the following major and minor comments:

Major comments:

1. The authors observed an anti-tumor effect of LRT *in vivo*. They also observed AMs and DCs stimulation *in vitro* by LT/LR/LRT-treated LLC cells. However, the significance of AM activation and DC maturation in this *in vivo* antitumor effect remains speculative and are not formally addressed with experiments. In this regard, the authors need to analyze the phenotypic and functional changes in AMs and DCs in tumor-bearing lung tissues, before and after nanomedicine treatment.

Response: We thank the Reviewer for this valuable suggestion. In addition to our *in vitro* experiments, we conducted further investigations to explore phenotypic and functional changes of AMs and DCs *in vivo*. We employed flow cytometry and multiplex immunohistochemistry to analyze the immune microenvironment modulations by the nanomedicine. After administrating the nanomedicine, we collected orthotopic lung tumors and processed them into single-cell suspensions for subsequent multi-color flow cytometry analysis. The utilization of the x-shift algorithm allowed us

to define 23 distinct subpopulations of CD45⁺ cells across the six groups, as demonstrated in **Figure 6c-e**. Remarkably, LRT treatment resulted in a significant increase in the populations of M1 phenotype AMs and matured DCs compared to the control groups without the nanomedicine treatment. **This evidences the notable activation of anti-tumor immunity in aspects of AMs and DCs induced by LRT therapy *in vivo*.**

Figure 6. Mechanism study of antitumor immunotherapy by LRT nanomedicine. (c) High-dimensional data analysis on tumor-infiltrating leukocytes from different groups displaying x-shift clusters projected on two Opt-SNE dimensions. The overlay plot shows concatenated events from all 24 samples, while the density plots show the differential population distributions among representative individual samples. (d) Proportions of CD45⁺ leukocytes in the whole tumors of different groups. (e) Quantifications of various kinds of immunocells in x-shift clusters in percentages over total CD45⁺ cells after diverse treatments.

In addition to quantifying the changes in immune microenvironment, the spatial distribution of immune cells and tumor cells offers a novel biomarker for predicting immunotherapeutic responses. To probe the spatial immune mechanisms underlying LRT therapy in-depth, we conducted fluorescent multiplex immunohistochemical analysis on the tumor tissues from both control and LRT-treated groups on day 8. **Figure 6f** distinguishes different cell types through the co-expression of various markers, such as M1 AMs (F4/80⁺CD11c⁺CD86⁺) and matured DC cells (F4/80⁻

CD11c⁺CD86⁺). More immunoactivation signals such as CD8 and CD86 can be identified in the **Figure 6f**, demonstrating the anti-tumor immune response activation by LRT.

Figure 6f. Representative mIHC images from the lung tumor sections of the mice with/without LRT treatment for 8 days. Scale bars: 100 μ m.

In addition, we employed HALO platform for subsequent analysis according to the mIHC results by focusing on the locations of defined immune cells from the tumor border, the minimum distance between cancer cells and immune cells, and the number of immune cells surrounding cancer cells.

According to the infiltration outcomes (Supplementary Figure 25 and Figure 6g), the LRT groups exhibit a more prevailing distributions of M1 AMs and matured DCs within the tumor tissue, characterized by greater average infiltration depth than the control groups. Such an intratumoral distribution of the

immune cells evidences immune activation within the tumor microenvironment (TME), a key for solid tumor immunotherapy.

S25

Supplementary Figure 25. (a) Images obtained after cell segmentation and cancer cell identification for mIHC image. The highlighted cells are cancer cells, and the gray cells are other cells. (b) Analysis area at different distances from the tumor border generated from infiltration analysis. (c) Images obtained after cell segmentation and identification for mIHC image. (d) The mIHC image and infiltration band at different depth. Scale bar: 100 μm . **Figure 6g.** Immune cells infiltration analysis across tumor border. The columnar stacking diagrams towards the left y axis show the distribution of immune cells at different locations of infiltration from tumor border. The scatterplots towards the right y axis represents the average infiltration depth.

The analysis of the shortest distances between cancer cells and immune cells is presented in **Supplementary Figure 26** and **Figure 6h**. **Notably, the most M1 AMs and matured DCs are located within a circle of 100-micron in radius from the cancer cells of interest in LRT groups, which enables the effective cancer cell killing by the immune cells.** In contrast, over 70% of cancer cells can be found to be more than 100 microns away from their nearest M1 AMs and matured DCs in the control groups, suggesting the less effective damages to cancer cells by the immune cells at such over-large distances.

Supplementary Figure 26. Representative mIHC staining images of lung tumor in control and LRT-8 days groups, and corresponding spatial distribution of the nearest distance between cancer cells and immune cells. The white dotted lines connect the cancer cell-immune cell pairs generated from the nearest neighbor analysis. The nearest neighbor analysis takes the cancer cell as the center and finds the closest immune cell to that cancer cell, so the result may be that one immune cell is the closest cell to multiple cancer cells. It also occurs that some immune cells are not selected as nearest cells by any other cancer cells. Scale bars, 100 μm . **Figure 6h.** The nearest neighbor analysis for each cancer cell by mIHC staining. The columnar stacking diagrams towards the left y axis show the spatial distribution of immune cells of the shortest distance from cancer cells. The scatterplots towards the right y axis represent the average shortest distance between cancer cells and immune cells.

To provide a more objective assessment of treatment efficacy and prognosis, we conducted proximity analysis to quantify immune cells in close proximity to cancer cells (**Supplementary Figure 27**). Specifically, we counted the number of immune cells within a circle of 100 microns in radius from each cancer cell as circle center to evaluate the immune response (**Figure 6i**). Remarkably, over 50% of cancer cells in the LRT group are positioned by at least one immune cell in their vicinity of 100 microns in distance. This signifies substantially higher immune cell-to-cancer cell ratios than

the control group, further elucidating the enhanced therapeutic effect of LRT.

Supplementary Figure 27. Representative mIHC staining images of lung tumor in control and LRT-8 days groups, and corresponding quantitative data generated from proximity analysis (rose plot). Proximity analysis first matches immune cells to the nearest cancer cells to form cell pairs, and then analyzes the number of immune cells matched to that cancer cell within a radius of 100 μm with the cancer cell as the center, so the result may be that one cancer cell is the closest cancer cell to multiple immune cells. It also occurs that some cancer cells are not selected as nearest cells by any other immune cells. The white dotted lines connect the cancer cell-immune cell pairs generated from the proximity analysis. Scale bars, 100 μm . **Figure 6i.** Proximity analysis of immune cells towards each cancer cell by mIHC staining. The columnar stacking diagrams towards the left y axis show the number and corresponding proportion of immune cells in the vicinity of cancer cells (within 100 microns away from that cancer cell center). The scatterplots towards the right y axis represent the average numbers of the immune cells in the vicinity of cancer cells.

In conclusion, LRT treatment not only effectively induces the M1-polarization of AMs and the maturation of DCs *in vivo*, but also enhances the infiltration of AMs and DCs.

2. In Figure 3a, c and Supplementary Figure 9, the effects of LR, LT and LRT on AM activation are similar. Does this mean a redundant effects of RRX-001 and TTA-Q6 in this culture system, or alternative mechanisms are involved in AM activation? The authors need to use CRT blocking or equivalent experiment to show the roles of CRT upregulation in AM activation (Figure 3). Similarly, the impact of CRT upregulation on DC maturation (Figure 4) needs to be experimentally addressed.

Response: Thanks for your insightful comment. The following is the point-by-point response to each comment:

(1) Why the effects of LR, LT and LRT on AM activation are similar?

Previous research suggests that both CD47 downregulation and CRT upregulation taking place in cancer cells will lead to M1-type macrophage polarization within a short period of time (*Proceedings of the National Academy of Sciences*, 2012, 109(17): 6662-6667; *Trends in Immunology*, 2018, 39(3): 173-184; *Cancer Cell*, 2023, 41(6):1014-1016; *Nature Immunology*, 2022, 23(4): 487–500). To qualitatively assess whether LR/LT/LRT-treated cancer cells can effectively trigger an anti-tumor response in macrophages or not, we employed excess quantities of nanomedicines in the *in vitro* experiments to thoroughly stimulate alveolar macrophages (AMs), which led to the activations of all AMs in use regardless of the types of the nanomedicines, LR, LT or LRT- (**Table S2**). Notably, in the revised manuscript, we re-evaluate the effects of LR, LT and LRT on macrophages activation at a lowered dosage of nanomedicines. It can be found that in this case, LRT induces significantly stronger activation effect on macrophages than LR and LT, further demonstrating the above perspective (**Figure 3b and d**).

(2) Does this mean a redundant effects of RRX-001 and TTA-Q6 in this culture system, or alternative mechanisms are involved in AM activation?

In fact, there no redundant effects in this anti-tumor strategies. As a matter of fact, studies on cancer immunotherapies focusing on elevating CRT in cancer cells are

frequently combined with the downregulation of CD47 expression, because cancer cells tend to evade macrophage phagocytosis by increasing CD47 expression if they express CRT on cell surface (*Science Translational Medicine*, 2010, 2(63): 63ra94). Therefore, the combination of RRX-001 and TTA-Q6 will be more effective to achieve a long-term anti-tumor effect. As illustrated in **Figure 6e**, LRT yielded a much pronounced effect on macrophages activation in comparison to LR and LT *in vivo*, owing to the combined regulations of CRT and CD47.

(3) The authors need to use CRT blocking or equivalent experiment to show the roles of CRT upregulation in AM activation (Figure 3). Similarly, the impact of CRT upregulation on DC maturation (Figure 4) needs to be experimentally addressed.

Thanks for your constructive suggestion. To confirm the roles of CRT upregulation in macrophage activation and DCs maturation, the transwell system was pre-coated with CRT antibodies. LLC cells were then seeded into the upper chamber and treated with LT for 16 hours. After removing the spent medium, macrophages or DCs were seeded into the lower chamber, as depicted in **Figures 3a and 4a**. Fresh medium added with CRT antibodies was added. Following a 24-hour co-incubation, the supernatant was collected to measure IL-6 concentrations, a common activation marker for both macrophages and DCs, using an ELISA assay. As indicated in **Figure R9**, macrophages in the antibody-added group secreted lower levels of IL-6 than the group without antibody addition, confirming that CRT upregulation plays a significant role in macrophage activation. Similar results and conclusions have been obtained in the DC experiments, as shown in **Figure R10**.

Concentrations of IL-6 secreted by the macrophages (**Figure R9**) and DCs (**Figure R10**) stimulated with LT-treated LLC cells.

3. The authors need also to exclude direct tumor cell cytotoxic effects of nanomedicine.

Response: We thank the Reviewer for this insightful suggestion. To assess the cytotoxicity of nanomedicine on tumor cells, we conducted a TUNEL assay to stain tumors immediately after the LRT treatment. As depicted in **Supplementary Figure 22**, following a single LRT treatment, only a slight amount of tumor cells displays TUNEL-positive signals. Even LRT-treated for three times, the proportion of TUNEL-positive tumor cells remains at a quite low level. The negligible cytotoxicity of the nanomedicine supports our conclusion that the tumor regression by LRT treatment is a result of the activation of the immune system's anti-tumor response.

Supplementary Figure 22. TUNEL staining of tumor sections after different treatment. Scale bars: 100 μm .

4. The authors claim that activated AMs can kill tumor cells directly *via* ADCC. However, in Figure 3e no exogenous antibodies were supplemented to the co-culture system. The authors need to clarify whether AMs kill LLC cells *via* ADCC.

Response: Thanks for your professional comment and sorry for our mistake. To address your concern, we stimulated macrophages with LRT-treated LLC cells and then co-cultured these activated macrophages with Hoechst 33342-labeled LLC cells in a 96-well V-bottom plate coated with IFN-gamma, TNF-alpha, IL-1beta and IL-6 antibodies, to shield the cancer cells from the potential killing by the cytokines. As shown in the **Figure R8**, in 24 hours of co-cultivation, the cell death rate of the LRT group is not much different from the control group but is significantly lower than those without antibody coating. This evidences that the observed cancer cell killing is not from the ADCC effect of the macrophages but due to the cytokines secreted by M1-type macrophages. In the revised manuscript, related mistakes have been corrected.

Figure R8. Killing effect for LLC cells of the stimulated macrophages, the cells were co-incubated in the V-bottom 96-well plate with/without coated with IFN-gamma, TNF-alpha, IL-1beta and IL-6 antibodies. Data are expressed as means \pm SD ($N = 4$).

5. In Figure 4e, the authors need to show T cell proliferation in all the five groups of DCs (Figure 4b,c). This will help to link the maturation of DCs to T cell activation.

Response: We are extremely grateful to the Reviewer for this valuable suggestion. T cell proliferation properties across all five groups of DCs are demonstrated in **Figure 4e**. Furthermore, the flow cytometric analysis of matured DCs in each group is depicted in **Figure 4d**. T cell proliferation can only be observed in the LT and LRT groups, effectively confirming the DC cell maturation in these groups, consistent with the conclusions drawn from **Figure 4b and 4c**.

Figure 4. *In vitro* BMDCs maturation induced by LLC upon LRT nanomedicine treatment. (a) Schematic illustration of BMDCs stimulation in a transwell system. LLC cells were placed in the upper chamber followed by various treatments for 16 h, then the resultant media containing nanomedicines were removed and BMDCs were cultured in the lower chamber. (b) Quantifications of the secretion levels of TNF- α , IL-1 β , IL-6, and IL-12p70 in the stimulated BMDCs suspensions. The data are expressed as means \pm SD ($N = 3$). (c) CLSM images of BMDCs after being co-cubated with LLC cells pre-treated with LDH/LR/LT/LRT. The cells were then stained with DiI and DAPI for lightening plasma membrane and nuclei, respectively. Scale bar, 30 μ m. (d) FACS plots of CD80 and CD86 expressions on BMDCs gated on CD11c⁺ cells. (e) Flow cytometric counts of CFSE-labeled T cells gated on CD3⁺CD8⁺ cells for T cell proliferation analysis.

6. The authors indicate that the *in vivo* anti-tumor effects of LRT involve anti-tumor effects of both activated AMs and T cells primed by DCs. They need to show the relative

contribution of AMs and DC-primed T cells in the anti-tumor effects of LRT *in vivo*. To do this, the authors can use Rag deficient mice to show the roles of anti-tumor AMs in the absence of T cells.

Response: Thank you for your constructive suggestion. Based on your professional suggestion, we assessed the immunotherapeutic effects of LRT on both wild-type and Rag-deficient mice. First, the orthotopic lung tumors were established in Rag1 deficient and wild-type C57/BL/6J mice through injecting LLC cells *in situ*. After three times of LRT administrations, the tumor growths were evaluated by Micro-CT. To better observe the lung tumor progression of mice in every group, lung 3-dimensional reconstruction and corresponding CT transverse section images are depicted in **Supplementary Figure 18a**, and the **Supplementary Figure 18f and g** give the lung volumes calculated from the 3D reconstruction. The Rag1^{-/-} mice showed faster cancer progression and shorter survival time than wild-type mice, fully demonstrating the import role of T cells during LRT therapy.

It's well known that macrophages don't possess long-term immunological memory effect, which is also supported by the less effective therapeutic outcomes shown in **Figure 5c** from the LR treatment, as LR therapy doesn't activate the antitumor immune response of T cells (**Figure 6e**). In summary, macrophages themselves can only inhibit tumor growth to a certain extent and in a short term, and adaptive T cell activation is necessary to achieve more effective and sustained efficacy in the tumor immunotherapy in LRT treatment.

Supplementary Figure 18. *In vivo* immunotherapeutic outcomes by various treatments for orthotopic Lewis lung tumor model on C57BL/6J mice ($N = 6$). (a) Representative lung 3-dimensional reconstruction and corresponding CT transverse section images from each group after varied treatments as indicated. (f, g) Quantifications of the healthy lung volumes by computational extraction through 3D reconstruction. The body weight fluctuation (h) and Kaplan-Meier survival curves (i) in between of LLC tumor-bearing mice after varied therapy.

7. “As shown in Supplementary Figure 19, the TNF- α and IL-12p70 levels in the serum of LRT-treated mice are largely upregulated as the typical markers of antitumor immune response”. High concentrations of systemic inflammatory cytokines in serum may cause damage to the body. Can LRT treatment induce these inflammatory cytokines in mice without tumors?

Response: We thank the Reviewer for these constructive comments. To address your concern, we administrated LRT to tumor-free mice through intravenous tail injections over three sessions, with each injection taking place every other day. In eight days post-injections, we collected serum samples and conducted ELISA assays to measure the concentrations of TNF- α and IL-12p70. As depicted in **Supplementary Figure 20a**, no significant increases in cytokine levels can be observed in the tumor-free mice group, indicating that LRT will not induce immune activation in healthy mice without tumors.

Supplementary Figure 20a. TNF- α and IL12p70 levels in sera from the mice without tumors after LRT treatment. The LRT group were treated with LRT three times and the sera were harvested on day 8. $N=4$. Statistical significances were calculated *via* Student's t test.

Additionally, we performed hematoxylin and eosin (H&E) staining on the major organs harvested from these mice. **Supplementary Figure 20b** illustrates no evident signs of inflammatory responses in the organs of healthy mice by the LRT treatments. These additional results further evidence the favorable biosafety of LRT treatment.

Supplementary Figure 20b. H&E staining of major organs (heart, liver, spleen, lung, and kidney) harvested from the mice without tumor. The LRT group were treated with LRT three times and the organs were harvested on day 8. Scale bar, 100 μm .

The good safety of LRT treatment can be attributed to the pH responsiveness of cargo releases from the LRT nanomedicine. As demonstrated in **Figure 1j**, the drugs encapsulated within LRT are selectively released in mildly acidic environments, leading to the crucial upregulation of CRT and downregulation of CD47 in cancer cells—a prerequisite for immune activation. Importantly, the drug release from LRT remains localized within the tumor environment, ensuring that the LRT treatment will not trigger inflammatory cytokines in mice without tumors.

Figure 1j. Release profiles of TTA-Q6 and RRX-001 from a nanocarrier in the phosphate buffer saline (pH 6.5 and 7.4), $N = 3$.

Minor comments:

1. Some conclusions are based on speculation but not experimental evidence and should be revised. For example: line 74 "...activates the antitumor effects of TAMs, which subsequently induces DC maturation" (is DC maturation directly related to/caused by TAM activation?), line 426 "because the CD47 downregulation will not induce significant DC maturation.", line 444 "...subsequently the CD8⁺ T cells will migrate to tumor sites for killing tumor cells...".

Response: We thank the Reviewer for the constructive suggestions and apologize for the unclear statement in the main text. Following your helpful advice, we have conducted a comprehensive review of the entire manuscript and made the following revisions:

(1) Line 74 "...activates the antitumor effects of TAMs, which subsequently induces DC maturation".

The maturation of DC cells is induced by cancer cells treated with LRT, which takes place independent of TAM. These pathways are distinct, as depicted in **Scheme 1**. To ensure clarity, the original text has been revised as follows: "which will activate the antitumor effect of tumor-associated macrophages (TAM), induces DC maturation, and activate T cell anti-tumor immune responses through the antigen presentation function of DC". (Line 76-78 in the revised manuscript)

Scheme 1. (a) Schematic diagram of LDH nanosheet synthesis and the subsequent drug loading. (b) Mechanism of antitumor immune responses against orthotopic lung tumors by TTA-Q6/RRX-001 co-delivery. (1) T-type calcium influx channel blockage by TTA-Q6, (2) calcium homeostatic deficiency leading to calreticulin overexpression in parallel with the CD47 downregulation by RRX-001, (3) ER stress induction due to the calcium deficiency, (4) calreticulin translocation towards the cell membrane.

(2) Line 426 “because the CD47 downregulation will not induce significant DC maturation.”

Thanks for your constructive suggestion and sorry for the inappropriate statement

in the main text. As you correctly highlighted, this statement lacks experimental data support and is unnecessary for the original text. We have removed this sentence from the revised manuscript. Thank you once again for your thorough reading and review.

(3) Line 444 “...subsequently the CD8⁺ T cells will migrate to tumor sites for killing tumor cells...”

To ensure scientific accuracy in the relevant descriptions, the original text has been revised as follows: “By profiling the tumor's immune microenvironment in 2 days post-LRT treatment, we observed a significant reduction in TAM-M2 populations compared to the control group, and the concurrent substantial population increases of TAM-M1 and matured DC. These findings suggest that the LRT nanomedicine can rapidly activate the antitumor functions of macrophages and DC within a short timeframe.”. (Line 481-485) In the revised version, we have eliminated conclusions drawn from speculative interpretations.

2. English writing and typos should be revised. For example, line 94 “clinical”?, line 524 “lymphopenia”

Response: Thank you for your thorough review, and we apologize for the errors. The typos have been thoroughly checked and corrected. Specifically, "Clinic" has been revised to "clinical" (Line 97), and "lymphopenia" has been rectified to "lymphocytes" (Line 564).

Table R5. Overview of the response of the question of Reviewer 4.

Order	Question	Response	Where
1	The AM activation and DC maturation in vivo should be evaluated.	[x] Explanation [ ] Supplementary experiments [ ] Modify text	Line 450-540 in the manuscript Pages 28-30 in the Supplementary Information Pages 61-66 in this Response
2	Show the role of CRT upregulation in AMs activation and DC maturation.	[x] Explanation [x] Supplementary experiments [ ] Modify text	Pages 67-68 in this Response
3	Exclude direct tumor cell cytotoxic effects of nanomedicines.	[x] Explanation [x] Supplementary experiments [x] Modify text	Line 440-449 in the manuscript Pages 26 in the Supplementary Information Pages 69 in this Response

Order	Question	Response	Where
4	The detail tumor killing pathways of AMs should be assessed.	[x] Explanation [x] Supplementary experiments [x] Modify text	Line 213-225, 265-267 in the manuscript Pages 69-70 in this Response
5	Other groups in Figure 4e should be provided.	[x] Explanation [ ] Supplementary experiments [x] Modify figure	Line 316 in the manuscript Pages 70-71 in this Response
6	Evaluate the antitumor effect of LRT in Rag deficient mice.	[x] Explanation [x] Supplementary experiments [x] Modify text	Line 400-405 in the manuscript Pages 23 in the Supplementary Information Pages 72-73 in this Response

Order	Question	Response	Where
7	Can LRT treatment induce these inflammatory cytokines in mice without tumors?	[x] Explanation [x] Supplementary experiments [x] Modify text	Line 417-424 in the manuscript Pages 24 in the Supplementary Information Pages 74-75 in this Response
8	Some conclusions should be revised to make it more accurate.	[ ] Explanation [ ] Supplementary experiments [x] Modify text	Line 76-78, 481-485 in the manuscript Pages 76-78 in this Response
9	English writing and typos should be revised.	[ ] Explanation [ ] Supplementary experiments [x] Modify text	Multiple places in the manuscript Pages 29-31 in this Response

REVIEWERS' COMMENTS

Reviewer #1 (Remarks to the Author):

All my concerns have been adequately addressed. Thank you!

Reviewer #2 (Remarks to the Author):

In the revised manuscript, the authors have addressed all the questions and comments raised by the reviewers from the first round review. I believe that the revised manuscript meet the requirement of Nature Communications, and I will recommend the publication of this manuscript in the current form.

Reviewer #3 (Remarks to the Author):

The authors have made a great effort in carrying out a more complete revised version of the original article, and have answered in a rather exhaustive and comprehensive way most of the queries. The present extensive study still seems to be a little skewed, with a somewhat biased interpretation of some data overlooking some cell types and biological processes. However, it is a thorough study and the in vivo data support a potent and long-lasting antitumor effect, suggesting in addition the generation of a long-term immunological memory.

Reviewer #4 (Remarks to the Author):

The authors have addressed all my concerns, and I don't have further questions.

Reviewer #1 (Remarks to the Author):

All my concerns have been adequately addressed. Thank you!

Response: We would like to thank you for your time and efforts in considering this work.

Reviewer #2 (Remarks to the Author):

In the revised manuscript, the authors have addressed all the questions and comments raised by the reviewers from the first round review. I believe that the revised manuscript meet the requirement of Nature Communications, and I will recommend the publication of this manuscript in the current form.

Response: Thank you very much for the positive comment and kind recommendation.

Reviewer #3 (Remarks to the Author):

The authors have made a great effort in carrying out a more complete revised version of the original article, and have answered in a rather exhaustive and comprehensive way most of the queries. The present extensive study still seems to be a little skewed, with a somewhat biased interpretation of some data overlooking some cell types and biological processes. However, it is a thorough study and the in vivo data support a potent and long-lasting antitumor effect, suggesting in addition the generation of a long-term immunological memory.

Response: We thank the reviewer for your time and energy in reviewing our work and providing constructive feedback.

Reviewer #4 (Remarks to the Author):

The authors have addressed all my concerns, and I don't have further questions.

Response: Thank you for your constructive comments, which are greatly helpful in improving the quality of this work.